# Phosphoproteomic analysis of neoadjuvant breast cancer suggests that increased sensitivity to paclitaxel is driven by CDK4 and filamin A

S. Mouron[1], M. J. Bueno [1], A. Lluch [2], L. Manso [3], I. Calvo[4], J. Cortes [5,6], J. A. Garcia-Saenz [7], M. Gil-Gil[8], N. Martinez-Janez[9], J. V. Apala[1], E. Caleiras[10], Pilar Ximénez-Embún [11], J. Muñoz[11], L. Gonzalez-Cortijo[12], R. Murillo[13], R. Sánchez-Bayona[3], J. M. Cejalvo[2], G. Gómez-López[14], C. Fustero-Torre [14], S. Sabroso-Lasa [15], N. Malats [15], M. Martinez[16], A. Moreno[17], D. Megias[18], M. Malumbres [19], R. Colomer [20,21] & M. Quintela-Fandino [1,21] ✉

Precision oncology research is challenging outside the contexts of oncogenic addiction and/or targeted therapies. We previously showed that phosphoproteomics is a powerful approach to reveal patient subsets of interest characterized by the activity of a few kinases where the underlying genomics is complex. Here, we conduct a phosphoproteomic screening of samples from HER2-negative female breast cancer receiving neoadjuvant paclitaxel ($N = 130$), aiming to find candidate biomarkers of paclitaxel sensitivity. Filtering 11 candidate biomarkers through 2 independent patient sets ($N = 218$) allowed the identification of a subgroup of patients characterized by high levels of CDK4 and filamin-A who had a 90% chance of achieving a pCR in response to paclitaxel. Mechanistically, CDK4 regulates filamin-A transcription, which in turn forms a complex with tubulin and CLIP-170, which elicits increased binding of paclitaxel to microtubules, microtubule acetylation and stabilization, and mitotic catastrophe. Thus, phosphoproteomics allows the identification of explainable factors for predicting response to paclitaxel.

Patient heterogeneity represents a major problem in clinical oncology therapeutics. Regarding breast cancer, massive efforts by international consortia have revealed a virtually unique genomic landscape for each patient[1–6]. In addition, with the exception of the HER2 subtype, and compared to other tumour types, most patients lack a clear oncogenic-addiction driver, which suggests that most HER2-negative breast cancers are the result of the accumulation of several non-sufficient, non-necessary oncogenic alterations[1–9]. This fact complicates the precision medicine framework of the biomarker-driven administration of targeted (or nontargeted) therapies in breast cancer. Taking advantage of phosphoproteomics, however, we have recently demonstrated that

considerably heterogeneous genomic landscapes converge into a discrete number of signalling aberrations, regardless of the presence or absence of mutations in the genes encoding for the aberrant signalling nodes[10]. A kinome-based taxonomy sorting ~200 TNBCs according to the hyperactivation of 6 kinases added considerable parsimony into the classification of this disease compared to a genomic-centred classification and allowed the identification of powerful therapeutic targeted doublets[10]. This concept is currently being tested in a prospective clinical trial (NCT04494958), suggesting that phosphoproteomic profiling could solve some limitations of gene-centric approaches on which most precision oncology applications rely.

More than 90% of breast cancers are diagnosed in a localized stage. In this stage, achieving a pathologic complete response (pCR) to neoadjuvant therapy is the most accurate factor to predict disease-free survival in the long term[11,12]. Over the last 30 years, the agents that have led to the greatest increases in pCR rates in HER2-negative breast cancer are the chemotherapeutic agents paclitaxel [for hormone-positive and triple-negative breast cancer (TNBC)][13] and carboplatin (for TNBC)[14]. Compared to targeted therapies and immuno-oncology drugs, research on predictive factors or resistance mechanisms against traditional cytotoxics is less active. Increasing data suggest that interrogating the proteome represents an exceptional resource for tackling biological processes or undertaking diagnostic, biomarker or drug discovery efforts[15–18]. We hypothesized that an interrogation of the phosphoproteome might inform about mechanistic traits not capturable in gene-centric layers[19,20], aiming to understand the basic kinome landscapes underlying the response to the cytotoxic agent paclitaxel in HER2-negative breast cancer. Paclitaxel is one of the most widely used agents against breast cancer, and, in contrast to the case for most targeted therapies, we currently lack specific predictive factors. We expected this approach to lead to the definition of predictive factors and an understanding of the mechanisms explaining paclitaxel sensitivity.

To that end, we conducted a phosphoproteomic screening of samples obtained from a clinical trial that compared paclitaxel against paclitaxel plus the antiangiogenic agent nintedanib in early HER2-negative breast cancer[21]. The screening led us to 11 candidate biomarkers (kinases and phosphorylated proteins). To transfer these results to potential biomarkers assessable with routine clinical tools, the candidate biomarkers were then tested in two independent early breast cancer datasets by immunohistochemistry. For two of them (filamin A and CDK4), the predictive association with the response to paclitaxel in TNBC was preserved. Subsequent experimental work allowed us to understand the mechanism of sensitization, which was driven by a functional axis linking CDK4, filamin A and CLIP-170; this mechanism enhanced sensitivity to paclitaxel but not to other chemotherapeutics. Thus, our approach allowed us to find predictive factors specific for the response to paclitaxel in HER2-negative breast cancer.

## Results

### Discovery patient set for phosphoproteomic analysis

Samples were obtained from the clinical trial NCT01484080, which enrolled 130 early HER2-negative breast cancer patients (29 TNBC, 101 hormone-positive). Patients were treated with single-agent weekly paclitaxel (Standard Arm) or single-agent oral nintedanib for 14 days followed by nintedanib plus weekly paclitaxel (Experimental Arm). The endpoint of the trial was the rate of achieving a pathologic complete response (pCR), determined in surgery and assessed by the Symmans and Pusztai residual cancer burden (RCB) method, where a pCR is equivalent to RCB = 0 [this classification includes both patients with no evidence of residual tumour in the breast or axillary nodes and patients with only residual noninvasive in situ carcinoma in the breast and no tumour in the axillary nodes after neoadjuvant treatment[11]]. Figure 1A shows the basic trial design and the number of harvested and valid samples (set at a minimum of 200 micrograms of purified protein isolated from a macrodissected tumour sample). Full trial results and patients' clinical characteristics are described elsewhere[21]. Although in the NCT01484080 trial patients were randomized 1:1, and thus the main clinical and pathologic characteristics of the patients were well balanced among both treatment arms, the cohort of patients with valid samples was slightly imbalanced (Supplementary Table 1): valid tumour samples from patients who received paclitaxel monotherapy ($N = 39$) were somehow smaller, with fewer involved axillary nodes, and of lower grade and replicative fraction than those from patients who received the combination treatment ($N = 46$); however, these differences did not reach statistical significance. The flow chart in

Fig. 1B depicts the steps followed and the samples used for biomarker discovery, confirmation, and experimental validation. In this study, we will refer exclusively to the baseline samples and their relationship with the response to paclitaxel; the effects of nintedanib on the phosphoproteome and their relationship with response or resistance against this agent will be reported elsewhere. These samples produced approximately 3 million spectra, of which 3834 unique phosphopeptides mapping to at least 1352 distinct phosphoproteins were identified (Supplementary Table 2 lists the proteomic data from the whole trial and by treatment arm). Matching the predominance of serine-phosphorylation sites in the proteome, we captured a predominance of serine over threonine phosphosites and a minority of tyrosine sites (Supplementary Table 2).

To search for potential differences in sample handling and/or preservation procedures across the different hospitals in which the samples were harvested that could result in signalling alterations (e.g., signalling changes in stress kinase pathways caused by tumour ischaemia[22]), hierarchical clustering was performed with the phosphopeptide intensity data matrix (Fig. 1C). Hierarchical clustering did not show that the samples seemed to be significantly grouped by the hospital in which they were harvested (Fig. 1C). Given the imbalance in certain patient characteristics when they were classified by treatment type (Supplementary Table 1), it was not surprising to observe a clustering of the samples by treatment arm (Fig. 1C). To further explore these clusters, we studied sample clustering by relying on other methodologies: consensus clustering[23] (Supplementary Fig. 1A–C) and the pvclust algorithm[24] (Supplementary Fig. 1D); however, we did not find a robustness of the association of the clusters with the trial arm (Supplementary Fig. 1A–C).

### Kinase and phosphopeptide enrichment among responders and nonresponders to paclitaxel

The functional impact caused by phosphorylation is unknown for most of the captured phosphorylation sites in high-throughput phosphoproteomic experiments. The advantage of phosphoproteomics over other -omic techniques, however, is that the phosphorylation status of all the encoded proteins is driven by a relatively low number of kinases (~500). We recently described an approach termed kinase set enrichment analysis (KSEA) that allows the mapping of phosphorylated peptides back to the kinase or kinases that can phosphorylate them: based on the affinity of the consensus sequences for each kinase, the main kinases that explain the phosphorylation status of a complex peptide mixture can be narrowed down[10]. By applying this technique and according to the amount of each detected phosphopeptide among two complex samples, the statistically significantly differentially functioning (hyper- or hypo-) kinases that account for each sample phosphoprofile can be determined. When applied to large sample sets (i.e., responders or nonresponders), we can infer which are the main kinases implicated in causing the differences between the sample sets' phosphoprofiles.

We compared the phosphoprofiles of the samples from patients who achieved a complete response to neoadjuvant therapy (pCR or RCB = 0; herein responders) versus those who did not (RCB > 0; herein nonresponders)[11,25]. This comparison was run in a data matrix comprising 2151 phosphopeptides that mapped to 1027 unique proteins. The comparison in the Standard Arm (Arm B, paclitaxel only) suggested that the activity of 5 kinases was enriched in the samples from responder patients compared to the samples from nonresponders (P70S6K, CDK4, PKC, AMPK1/2 and CaMK-IV; the KSEA plots are shown in Fig. 2A). When the same comparison was run considering all the trial samples together (i.e., mixing patients who received only paclitaxel and patients who received paclitaxel plus nintedanib; 2757/1252 phosphopeptides/unique proteins), P70S6K, CDK4 and AMPK1/2 activity enrichment—albeit present—lost statistical significance (Fig. 2B). A new kinase (HMG-CoA reductase kinase) was found to be

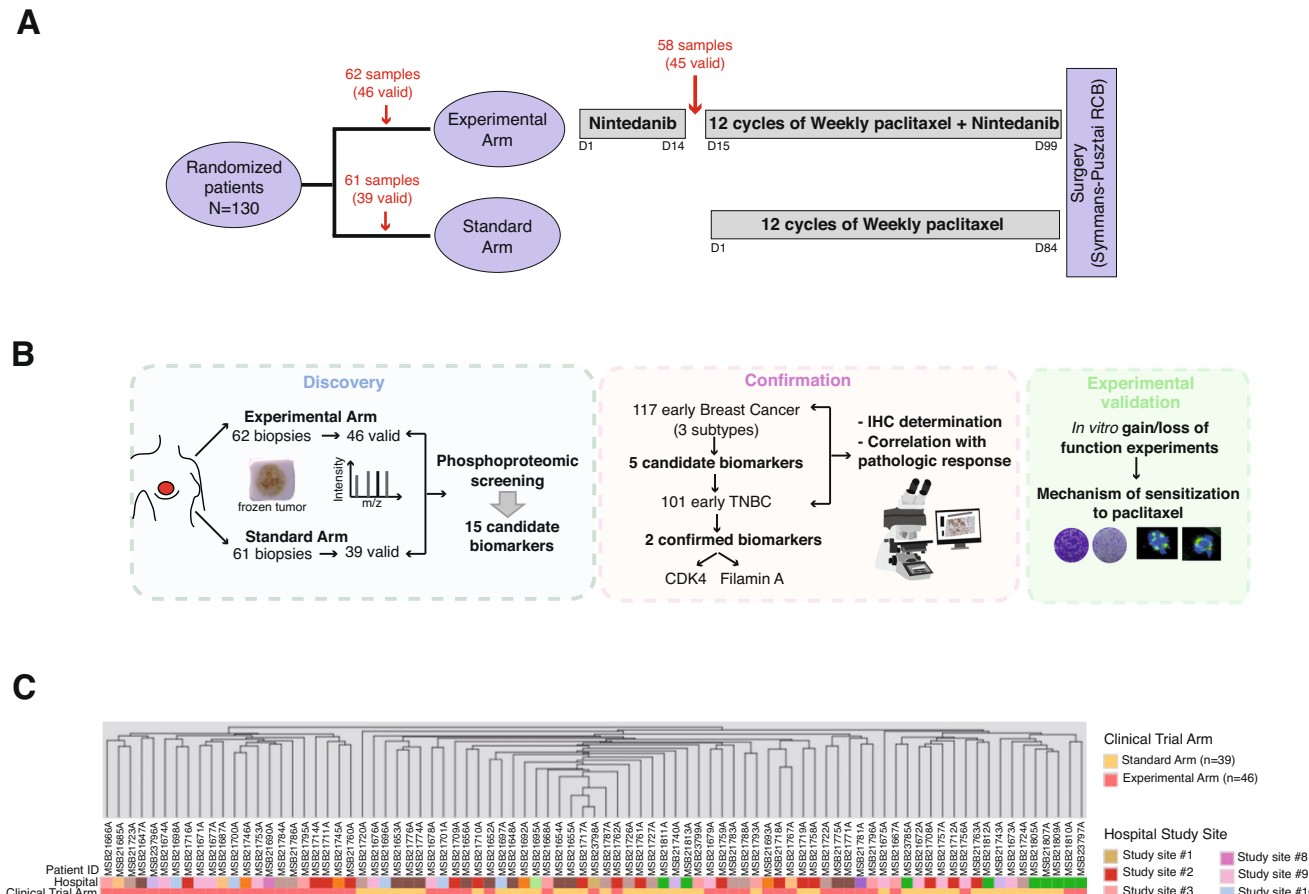

**Fig. 1 | Clinical trial tumour samples: the phosphoprofiles were independent of the study site and/or treatment arm. A** Clinical trial treatment and sampling schedule. After randomization, patients were scheduled for a fresh tumour biopsy. Sixty-two and sixty-one patients allocated to the Experimental and Standard arms, respectively, consented to and underwent a baseline biopsy (out of 130 patients). Patients allocated to the experimental arm underwent a 2-week course of single-agent nintedanib (150 mg orally twice a day), and then a second tumour sample was harvested (N = 58 patients consented to this second biopsy) prior to undergoing 12 weekly courses of paclitaxel combined with nintedanib. Those allocated to the standard arm immediately started weekly paclitaxel without the 14-day delay and did not have a second tumour sample harvested. The endpoint (tumour response according to the RCB score) was determined at the time of surgery, and patients then received standard treatment according to the referring physician's choice (radiation or hormonal therapy or further chemotherapy if indicated). **B** Flow chart depicting the study steps: biomarker discovery, biomarker confirmation and experimental validation. **C** Unsupervised hierarchical clustering. A phosphopeptide intensity data matrix was used for clustering analysis. Patient IDs are listed horizontally. The two following rows indicate, for each sample, whether they were allocated to the standard or experimental arm and the study site origin.

---

enriched in responder samples (Fig. 2B). Finally, when responder and nonresponder samples from the paclitaxel plus nintedanib arm were compared, we did not find any significantly enriched kinase (2594/1212 phosphopeptides/unique proteins). The fact that the KSEAs lost significance and displayed reduced enrichment scores as we increased the proportion of samples from patients who received combination therapy relative to those who received monotherapy in the analysis suggests a certain specificity of the enrichment of the activity of P70S6K, CDK4, PKC, AMPK1/2 and CaMK-IV in responders to paclitaxel monotherapy (i.e., the enrichment score was lower when samples from both trial arms were mixed than when only samples from the monotherapy arm were evaluated; furthermore, when we included only samples from the combination arm, no enrichment was observed). Of note, in addition to the indirect detection of P70S6K, CDK4, PKC, AMPK1/2 and CaMK-IV activity enrichment by KSEA, more phosphorylated peptides mapping to the regulatory regions of these kinases were detected in samples from responders to paclitaxel than in those from nonresponders (Supplementary Fig. 2). Since the detected phosphorylation sites of these kinases are known to be involved in

kinase activation, this finding further supports the involvement of those kinases in the responder phenotype.

Regarding potential resistance biomarkers (i.e., kinases enriched in samples from nonresponders compared to samples from responders), the data are shown in Supplementary Fig. 3. Little overlap was observed when the analysis was run comparing responders and nonresponders in the whole trial (enrichment of SRC, JAK2, CDK5 and CK1; Supplementary Fig. 3A), in the experimental arm only (B-adrenergic receptor kinase, PIM1 and PDK1; Supplementary Fig. 3B), or in the standard arm only (RAF1, CK1 and B-adrenergic receptor kinase; Supplementary Fig. 3C).

Key information about potential predictive markers can also be obtained by examining the significantly up-/downregulated phosphoproteins in responders or nonresponders on top of the kinases predicted by KSEA, since phosphorylated proteins can mediate the observed effect from a mechanistic point of view. Figure 3 shows the volcano plots of the phosphopeptides enriched in responders or nonresponders. Figure 3A displays the differentially regulated peptides in responders and nonresponders in the paclitaxel-only arm

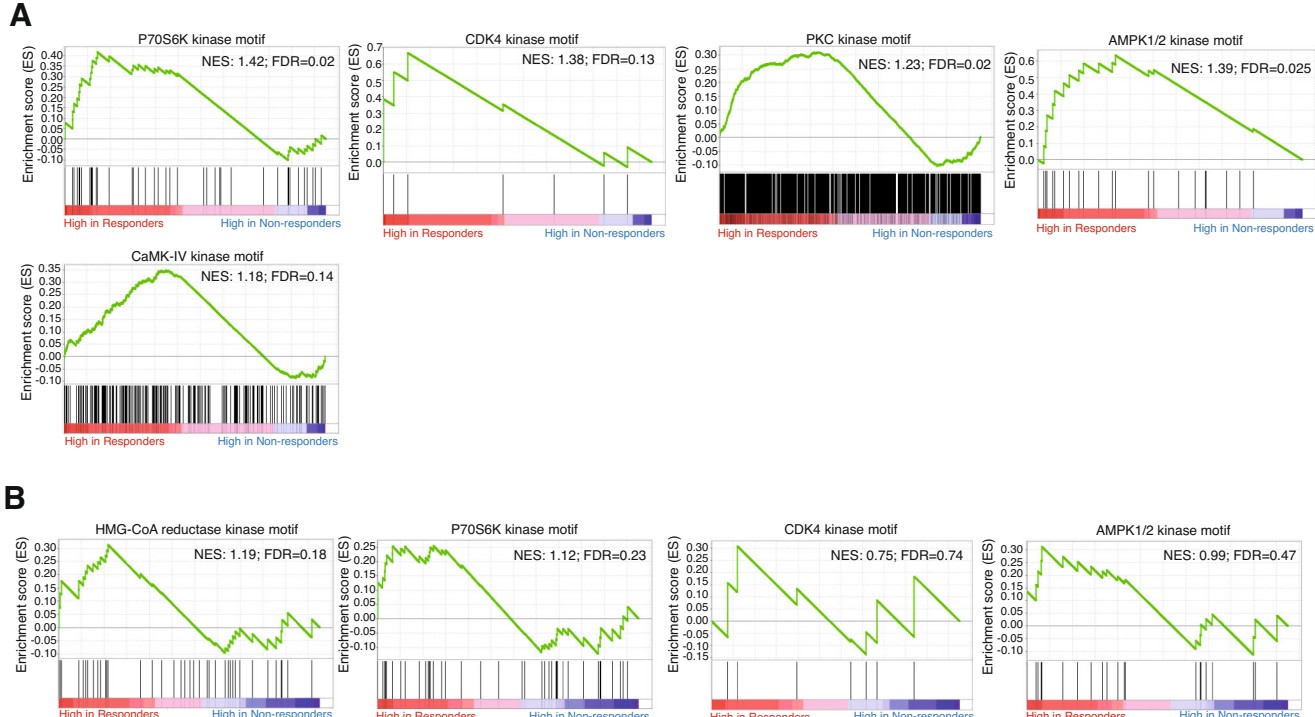

**Fig. 2 | Kinases driving the phosphoprofiles of responders to paclitaxel.** Panel **A** displays the significantly enriched kinases found in the baseline samples for patients who achieved a pCR in the paclitaxel monotherapy arm. However, Panel **B** shows that when the analysis was repeated combining the samples from the paclitaxel-only arm with those from the combination arm, virtually no significant enrichment was observed. For all KSEA plots, each vertical black line represents a phosphopeptide that can be phosphorylated by the depicted kinase that was detected in the samples from one of the compared conditions in the KSEA. High in responders and high in nonresponders refers to the increased abundance of phosphopeptides in the baseline samples of patients who achieved a pCR or patients who did not achieve a pCR, respectively. NES (normalized enrichment score) and FDR (false-discovery rate) values are depicted for each KSEA. A relaxed FDR boundary (up to 0.20) was allowed to ensure as little information loss as possible in the mass spectrometry-to-immunohistochemistry translation step, since the biomarker candidates underwent a subsequent 2-patient series filter.

(Arm B), whereas Figs. 3B, C show the regulated peptides combining both treatment arms and Arm A only, respectively. In all volcano plots, the central dot cloud represents those peptides that were present both in responders and nonresponders and experienced some degree of regulation in one or another group of patients. The two additional lateral dot clouds represent phosphorylation sites that were undetectable in one group (responders or nonresponders) but were present in the other, suggesting that dramatic regulatory events were involved in the response or lack thereof. Phosphopeptide IDs with >$2^4$ (16-fold) regulation, together with their *P* value and FDR, are listed in Supplementary Data 1. Many peptide IDs map to proteins with little or uncharacterized functional significance; however, a considerable number of peptides upregulated in responders to paclitaxel mapped to proteins implicated in cytoskeletal polymerization and rearrangement, such as vimentin, laminin, plectin, tensin, filamin, and Rab7 (Group 1 and Group 2 phosphopeptide cloud in Fig. 3A–C; Supplementary Data 1). As we show below, this fact is of key relevance, since paclitaxel exerts its antitumour effects by stabilizing and thereby blocking microtubule polymerization/depolymerization dynamics[26–28]. No significantly regulated peptides were observed in tumours resistant to paclitaxel (no dots above the significance level in the far-left clouds are observed in Fig. 3A–C). For subsequent experiments, only those phosphopeptides with FDR < 0.25 were studied.

**Elevated CDK4 and filamin A levels narrow down a subset of patients with increased sensitivity to paclitaxel**
Mass spectrometry is not yet an over-the-counter technology in cancer hospitals. In addition, the phosphoscreening data (KSEAs or volcano-plot hits) are the result of a training set with relaxed FDR values to

enhance the number of biomarker candidates, a strategy that was proven successful previously[10]. Because of these two reasons and aiming for the clinical applicability of our results, we sought to determine these potential hits by relying on immunohistochemistry (IHC) in additional patient series, a widely available technique in pathology diagnostic departments in hospitals for detecting protein and protein phosphorylation levels. IHC (as opposed to mass spectrometry) is optimized for formalin-fixed, paraffin-embedded samples, which in turn are the most common tissue vehicle in clinical routine.

To translate the phosphoscreening results (KSEAs and volcano plots of the proteins listed in Supplementary Data 1) to measurable data by immunohistochemistry, we followed our previously described mass spectrometry-to-immunohistochemistry approach[10]. Following the algorithm depicted in Supplementary Fig. 4, the approach yielded 11 potential antibodies for biomarkers of sensitivity to paclitaxel: p-P70S6K (Thr389), CDK4, filamin-A, HMG-CoA reductase, p-vimentin (Ser56), p-AMPK1/2 (Thr172), p-Pan-PKC (Thr497), p-CaMKIV (Thr196/200), p-filamin A (Ser2512), p-YAP1 (Ser127) and plectin. Antibody setup and control stainings are shown in Supplementary Fig. 5A–K.

To account for the relaxed FDR boundaries of the training set, we applied a biomarker selection filter consisting of a two-step process in independent sample sets. The candidate biomarkers were first tested in a patient set of 117 high-risk early breast cancer patients of the three subtypes who were treated with paclitaxel-based neoadjuvant chemotherapy (Set 1; clinical and pathological characteristics shown in Supplementary Table 3). In this first step, we aimed to detect which of the 11 potential biomarkers maintained an association with the response to paclitaxel after the mass-spectrometry-to-immunohistochemistry translation step, in a population of mixed breast cancer subtypes.

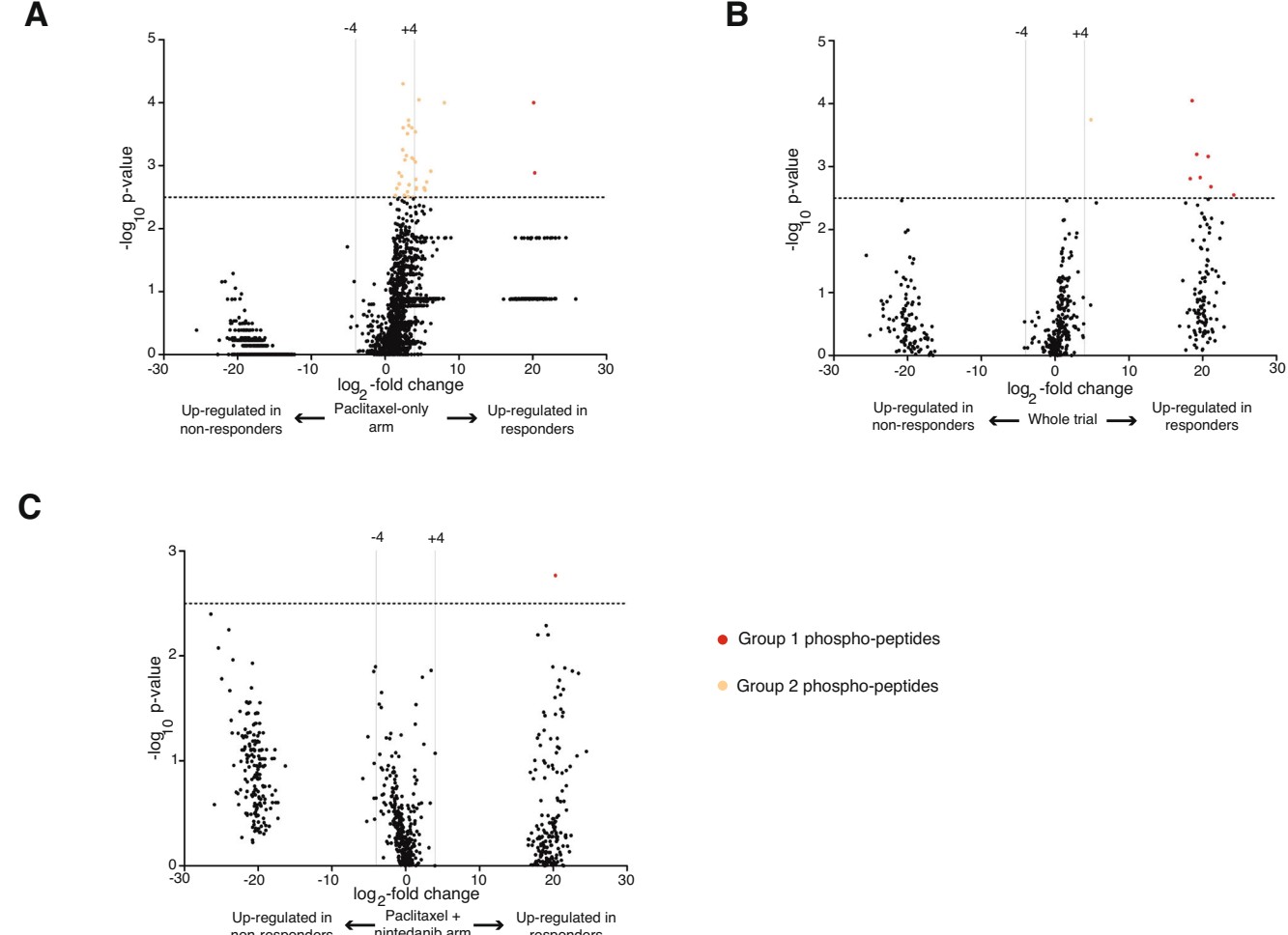

**Fig. 3 | Volcano plots with regulated phosphopeptides among responders and nonresponders to paclitaxel.** Regulated phosphopeptides in patients who achieved a pCR versus those who did not in the standard arm (**A**), whole trial (**B**) and the experimental arm (**C**). Phosphopeptides with greater than $2^4$ (16-fold) up- or downregulation in one or another condition with Mann–Whitney Wilcoxon $P < 10^{-2.5}$ are highlighted and colour-coded for each comparison; their IDs and FDRs are listed by colour group and comparison in Supplementary Data 1.

We next filtered those that held significance through a second patient set of exclusively TNBC patients ($N = 101$; Set 2: clinical and pathologic characteristics shown in Supplementary Table 4) undergoing neoadjuvant paclitaxel-based treatment. By doing so, we expected to have available a small set of robust biomarkers for the breast cancer subtype where paclitaxel chemotherapy is more relevant.

The 117 samples from Set 1 were stained with the 11 antibodies against the potential biomarkers, and an H-score was calculated. The H-score was divided into quartiles, and the association with response was studied by determining the probability of obtaining a pCR (RCB = 0) for the patients in the top quartile versus the remaining patients. A staining example of two patients from the upper and lower quartiles of filamin A is shown in Fig. 4A. The upper quartile cut-off value for each staining is depicted in Supplementary Table 5.

We found that the following biomarker candidates were associated with a pCR to paclitaxel-based neoadjuvant therapy in Set 1: p-P70S6K (Thr[389]) (2.89-fold higher chance or achieving a pCR for patients in the upper H-score quartile versus patients with H-score in quartiles 2 to 4; $P = 0.037$), CDK4 (2.85-fold; $P = 0.048$), filamin A (3.28-fold; $P = 0.062$), HMG-CoA reductase (4.00-fold; $P = 0.064$) and p-Vim (Ser[56]) (2.93-fold; $P = 0.047$). The pCR rate in the whole Set 1 cohort, or divided by breast cancer subtype (luminal, HER2 or TNBC) according to the value of p-P70S6K (Thr[389]), CDK4, filamin A, HMG-CoA reductase and p-Vim (Ser[56]), is shown in Fig. 4B. Conversely, despite showing a potential association in the KSEA or volcano plot analysis, when translated to immunohistochemical staining, the following biomarker candidates did not show an association with a pCR in Set 1: p-AMPK (Thr[172]) (2.07-fold higher chance or achieving a pCR for patients in the upper H-score quartile versus patients with H-score in quartiles 2 to 4; $P = 0.25$), p-Pan-PKC (Thr[497]) (0.4-fold, $P = 0.12$), p-CaMK-IV (Thr[196/200]) (1.31-fold; $P = 0.72$), p-filamin A (Ser[2152]) (1.43-fold, $P = 0.55$), p-YAP1 (Ser[127]) (1.26-fold; $P = 0.62$), and plectin (0.88-fold; $P = 0.193$).

We next proceeded to the final biomarker filtering in Set 2. The upper quartile cut-off value for each staining in Set 2 is shown in Supplementary Table 6. Patients in the upper quartile of CDK4 displayed a 2.71-fold ($P = 0.04$) higher chance of achieving a pCR in response to neoadjuvant chemotherapy than patients in Q2-Q4. Similarly, patients with upper quartile filamin A staining exhibited a 3.36-fold greater probability of achieving a pCR ($P = 0.039$). Conversely, no statistically significant associations were confirmed for p-Vim (Ser[56]) (1.67-fold higher chance of achieving a pCR for patients in the upper H-score quartile versus patients in quartiles 2 to 4, $P = 0.31$), p-P70S6K (Thr[389]) (1.1-fold; $P = 0.90$), or HMG-CoA reductase (0.4-fold, $P = 0.11$). The percentages of patients achieving a pCR according to their staining levels of CDK4, filamin A, p-Vim (Ser[56]) and p-P70S6K (Thr[389]) are shown in Fig. 4C. Ninety percent of patients with both CDK4 and filamin A staining levels in the upper quartile achieved a pCR (Fig. 4D). Although this study is retrospective, its results

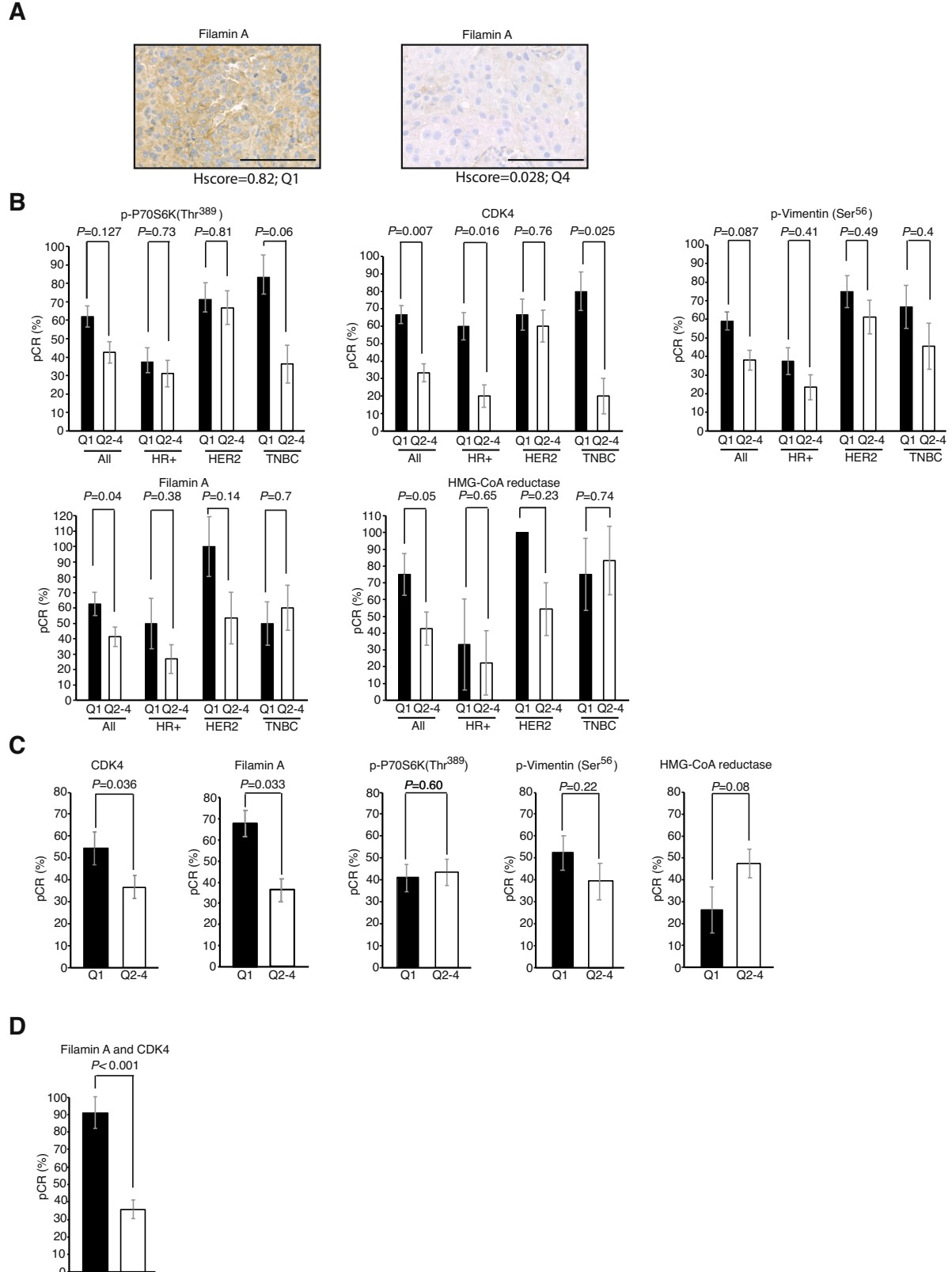

**Fig. 4 | Independent testing of biomarkers of response to paclitaxel.**
**A** Immunohistochemical staining of filamin A from patients with H-scores in the upper (Q1, left) and lower (Q4, right) quartiles. Scale bars: 100 μm. **B** Comparison of pCR rates for patients in the Q1 versus Q2-4 subgroups among the whole Set-1 series ($N = 117$) or divided by subtype (HR+: Hormone-positive, $N = 62$; HER2: HER2-amplified, $N = 35$; TNBC: triple-negative breast cancer, $N = 20$), according to the staining of p-P70S6K (Thr[389]), CDK4, p-Vimentin (Ser[56]), filamin A or HMGCOA reductase. **C** Same as in (**B**) but for Set-2 patients ($N = 101$). **D** pCR rates of patients from Set-2 with combined elevated levels of CDK4 and filamin A compared to patients with low CDK4 and filamin A levels. Proportions were compared with a two-sided Xi-square test. $P$ values are adjusted (Bonferroni). Error bars: standard error of the proportion. Source data are provided as a Source data file.

compare favourably with commonly reported pCR rates in unselected TNBCs achieved with polychemotherapy regimens alone (40–50%)[14,29] or in combination with immunotherapy (55–65%)[30–32]. Finally, it is worth mentioning that patients did not show significant staining heterogeneity regarding tumour cells/stromal or nuclear/cytoplasmic CDK4 or filamin A staining (Supplementary Fig. 6).

### CDK4 and filamin A lead to increased microtubule stability and sensitivity to paclitaxel through CLIP-170

The notion that tumours with high replicative fractions are more sensitive to classic cytotoxics is commonly assumed in clinical oncology. However, a specific mechanistic explanation for each cytotoxic agent is lacking. Thus, we sought to understand how high levels of CDK4 and filamin A specifically sensitize tumour cells to paclitaxel. CDK4 is a well-known regulator of the mitotic process, as it phosphorylates and inactivates RB1, which in turn liberates the E2F transcription factor during the G1/S transition[33]. Filamin A is a scaffolding protein that crosslinks actin filaments into networks and participates in anchoring membrane proteins to the actin cytoskeleton and in transducing signals from the tumour microenvironment[34]. How these functional features sensitize cells to paclitaxel treatment is currently unknown.

We generated stable variants of the TNBC cell line MDA-MB-231 that constitutively expressed elevated levels of CDK4 (MDA-MB-231 CDK4; Fig. 5A). Compared to the parental MDA-MB-231 cells, sensitivity to paclitaxel was increased 2.7-fold ($P = 0.0012$), as evidenced by the colony assays and IC50 calculations shown in Fig. 5A, in line with the data observed in patients. A previous study in the field of the regulation of cell motility and invasion described a physical interaction between cyclin D1 and filamin A; in addition, a positive correlation between CDK4 and filamin A (Ser$^{2152}$) phosphorylation levels was found, but direct phosphorylation was not demonstrated[35]. We were unable to show the coimmunoprecipitation of CDK4 and filamin A in either parental MDA-MB-231 or MDA-MB-231 CDK4 cells (Supplementary Fig. 7A). Confocal imaging did not demonstrate the colocalization of CDK4 and filamin A in either cell line (Supplementary Fig. 7B). In addition, an in vitro kinase assay did not show kinase activity of CDK4 over filamin A (Supplementary Fig. 7C). The levels of CDK4 and filamin A, however, demonstrated a positive and statistically significant correlation in patients from Set 2 (Supplementary Fig. 8). Similarly, MDA-MB-231 CDK4 cells showed considerably higher levels of filamin A than parental MDA-MB-231 cells (Fig. 5B). Real-time PCR revealed a transcriptional mechanism of filamin A upregulation in MDA-MB-231 CDK4 cells (Fig. 5C). This observation is consistent with the fact that while active, RB1 sequesters and represses the E2F1 transcription factor, which is liberated when CDK4 phosphorylates and inactivates RB1. E2F1 has ~3000 transcriptional targets, among which is filamin A, according to the last update of the ENCODE project (https://maayanlab.cloud/Harmonizome/gene_set/E2F1/ENCODE+Transcription+Factor+Targets)[36]. We also generated stable MDA-MB-231 cells with elevated stable expression of filamin A (MDA-MB-231 FLNA, Fig. 5D). The increased expression of filamin A did not change CDK4 levels (Fig. 5D). These cells showed a similar sensitization to that observed for MDA-MB-231 CDK4 (approximately 3-fold to 0.32 nM; $P = 0.0004$; Fig. 5D). Together with the fact that transient filamin A knockdown in MDA-MB-231 CDK4 cells restored paclitaxel sensitivity to levels similar to those observed in MDA-MB-231 WT cells (IC50 = 0.92 nM; the comparison with the IC50 displayed by MDA-MB-231 CDK cells was statistically significant; $P < 0.0001$; Fig. 5E), the transcriptional link between CDK4 and filamin A suggests that filamin A is a mediator of the increased sensitivity to paclitaxel in MDA-MB-231 CDK4 cells. High proliferation rates have been related to nonspecific sensitization to cytotoxic agents; interestingly, neither CDK4 nor filamin A overexpression sensitized tumour cells to the other cytotoxics received in the neoadjuvant TNBC setting (anthracyclines or platins;

Supplementary Fig. 9A, B). Together with the limited correlation observed among CDK4, filamin A and the Ki67 replicative fraction in the TNBC patient series (Supplementary Fig. 9C), these data suggest that CDK4 and filamin A are specifically involved in sensitization to paclitaxel only.

Paclitaxel exerts its cytotoxic effects by binding to beta-tubulin subunits of assembled microtubules, stabilizing them and interfering with the polymerization/depolymerization equilibrium that allows their extension and dynamics[26,37]. Microtubules in the mitotic spindle have rapid dynamics, and when these dynamics are interrupted, mitosis results in chromosomal mis-segregation, mitotic arrest or catastrophe and the formation of micronuclei, often leading to cell apoptosis[27]. Although the link between CDK4 and elevated filamin A levels seemed clear and increased filamin A levels seemed to be a necessary link between CDK4 and increased sensitivity, the obvious remaining question was how to relate elevated filamin A levels with altered tubulin dynamics that could explain the increased sensitivity to paclitaxel. Filamin A and tubulin are reported to physically interact in response to forces in the process of the transduction of mechano-transcriptional signals by the microtubule network[38]. We analysed the spatial distribution of filamin A and tubulin and found a colocalization of these two proteins both in the cytoplasm and submembrane compartments (Fig. 6A). Accordingly, we hypothesized that changes in filamin A levels could lead to quantitative or qualitative changes in the protein complexes normally formed with tubulin or filamin A. In an attempt to identify regulated proteins within the filamin-tubulin complexes that could be implicated in microtubule dynamics, we performed a pull-down analysis of tubulin and filamin A, quantifying and comparing the complexes between MDA-MB-231 WT and MDA-MB-231 CDK4 cells. The plots shown in Fig. 6B display the ratios of proteins bound to anti-filamin A antibody or isotype control in MDA-MB-231 WT (left panel) or MDA-MB-231 CDK4 cells (right panel). A migration in either the ratio or the intensity indicates potential regulation. Figure 6C shows the results of the tubulin pull-down comparison between MDA-MB-231 WT (left) and MDA-MB-231 CDK4 cells (right). The protein IDs bound to filamin A or tubulin (and their spectral intensity) in the two cell lines are gathered in Supplementary Data 2 and 3. Cytoplasmic linker protein 170, or CLIP-170, is a protein that belongs to the groups of plus-end tracking proteins, or +TIPs. Microtubules are nucleated out of the microtubule organizing centre (MTOC), where their minus end is anchored; their plus ends grow out of the MTOC[39]. The plus ends are highly dynamic, and +TIPS are a class of microtubule-binding proteins that accumulate at and track with growing tubules, regulating their dynamics and interactions[40], of which CLIP-170 was the first discovered member[41]. It has been shown that enhanced CLIP-170 accumulation leads to reduced and slower tubulin polymerization[42]. In addition, others have found that CLIP-170 promotes paclitaxel binding to microtubules[43]. Quantitative mass spectrometry showed that the amount of CLIP-170 bound to tubulin and the amount of CLIP-170 bound to filamin A increased by >7-fold and >1000-fold, respectively, when CDK4 was overexpressed (Supplementary Data 2 and 3).

Whether CIP-170 and filamin A have a physical interaction or not is currently unknown. Thus, despite our mass spectrometry data, we sought to confirm this interaction through coimmunoprecipitation. Figure 6D proves that filamin A can bind CLIP-170. Taken together, these data led us to hypothesize that in the presence of increased filamin A, tubulin-filamin A complexes would be enriched in CLIP-170, which would lead to increased paclitaxel sensitivity through a two-pronged mechanism: increased paclitaxel accumulation and increased microtubule stabilization (Fig. 6E).

To confirm this hypothesis, we experimentally tested the two potential mechanisms. Figure 6F shows the amount of paclitaxel bound to microtubules in MDA-MB-231 WT, CDK4 and FLNA variants. It can be appreciated that the CDK4 and FLNA cell lines accumulated

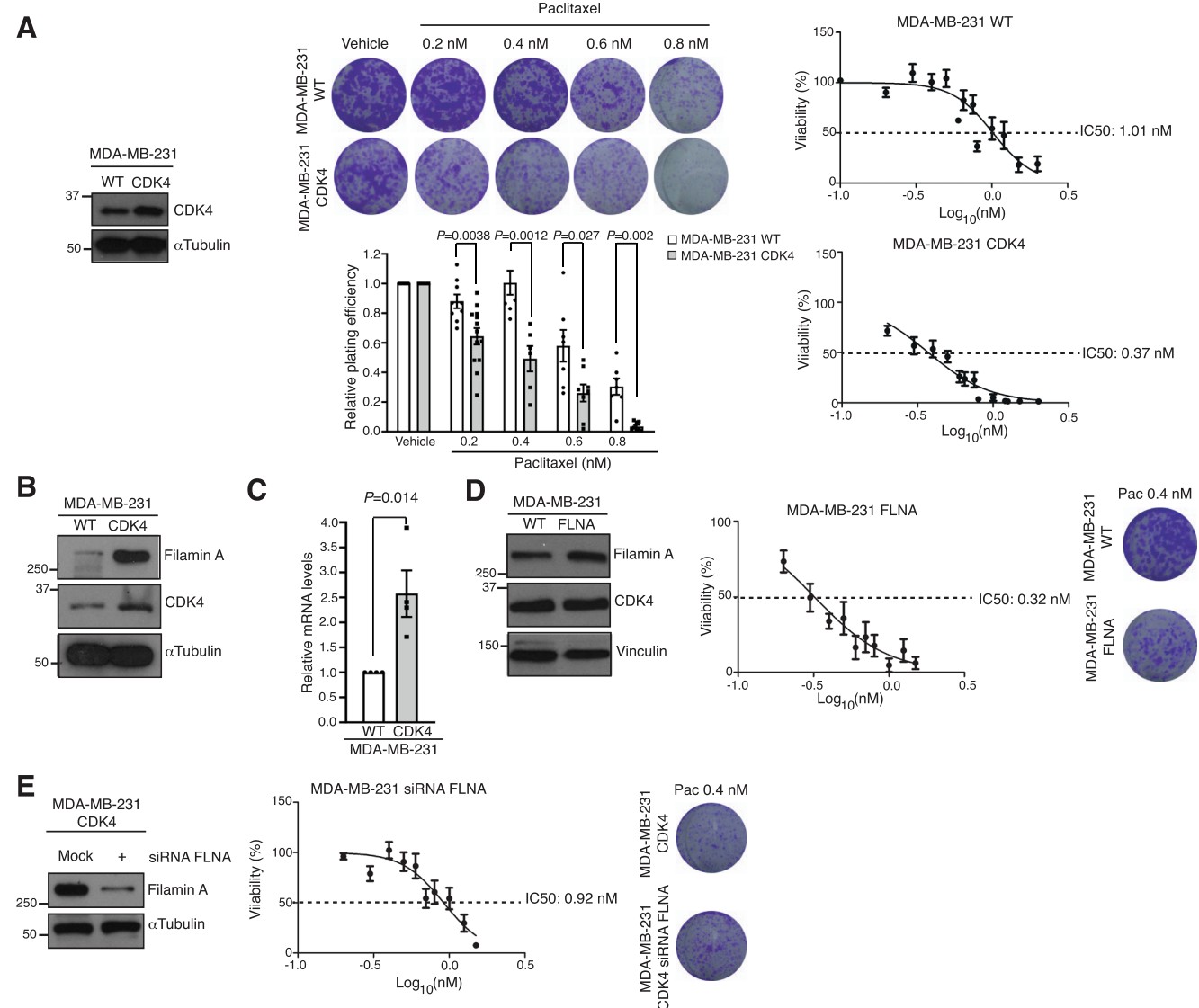

**Fig. 5 | CDK4- and filamin A-driven increased sensitivity to paclitaxel in TNBC models. A** Left: Immunoblot showing different CDK4 levels between parental MDA-MB-231 and MDA-MB-231 CDK4 cells. Tubulin is shown for housekeeping purposes. Experiment was repeated 3 times with similar results. Middle: colony assay pictures (upper panel) and quantitation or relative plating efficiency chart (lower panel) of parental MDA-MB-231 and MDA-MB-231 CDK4 cells exposed to various concentrations of paclitaxel. In the relative plating efficiency chart, each condition is normalized to the untreated condition for each MDA-MB-231 variant. Data are presented as the mean ± SEM, $n \geq 6$ independent experiments; two-sided unpaired t-test. Right: IC50 plots showing the ~2.7-fold variation in paclitaxel sensitivity. Data are presented as the mean ± SEM, $n \geq 3$ independent experiments; **B** Immunoblot showing the upregulation of filamin A in MDA-MB-231 CDK4 cells versus the parental cells; tubulin is shown for housekeeping purposes. Samples derived from the same experiment and blots were processed in parallel. Experiment was repeated 3 times with similar results. **C** Real-time PCR showed a 2.5-fold increase in filamin A

mRNA levels in MDA-MB-231 CDK4 cells. Data are presented as the mean ± SEM, $n = 4$ independent experiments; two-sided unpaired t-test. **D** Left: Immunoblot showing filamin A levels in MDA-MB-231 FLNA and MDA-MB-231 WT cells. No changes were observed in CDK4 levels in the former. Vinculin: housekeeping. Samples derived from the same experiment and blots were processed in parallel. Experiment was repeated 3 times with similar results. Right: IC50 charts showing the sensitization to paclitaxel in MDA-MB-231 FLNA cells compared to the parental cells (Mean±SEM, n≥3 independent experiments) together with examples of colony assays. **E** Left: Transient filamin A knockdown with siRNA in MDA-MB-231 CDK4 cells. Experiment was repeated 3 times with similar results. Right: phenotype recovery: the increased sensitivity to paclitaxel in MDA-MB-231 CDK4 cells is restored back to normal (i.e., similar IC50 to that of MDA-MB-231 WT) when filamin A is downregulated. Data are represented as mean ± SEM, $n \geq 14$ independent experiments. Examples of colony assays are shown. Source data are provided as a Source data file.

more paclitaxel and did so earlier than their WT counterparts. In addition, when filamin A was knocked down in MDA-MB-231 CDK4 cells, the phenotype was reversed: these cell lines accumulated the same amount of paclitaxel and at a similar rate as that observed in WT cells.

Then, we studied the effects on microtubule stability. The status of microtubule dynamics can be assessed by determining several post-translational modifications. Alpha-tubulin lysine acetylation is a key modification that stabilizes microtubules[44]. We measured the levels of

Lys-40 tubulin acetylation in MDA-MB-231, MDA-MB-231 CDK4 and MDA-MB-231 FLNA cell lines in response to vehicle or paclitaxel (Fig. 7A). Both the baseline and paclitaxel-treated levels of acetylated tubulin were increased in the two cell variants, indicating increased microtubule stability as a result of elevated CDK4 and/or filamin A. When filamin A was knocked down in MDA-MB-231 CDK4 cells, we observed a decrease in tubulin acetylation levels (Fig. 7B). Interestingly, experimental replicates in which siRNA against filamin A knocked down filamin A with variable efficiency in MDA-MB-231 CDK4

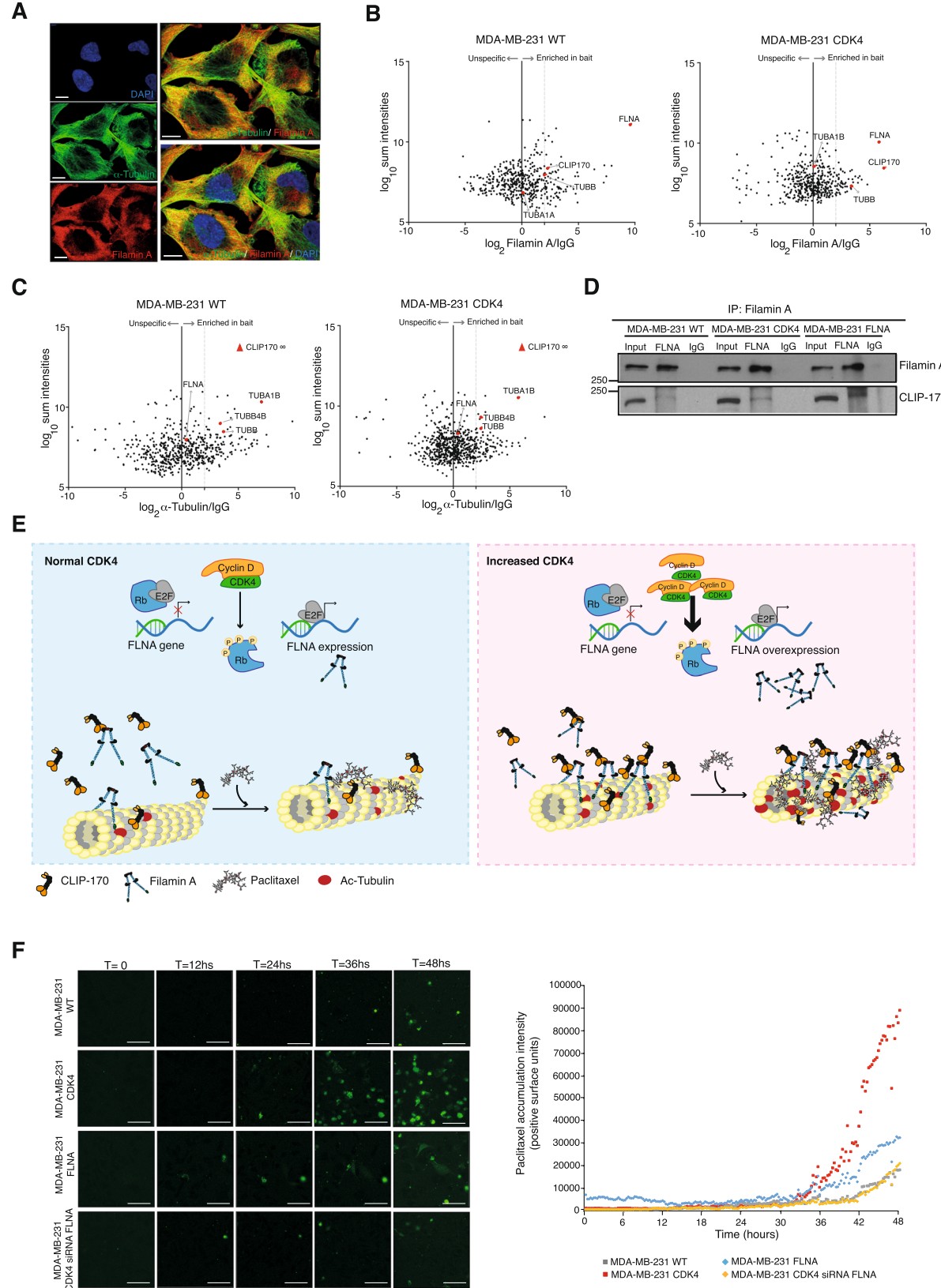

cells showed that tubulin acetylation levels were modified accordingly with filamin A protein levels (Supplementary Fig. 10).

Increased binding of paclitaxel to microtubules combined with a context prone to increased microtubule stabilization should enhance the consequences of tubulin polymerization/depolymerization equilibrium arrest in cell replication, explaining the observed increased sensitivity to paclitaxel. Figure 7C and D shows the consequences of paclitaxel exposure in parental MDA-MB-231 cells and the CDK4 and FLNA variants. The percentage of normal cell divisions after exposure to paclitaxel decreased >4-fold (9.86% and 9.08% for the MDA-MB-231 CDK4 and FLNA variants vs. 41.5% in the parental cell line; $P = 0.03$), whereas the percentage of multipolar metaphases and other

**Fig. 6 | Tubulin and filamin A form a complex with CLIP170, which elicits increased binding of paclitaxel to microtubules. A** Spatial colocalization of alpha-tubulin and filamin A in MDA-MB-231 cells (Pearson's colocalization coefficient = 0.5, $n = 76$ cells). Scale bars: 10 μm. **B** Filamin A pull-down in MDA-MB-231 WT or CDK4 cell lines. For each isolated protein, the X axis represents the $Log_2$ of the average ratio of the protein isolated in the anti-filamin A pull-down and the protein isolated in the IgG control antibody. The Y axis, conversely, represents the $Log_{10}$ of the sum of the average intensities found bound to the anti-filamin-A antibody and the IgG isotype control. **C** Same as in (**B**) for tubulin pull-downs. **D** Coimmunoprecipitation of CLIP-170 and filamin A. Filamin A was immunoprecipitated in whole-cell lysates from the three cell lines: MDA-MB-231 WT, CDK4 and FLNA. Three samples are shown for each cell line: total lysate, immunoprecipitated with anti-filamin A, and immunoprecipitated with isotype IgG control. Samples derive from the same experiment and blots were processed in parallel. Experiment was repeated 3 times with similar results. **E** Schematic representing the proposed mechanism. On the left-hand side, low CDK4 levels lead to average filamin A expression, which does not enhance the binding of CLIP-170 to microtubules. On the right-hand side, tumour cells with increased CDK4 levels would lead to the overexpression of filamin A. Filamin A would recruit an excess of CLIP-170 to tubulin, which ultimately leads to increased binding of paclitaxel and microtubule acetylation and hyperstabilization. **F** Paclitaxel-binding experiment. Fluorescently labelled paclitaxel was added to live cultures of MDA-MB-231 WT, CDK4 or FLNA cells. MDA-MB-231 CDK4 cells with filamin A knockdown were added to the experiment as well. The greater the green signal is, the higher the amount of paclitaxel bound to microtubules. It can be appreciated how both CDK4- and filamin A-overexpressing cell lines display both earlier and higher paclitaxel binding. Scale bar: 75 μm. The chart on the right-hand side depicts the signal (in fluorescent surface units) tracing paclitaxel accumulation over the 48-h time course, displaying a clear increase in the two overexpressing transfectants (CDK4 and FLNA) compared to the parental cell line and a reversion of the phenotype by filamin A knockdown in MDA-MB-231 CDK4 cells. Each dot represents mean signal intensity of six independent fields. Source data are provided as a Source data file.

aberrations increased in the MDA-MB-231 CDK4 and FLNA variants (Fig. 7C; images are split by fluorescence channel in Supplementary Fig. 11A; the percentages of each aberration are shown in Supplementary Data 4). As a consequence of these aberrations in the cell division process, the percentage of abnormal cell nuclei also increased (88.7% and 82.5% in the CDK4 and FLNA variants vs. 71.6% in the parental MDA-MB-231 cell line; $P = 0.015$; Fig. 7D; split by channel in Supplementary Fig. 11B; Supplementary Data 4).

Taken together, our data suggest that CDK4 increases filamin A levels through a transcriptional mechanism. Increased filamin A, in turn, binds CLIP-170, increasing the amount of CLIP-170 in the tubulin-filamin complexes. This accumulation of CLIP-170 leads to increased microtubule acetylation and stabilization and increased paclitaxel binding to microtubules. These effects, combined, lead to mitotic catastrophe, explaining the increased sensitivity to this drug in tumours with elevated CDK4.

## Discussion

Precision oncology is highly based on gene-centric approaches. Although this perspective has led to considerable advances that have positively impacted clinical care, it falls short of answering certain translational research questions. One scenario of particular difficulty is that in which tumours are not driven by oncogene addiction mutations. In most tumours, each mutation confers only a small fitness advantage, but several of those low-penetrant mutations might cooperate and promote tumour progression[45,46]. However, most of those mutations are nonrecurring (i.e., limited to one or a few patients, making the number of potential combinations of mutations potentially immense). In addition, the functional impact of each of those mutations still requires functional characterization. Together, these facts make it very complicated to issue accurate individual predictions in most cases. The most common solution for this situation relies on gene-expression panels, which have been successful in difficult tasks such as predicting benefit or not from hormonal treatment alone in non-HER2-positive breast cancers by grouping cancers on the basis of the expression of a number of genes[47,48]. However, the functional characterization of the majority of these genes or how they contribute functionally to the final tumour phenotype, depending on whether they are mutated or not, is not always available. Regardless of how accurate these approaches can be, they still lack functional specificity, hampering subsequent developments such as the rational design of therapies for adverse prognosis subgroups.

In the past, we have solved similarly complex questions in translational oncology by means of mass spectrometry-aided phosphoproteomics. The justification for relying on this technique is that different phenotypes (i.e., drug response or resistance) could be achieved by multiple different genomic landscapes, which ultimately collapse into the functional hyper-/hypoactivation of a discrete number of signalling axes, which in turn are the effectors of the tumour genotypes. This approach, for example, allowed us to solve a practical kinase-based taxonomy of TNBC[10] and to solve the problem of acquired resistance to antiangiogenics[49].

This time, we aimed to understand what drives sensitivity to the most commonly used cytotoxic in breast cancer: paclitaxel. We took advantage of the samples obtained in a clinical trial that compared paclitaxel monotherapy in treatment-naive early breast cancer patients with paclitaxel plus the multikinase inhibitor nintedanib[21]. Our study has several features worth highlighting.

Our study provides a specific predictive factor of the efficacy of paclitaxel. Similar to our previous study[10], from a relatively high number of candidate markers yielded from the mass spectrometry data (Supplementary Fig. 4; $N = 11$), only a limited number of them were successfully confirmed by immunohistochemistry in independent patient sets (Fig. 4). Those markers that showed association in Set 1 were further filtered in a second set (Set 2) of a homogenous population of TNBC patients, leaving a final number of 2 markers: CDK4 and filamin A (Fig. 4B, C). Filamin A and CDK4 were highly accurate in predicting a pCR, with patients with upper-quartile staining of both CDK4 and filamin A achieving a 90% pCR in response to paclitaxel-based chemotherapy (Fig. 4D), which compares favourably with expected ratios in unselected patients receiving paclitaxel-based combinations[14,29]. These markers could simplify complex treatment schedules now including up to 5 drugs, reserving immunotherapy and other combinations for patients with low levels of CDK4 and filamin A.

Interestingly, according to our data, elevated CDK4 and filamin A levels appear to be specifically linked to the mechanism of action of paclitaxel, adding robustness to their biomarker role. Filamin A upregulation by CDK4 (Fig. 5B, C) is followed by increased binding of CLIP-170 to filamin A and tubulin, mediating an increased binding of paclitaxel to microtubules (Fig. 6). This results in acetylated microtubules (Fig. 7A, B), mitotic arrest and mitotic catastrophe (Fig. 7C, D). In the absence of filamin A, paclitaxel binding to microtubules decreases (Fig. 6F) along with tubulin acetylation (Fig. 7B), reverting the increased sensitivity to paclitaxel (Fig. 5D), which suggests a specific link between CDK4 and filamin A in regulating sensitivity to this drug. The experiments shown in Supplementary Fig. 9 further confirm drug specificity: although high replicative fractions have been non-specifically associated with sensitization to cytotoxics, the role of CDK4 and filamin A seems specific for paclitaxel, since (1) these biomarkers were discovered in a series of patients who received single-agent paclitaxel (Figs. 1–3); (2) CDK4 and filamin A did not sensitize tumour cells to other cytotoxics used in the validation sets (Supplementary Fig. 9A, B); (3) the findings of mechanistic experiments suggested specific accumulation in their target (microtubules) elicited by filamin-A-mediated CLIP170 binding (Figs. 5–7); and (4) CDK4 and filamin A showed only low and no correlation, respectively, with Ki67 in

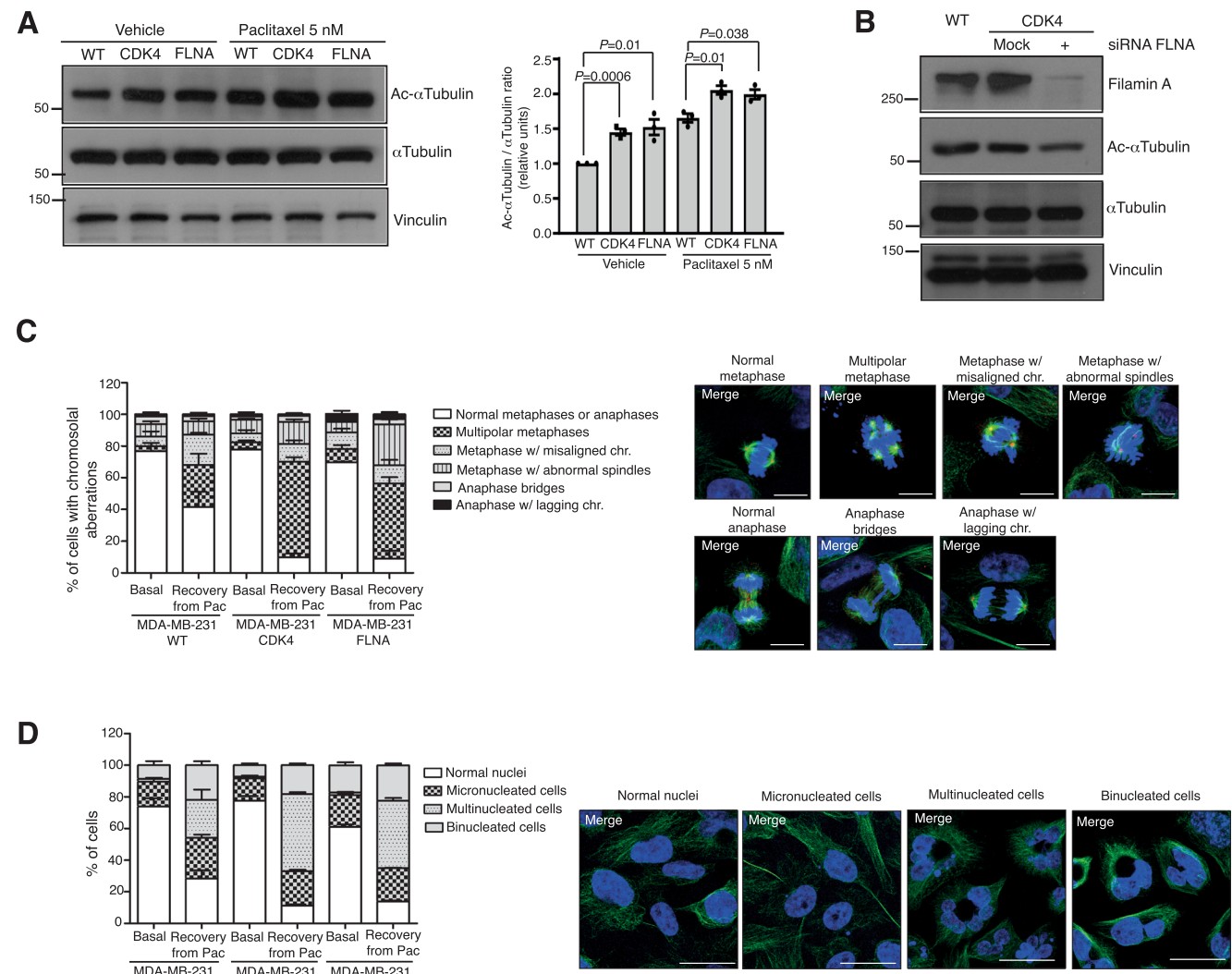

**Fig. 7 | Elevated CDK4 and filamin A lead to increased tubulin acetylation and stabilization, enhancing the effect of paclitaxel on mitotic aberrations.**
**A** Acetylated tubulin (i.e., stabilized microtubules) in parental, MDA-MB-231 CDK4 and MDA-MB-231 FLNA cells, untreated or in response to 5 nM paclitaxel. Samples derived from the same experiment and blots were processed in parallel. On the right-hand side, a bar chart showing the quantification of acetylated tubulin normalized to total tubulin in the different experimental conditions is shown. The quantification is the result of averaging three experimental replicates. Data are presented as the mean±SEM, *n* = 3 independent experiments; two-sided unpaired t-test. **B** Reversion of increased tubulin acetylation in MDA-MB-231 CDK4 cells by filamin A knockdown. Samples derived from the same experiment and blots were processed in parallel. **C** Evaluation of abnormal cell divisions, including metaphases and anaphases, in parental, CDK4 and FLNA MDA-MB-231 variants. Cells were exposed to 5 nM paclitaxel for 4 h, washed, and allowed to recover for 48 h. At this point, the percentages of normal metaphases/anaphases, multipolar metaphases, metaphases with misaligned chromosomes, metaphases with abnormal mitotic spindles, anaphases bridges, and anaphases with lagging chromosomes were quantified in each cell line. Data are presented as the mean ± SEM, *n* = 3 independent experiments. Supplementary Data 4 shows the percentage of cells displaying each aberration in either baseline or treated parental, CDK4 or FLNA MDA-MB-231 variants. The pictures on the right-hand side depict examples of each mitotic aberration type. Scale bars: 10 μm. **D** Same as in (**C**) regarding abnormal nuclei resulting from the previous aberrant processes. Scale bars: 25 μm. Source data are provided as a Source data file.

the TNBC series (Supplementary Fig. 9C), but both were strongly implicated in the sensitization mechanism. The previous state-of-the-art knowledge in the field of predicting sensitivity to chemotherapy in breast cancer is limited to allocating chemotherapy to those patients with a high replicative fraction (i.e., high KI67 staining) or high-risk score in multigene expression panels. This knowledge, while correct in the practical sense [i.e., low-replication breast cancers derive low benefit from cytotoxics[47,48,50]] is ultimately nonspecific and conceptually incomplete. For example, one broad validation of a gene-expression score (the 21-gene assay) was performed in a patient cohort treated with single-agent tamoxifen. Patients with high scores experienced relapse[51], and since then, they have been considered candidates for chemotherapy. The interpretation, however, is not that those patients obtained benefit specifically from one or another cytotoxic drug; it simply allows concluding that the benefit from tamoxifen was insufficient and thus they were offered multiagent chemotherapy regimens without a rationale choice of one cytotoxic over another. Understanding the mechanisms behind increased sensitivity in tumours with an increased replication fraction or increased CDK4/filamin A levels allows, on the one hand, the making of rational treatment allocation decisions (for example, paclitaxel for patients with high CDK4 and filamin A levels and other cytotoxics for other patients), while on the other hand, it allows a more rational search for treatment approaches for resistant phenotypes (in this example, aiming for filamin A or CLIP-170 regulation in high-risk tumours that will require chemotherapy but do not show positive biomarkers of sensitivity).

Our study has several limitations. One potential limitation is the lack of a microdissection of the samples of the discovery set.

The contamination of test samples by nontumor tissue is an important limitation in translational cancer studies (e.g., low variant allele frequency mutations can be overlooked if sequenced with low depth). However, phosphorylation events display manyfold regulation from one tissue to another; thus, even in cases where tumour and nontumor tissue are mixed, these post-translational modification events can be detected. Microdissection reagents decrease the amount of protein that can be obtained from tumour samples and interfere with the phosphopurification experimental protocols; since the amount of protein required for phosphoproteomic analyses is approximately 100-fold higher than what it is required for proteomic runs, we chose to proceed with macrodissection. All samples were macrodissected by an expert pathologist and were of >75% tumour purity. Thus, it is unlikely that the lack of microdissection impacted our results for two reasons: first, because highly regulated events such as phosphorylation would still be detected in the case of low tumour purity; second, because all the candidate biomarkers that were detected in the phosphoproteomic runs underwent confirmation in two external patient series. Thus, although contamination by nontumor tissue may have led us to overlook some additional potential biomarkers, it does not affect our conclusions about the involvement of CDK4 and filamin A in sensitizing tumour cells to paclitaxel.

The second limitation also concerns the discovery experiment: the lack of a normalization of phosphopeptide intensities by total protein intensity could be perceived as a shortcoming. Although the native protein can experience regulation from one sample to another, this is always of much smaller magnitude than the changes in phosphorylation. Thus, it is unlikely that many of the detected events with $10^5$–$10^7$-fold regulation had not been detected if normalization techniques were used. Regardless, our strategy was to maximize the number of candidate biomarkers found in the proteomic experiment, applying low FDR boundaries for the KSEAs and, deliberately, not normalizing phosphoprotein intensity by total protein intensity (since this would have duplicated the cost and duration of the discovery step): no matter how reliable these candidate markers would have been because of the boundaries set in the discovery experiment, they still would have had to undergo confirmation in external sets and experimental mechanistic validation. Studies that base their conclusions exclusively on proteomic techniques (or other -omic techniques) obviously require more strict boundaries and different experimental processes[15–17,52,53]. However, in a clinical oncology biomarker study, what matters most is whether the biomarkers can be confirmed and actually maintain a mechanistic relationship with the feature under study regardless of the origin of the candidate markers (literature, serendipitous experimental results, or discovery experiments, just to name a few), which we believe we have achieved. We have successfully applied the same strategy in the past[10,49], and we think that it can be safely stated that this is an acceptable discovery biomarker strategy in clinical proteomics, particularly for difficult-to-solve translational oncology problems.

Regarding the discovery experiment, another point of criticism could be raised, which is the imbalance in certain clinical/pathologic characteristics between the patients who received paclitaxel monotherapy or combination treatment (Supplementary Table 1). Correlative studies are often performed with samples originating in large randomized trials, and usually, it is uncommon to observe 100% success in sample retrieval or sample validity. Thus, the question of whether the patient characteristics of the patient subcohort constituted by those with valid samples resemble those of the full cohort is normally addressed in such studies to be able to conclude whether the obtained results apply to the full cohort. We cannot confidently conclude that the nonsignificant differences in tumour size, grade, nodal status or Ki67 resulted in meaningful biases in the discovery set experiment, since although hierarchical clustering showed sample clustering by treatment arm (Fig. 1C), other clustering techniques

(Supplementary Fig. 1) did not. Clustering algorithms have limitations[54] and are subjective since there are many choices of distances, linkages and numbers of groups that can affect the results. The differences obtained when the three clustering methods were applied may stem from the fact that the imbalances between treatment arms were not statistically significant or from the cluster methodologies per se. Nevertheless, potential biases in the discovery patient set and disagreement between clustering results when different methodologies are used would be relevant if we were aiming to establish conclusions just on the basis of the screening, attempting to extract conclusions about the biomarker role of filamin and CDK4 just for the NCT01484080 trial based only on this screening, or trying to classify patient subgroups on the basis of the discovery set. An imbalanced discovery set can impact the percentage of hits that are confirmed out of the screening candidates (introducing noise, i.e., yielding many hits that are not subsequently confirmed; in our study, only 2 out of 11 initial candidates were so). However, since our objectives were to determine potential biomarkers for the general TNBC population treated with paclitaxel and understand them from the biological point of view, filtering the candidate biomarkers through two external patient cohorts and preclinical experimentation can serve those purposes. Thus, we think it is unlikely that such an imbalance affected the results and conclusions of this study. However, the imbalances between the two arms, and above all, the fact that one arm studied the response to a single drug and the other arm studied the response to a combination of drugs with two different mechanisms of action (microtubule binder – paclitaxel – or kinase inhibitor – nintedanib), may explain why there are differences between global appearance of the volcano plots shown in Fig. 3A and C (different lower bounds for nonadjusted $P$ values in responders and nonresponders, or apparent flatlines in the $P$ values, which are the results of changes only in the 3rd to 5th decimal place in the $P$ values of the nonsignificantly regulated peptides, thus not being truly flat). Examples of apparent flatlines in the nonsignificant portion of the data are abundant in the literature reporting phosphoproteomic screenings[55–57], which may be an inherent issue of this discovery technique but can also be seen in gene-expression volcano plots[58].

Taken together, our results suggest that although elevated CDK4 levels constitute a tumour progression factor, they also sensitize tumour cells to paclitaxel in a specific manner. This collateral effect, mediated by filamin A and CLIP-170, uncovers a potential liability with an easy-to-determine biomarker for TNBC. Our data ultimately showcase clinical phosphoproteomics as a tool to understand tumour biology in the context of cancer treatments, evidencing a great potential for complex contexts characterized by a lack of oncogene addiction hits.

## Methods

### Patient tumour samples

Patients enrolled in the CNIO-BR-2010-03 trial that consented for tumour sampling underwent image-guided (ultrasound or MRI) tumour biopsy. Three tissue cores obtained with a 14G needle were obtained and snap-frozen upon verification of presence of tumour tissue by the imprinting procedure, within less than 15 min after biopsy. The samples were stored in each study site and shipped to CNIO after trial completion.

Regarding the two patient sets samples used for immunohistochemical biomarker filtering and confirmation, an ad-hoc protocol was approved at three collaborating hospitals (Hospital Universitario Quiron, Hospital de Fuenlabrada and Hospital 12 de Octubre; protocol approval number CEI: 11/37). Formalin-fixed, paraffin-embedded samples from the pathology archive were retrieved for those patients that signed the informed consent form. The study was conducted in accordance with the Declaration of Helsinki and approved by the Institutional Review Board.

Both in the samples from the CNIO-BR-2010-03 trial, and in the other two sample sets used for biomarker confirmation, the definition of response to neoadjuvant paclitaxel-based treatment was achieving a pCR according to the Residual Cancer Burden method described by Symmans and Pusztai[11,25].

## Proteomics

**Sample preparation.** Protein extraction from the frozen tumour samples was performed with a urea lysis buffer (8 M urea, 50 mM Tris.HCl pH 8, 100 mM NaCl, 1X Roche Protease inhibitor, 1X Roche PhosSTOP phosphatase inhibitor) and using mechanical disruption with a Next Advance Bullet Blender with 1.0 mm glass beads and the following instrument settings: Speed 8, Time 3. Post disruption the buffer was removed and the beads discarded. The tissue extract was incubated at room temperature for 2 h with mixing at 2000 rpm in an Eppendorf ThermoMixer. The extract was centrifuged for 10 min at $5000 \times g$ and 4 °C. The protein concentration of the cleared extract was determined using a Qubit protein assay (Invitrogen). Protein was reduced with 10 mM dithiothreitol at 25 °C for 30 min followed by alkylation with 15 mM iodoacetamide at 25 °C for 45 min in the dark. Proteins were concentrated by addition of 4× volumes of −20 °C acetone and overnight incubation at −20 °C. Precipitated protein was collected by centrifuging for 10 min at $5000 \times g$ and 4 °C. Protein pellets were washed twice with −20 °C acetone. Pellets were air dried to remove residual acetone. The washed pellets were reconstituted in 400 μL of urea lysis buffer the protein concentration was determined using a Qubit protein assay (Invitrogen). Protein digestion was performed by addition of 20 μg sequencing grade trypsin (Promega) to each sample and overnight incubation (16 h) at 37 °C. The final digest volume was 2 mL adjusted with 25 mM ammonium bicarbonate. The digest was terminated with the addition of 10 μL TFA.

**Mass spectrometry.** Each digest sample was processed by solid phase extraction (SPE) using a Waters HLB PRiME 30 mg capacity C18 cartridge and gravity flow. Firstly, samples were loaded under vacuum at 5inHg and the cartridge was washed twice with 1 mL 0.1% TFA at 5inHg. Peptides were eluted with 2 × 500 μL of 90% acetonitrile, 0.1% TFA at 5inHg and peptide concentrations were determined by UV absorbance at 280 nm.

Phosphopeptides were enriched using Titansphere $TiO_2$ tips from GL sciences using the vendor protocol. Briefly, phosphopeptides were eluted from the tips using two eluents:

50 μL 5% NH4OH in water and 50 μL 5% Pyrrolidine in acetonitrile. The two elutions were pooled together and neutralized with acetic acid 50% and dried. Samples were reconstituted in 100 μL 0.1% trifluoroacetic (TFA) acid. Each enriched sample was desalted using a StageTip (ThermoFisher P/N SP301) per the vendor protocol. Peptides were dried and reconstituted in 70 μL of 0.1% TFA prior to analysis. Half of each enriched sample was analysed by nano LC-MS/MS with a Waters NanoAcquity HPLC system interfaced to a ThermoFisher Q Exactive mass spectrometer. Peptides were loaded on a trapping column and eluted over a 75 μm analytical column at 350 nL/min; both columns were packed with JupiterProteo resin (Phenomenex). The injection volume was 30 μL. The mass spectrometer was operated in data-dependent mode, with the Orbitrap operating at 60,000 FWHM and 17,500 FWHM for MS and MS/MS respectively. The fifteen most abundant ions were selected for MS/MS.

**Data processing.** Data were processed with MaxQuant version 1.5.0.25 (Max Planck Institute for Biochemistry). The mass spectrometry proteomics data have been deposited to the ProteomeXchange Consortium via the Pride partner repository with the dataset identifier PXD034355. The fragmentation spectra were searched against the *Homo sapiens* Uniprot database (downloaded on 23-12-2013), using Andromeda as the search engine. The precursor mass tolerances were

set to 20 ppm for the first search and 4.5 for the main search. Also, 0.05 and 0.5 Da were used for FT and IT detectors. Carbamidomethylation of cysteine was considered as fixed modifications, whereas oxidation of methionine (M); phosphorylation on serine (S), threonine (T) and tyrosine (Y); and protein N-terminal acetylation were chosen as a variable modification, and up to two tryptic missed cleavages were allowed. The match between run function was enabled. A target-decoy database searching strategy was used to evaluate the false-discovery rates (FDRs) at the peptide and protein level.

The identification of kinase-specific substrates was evaluated using linear sequence motifs analysis implemented in MaxQuant. For the identification of phosphorylated motifs Maxquant used the PhosphoMotif Finder search tool at Human Protein Reference Database was used (http://www.hprd.org/PhosphoMotif_finder).

## Pull down and mass spectrometry analysis

For the immunoprecipitation studies, whole-cell lysates were prepared in RIPA lysis buffer (Sigma-Aldrich, #R0278) containing 1% Halt™ Protease & Phosphatase inhibitor cocktail, EDTA-free (Thermo Scientific #78441). Antibodies (anti-Filamin A -Abcam #ab254184-, anti-CDK4 -Invitrogen #MA5-12984- and anti-alpha Tubulin -Abcam#ab7291-) and control isotypes IgGs were firstly incubated with protein lysates for an hour on rotation at 4 °C (4 μg antibody/mg protein lysate). Then, protein A/G Plus agarose beads (Santa Cruz, #sc-2003) were added and the mix were incubated in rotation overnight at 4 °C. For western blotting agarose beads were washed three times with lysis buffer and then boiled in presence of laemmli buffer (1X) (Sigma-Aldrich, #S3401). For mass spectrometry studies, proteins were eluted from the agarose beads in two consecutive steps by shaking for 10 min at 1250 rpm in an Eppendorf Thermomixer in 100 μL of elution buffer (8 M Urea, 100 mM Tris-HCl pH = 8.0). The supernatant obtained was digested by means of standard FASP protocol. Briefly, proteins were reduced (15 mM TCEP, 30 min, RT), alkylated (50 mM CAA, 20 min in the dark, RT) and sequentially digested with Lys-C (Wako) (protein:enzyme ratio 1:50, 4 h at RT) and trypsin (Promega, #V5071) (protein:enzyme ratio 1:50, o/n at 37 °C). Resulting peptides were desalted using home-made C18 Stage-tips. For the proteomic analysis, LC-MS/MS was carried out by coupling an Ultimate 3000 RSLCnano System (Dionex) with a Q-Exactive HF-X mass spectrometer (Thermo-Scientific). Peptides were loaded into a trap column (Acclaim PepMapTM 100, 100 μm × 2 cm, ThermoScientific) over 3 min at a flow rate of 10 μl/min in 0.1% FA. Then peptides were transferred to an analytical column (PepMapTM RSLC C18, 2 μm, 75 μm × 50 cm, ThermoScientific) and separated using a 60 min effective linear gradient (buffer A: 0.1% FA; buffer B: 100% ACN, 0.1% FA) at a flow rate of 250 nL/min. The gradient used was: 0–3 min 2% B, 3–5 min 6% B, 5–36 min 17.5% B, 36–60 min 25% B, 60–63 min 33% B, 63–65 min 45% B, 65–70 min 98% B, 70–80 min 2% B. The peptides were electrosprayed (1.5 kV) into the mass spectrometer through a heated capillary at 300 °C and an Ion-funnel RF level of 40%.

The mass spectrometer was operated in a data-dependent mode, with an automatic switch between MS and MS/MS scans using a top 12 method (minimum AGC target 1E3) and a dynamic exclusion of 20 s. MS (350–1400 *m/z*) and MS/MS spectra were acquired with a resolution of 60,000 and 30,000 FWHM (200 *m/z*), respectively. Peptides were isolated using a 1.4 Th window and fragmented using higher-energy collisional dissociation (HCD) at 27% normalized collision energy. The ion target values were 3E6 for MS (25 ms maximum injection time) and 1E5 for MS/MS (54 ms maximum injection time). Samples were analysed twice.

For data analysis, raw files were processed with MaxQuant (v 1.6.10.43) using the standard settings against a human protein database (UniProtKB/Swiss-Prot, 20,373 sequences) supplemented with contaminants. Label-free quantification was done with match between runs (match window of 0.7 min and alignment window of 20 min).

Carbamidomethylation of cysteines was set as a fixed modification whereas oxidation of methionines and protein N-term acetylation as variable modifications. Minimal peptide length was set to 7 amino acids and a maximum of two tryptic missed-cleavages were allowed. Results were filtered at 0.01 FDR (peptide and protein level).

For each pair bait/control, the data were normalized by the median of the ratio. For the purpose of calculating ratios, given the fact that proteins identified in the bait pull-downs are in the order of $10^7$ to $10^9$ psm, and the same proteins in the isotype pull-downs are in the order of $10^4$–$10^5$ psm, or simply non-detected (0 psm), non-detected proteins (0 psm) were transformed to 1 psm. This transformation allows performing ratio calculations (i.e., dividing a protein of high abundance in one condition by "0" in other condition turns into dividing it by 1, which still has the biological meaning of negligible protein abundance). Proteins with fold-change in Log2 > 2 or not identified in IgG control and with at least 3 psm in bait or 1 psm in both technical replicates, were considered as potential candidates.

### Antibody setup and immunohistochemistry staining

In order to confirm or reject the targets discovered by mass spectrometry analysis, immunohistochemistry stainings were performed when available antibodies were found. For this purpose, breast cancer tissues for routine histological analysis were fixed in 10% buffered formalin (Sigma-Aldrich, #HT501128) and embedded in paraffin. Tissue microarrays were mounted with two 1-mm cores per sample (Quick-Ray Instruments, UNITMA). An expert pathologist examined a template H&E slide from each sample to select the areas for core selection. Immunohistochemical staining was performed on 2.5-μm TMA sections. Immunohistochemistry was performed using an automated protocol developed for the Autostainer Link automated slide staining system (DAKO, Agilent). All steps were performed on this staining platform using validated reagents, including deparaffinization, antigen retrieval (cell conditioning), and antibody incubation and detection.

The following antibodies were used for IHC: phospho-P70S6K (Thr389) (clone 1A5, Cell Signaling #9206, 1:150), CDK4 (clone DSC5, Millipore #MAB8879, 1:25), phospho-PKC-PAN (Sigma-Aldrich #SAB450499, 1:100), phospho-AMPK1/2 (Thr172) (Cell Signaling #2531, 1:100), HMGCR (Abcam #ab242315, 1:50), phospho-CAMKIV (T196 + T200) (Abcam #ab59424, 1:200), phospho-Vimentin (Ser56) (Abcam #ab227081,1:100), Filamin A (Abcam #ab189183, 1:400), phospho-Filamin A (Ser2152) (Invitrogen #PA5-104838, 1:200), phospho-YAP1 (Ser127) (Abcam #ab76252,1:750) and Plectin (clone E398P, Abcam #ab32528, 1:600).

Corresponding TMA were acquired and digitalized using the AxioScan.Z1 system (Zeiss). Digitalized images were automatically analysed with the ZEN 2.3 lite software (Zeiss). For staining quartile determination, H-scores were calculated by formula: ((% of Area High Intensity × 3) + (% of Area Medium Intensity × 2) + (% of Area Low Intensity × 1))/100.

### Cell lines, cDNA transfection and siRNA knockdown

The human triple-negative breast cancer cell line MDA-MB-231 was acquired from the American Type Culture Collection (ATCC) (ATCC, #HTB-26). Cells were maintained following the ATCC recommendations and routinely tested for mycoplasma using the MycoalertTM Mycoplasma Detection Kit (Lonza, #LT07-318).

A vector for human CDK4 overexpression, pRc/CMV-CDK4 was kindly provided by M. Malumbres to generate stably MDA-MB-231 cells overexpressing CDK4. Cells were transfected at 60–70% density with Lipofectamine 3000 (Thermo Fisher Scientific, #L300008) according to the manufacturer's instructions. An empty vector was used as control; these cells were named MDA-MB-231 WT along the manuscript's main text. Transduced cells were selected in 1500 μg/mL neomycin for 2 weeks. For Filamin A knockdown, MDA-MB-231 CDK4 cells were seed in 6-well plates and transfected with Stealth siRNA Filamin A (250 pmol per well) (Thermo Fisher Scientific) using Lipofectamine 2000 (Thermo Fisher Scientific, #12566014) according to the manufacturer's instructions. The knockdown efficiency of Filamin A was tested by western blot after 48 h. A nontargeting siRNA (scramble) was used as a control.

To generate stably MDA-MB-231 cells overexpressing Filamin A, $1 \times 10^6$ cells were resuspended in 100 μL mixture of Opti-MEM (Thermo Fisher Scientific, #11524456) and 10 μg of pcDNA3-myc-FLNa vector (Addgene, #8982). Then, the mixture was transferred to a sterile Amaxa nucleofection cuvette (Clontech Laboratories) and cells were electroporated with a NEPA21 Super Electroporator (Nepagene) using the appropriate nucleofection programme. After nucleofection, the cells were immediately transferred into 3 mL of prewarmed medium in a 6-well plate. Transduced cells were selected in 1500 μg/mL neomycin for 2 weeks.

### Colony-formation assays

Colony-formation assays were conducted as follows: MDA-MB-231 WT, MDA-MB-231 CDK4 and MDA-MB-231 FLNA and MDA-MB-231 CDK4 siRNA FLNA cell lines were seeded at densities of 2000 cells per well in 12-well plates. After overnight incubation, medium was replaced with fresh medium with either vehicle (control) or drugs. Media and drugs were refreshed every 3–4 days. After 10 days of culture, cells were fixed and stained with 0.1% (w/v) crystal violet in 10% (v/v) ethanol. All experiments were performed at least in triplicate. The well area covered by colonies (colony area intensity) was quantified automatically from flatbed scanner-acquired images of colony assays conducted in multi-well plates using the ImageJ software.

To determine the inhibitory concentration of 50% (IC50) of paclitaxel, adriamycin or cisplatin in MDA-MB-231 cell lines, clonogenic survival assays were performed. Cells were exposed to increasing concentration range of each drug and the IC50 values were derived by a sigmoidal dose-response (variable slope) curve using GraphPad Prism software version 5.04. Values represent the mean of at least three independent experiments.

### Immunoblots

Cells were washed 2× with PBS and harvested in cold RIPA Buffer containing 1% protease and phosphatase inhibitor cocktail. Cell lysates were incubated at 4 °C for 15 min, sonicated for 15 min and clarified by centrifugation at $14,000 \times g$ at 4 °C for 30 min. Protein concentration was estimated by BCA protein assay kit (Pierce TM BCA Protein Assay Kit, #23227) following the manufacture's instruction. 20 μg of proteins per sample were loaded on SDS-PAGE gel and transferred to nitrocellulose membranes for further processing. 5% BSA was used to block the membrane for 60 min at room temperature, followed by overnight incubation at 4 °C with the primary antibodies.

The following primary antibodies were used: CDK4 (Cell Signaling, #12790, 1:1000), Filamin A (Abcam, #ab189183, 1:1000), acetylated alpha-Tubulin (K40) (Santa Cruz Biotechnology, #sc-23950, 1:5000), Vinculin (Sigma-Aldrich, #V913, 1:10,000), βActin (clone AC-15, 1:10,000) (Sigma-Aldrich, #A1978), CLIP-170 (Abcam #ab134907, 1:1000) and alpha-Tubulin (Abcam#ab7291, 1:10,000). Membranes were incubated with appropriate peroxidase-conjugate secondary antibodies (Sigma-Aldrich). Bands were visualized by the enhanced chemiluminescence (ECL) method (Clarity™ Western ECK substrate, Bio-Rad, #170-5060).

Regarding acetylated tubulin bands, band intensities were quantified using ImageJ software (NIH, Bethesda, MD, USA). For this purpose, developed films were scanned and band intensities representing acetylated alpha tubulin and total alpha tubulin expression were quantified. For each sample, the ratio between the amounts of acetylated and total alpha tubulin was calculated. Uncropped scans of blots are supplied in the Source data file.

## RNA extraction and quantitative RT-PCR

Total RNA was extracted from cells with TRIzol reagent (Life Technologies, #15596026) in accordance of the manufacturer's instructions. The same quantity of total RNA (1 μg) was retrotranscribed to cDNA using the Quantitect Reverse Transcription Kit (Qiagen, #205313) (2 min at 42 °C, 15 min at 42 °C, and 3 min at 95 °C). One microliter of cDNA (dilution 1:3) was placed in a 384-well plate with 5 μL of Fast SYBR™ Green Master Mix (Applied Biosystems, #1129726) and 2 μL of the corresponding primers (Filamin A-Fw: TGCTGCCTACTCATGATGC; Filamin A-Rv: GGATG TGTGTCTTCTTCGGC) in a final volume of 10 μL. PCR amplification was performed using the QuantStudio™ 6 Flex Real_time PCR System (Applied Biosystems) under the following thermal cycler conditions: 2 min at 50 °C, 10 min at 95 °C, and 40 cycles (15 s at 95 °C and 1 min at 59 °C). To quantify transcription, the mRNA expression levels of the target genes were normalized to β-Actin. All samples were run in triplicates and relative quantification (RQ) was calculated following the $\Delta Ct$ method: $RQ = 2 - \Delta Ct$, where $\Delta Ct$ is the difference between the Ct of the gene of interest and the Ct of the endogenous gene control β-actin.

## Confocal studies

In order to evaluate the recovery of cells to paclitaxel treatment, we evaluate the percentage of abnormal divisions. The day after seeding cells on glass coverslip a concentration of 5 nM of paclitaxel was added for 4 h. After washing cells with PBS to eliminate remanent paclitaxel, normal medium was added and cells were grown for 48 h. Then, cells were fixed in 4% paraformaldehyde (PFA) for 10 min and permeabilized with 0.5% Triton X-100/PBS plus 0.05%SDS (20 min at room temperature). Coverslips were incubated in blocking solution (3% Bovine Serum Albumin in PBS-T (PBS + 0.05% Tween-20) for 1 h. Primary antibody solution was applied overnight at 4 °C. The following primary antibodies were used: alpha-Tubulin (Abcam, #ab7291) (1:1000) and gamma-Tubulin (Invitrogen, #PA5-34815) (1:1000). Cells were then washed with PBS-T and incubated with Alexa Fluor 488-(Molecular Probes, #A11029) (1:200) or 555-conjugated secondary antibody (Molecular Probes, #A21429) (1:200). Cell nuclei were stained with DAPI (Sigma-Aldrich, #D9542) (1:2000). After washing, coverslips were mounted onto glass microscope slides with Prolong (Invitrogen, #P36930). Images were acquired in a Leica-TCS SP5X confocal microscope, with a HCX PL APO 63× 1.4 numerical aperture (NA) oil-immersion objective using LAS AF version 2.5.1 software. To estimate the abundance of abnormal divisions, all metaphases and anaphases found in each condition were acquired. For abnormal nuclei, 15-random fields were acquired and around 1000 cells at interphase were scored in each assay. All experiments were done by triplicate.

In order to study the cellular localization of CDK4, alpha-Tubulin and Filamin A, cells grown on coverslips were subject to immunofluorescence studies as previously described. The following primary antibodies were used: Filamin A (Abcam, #ab254184) (1:500) and CDK4 (clone DSC5, Millipore, #MAB8879) (1:250) and alpha-Tubulin (Abcam, #ab7291; 1:1000).

For colocalization determination of CDK4 (green) and Filamin A (red), and α-Tubulin (green) and Filamin A (red) 30 fields were acquired and the correlation between pixels were estimated using Definiens Developer v2.5 software. Pearson index for colocalization was performed by mean of customized algorithms programmed in Definiens Developer using more than 250 cells per genotype. When Pearson index values are between 0.5 and 1 cells are considered as positive for colocalization.

## Paclitaxel accumulation and live cell imaging

In order to study the kinetics of paclitaxel accumulation in cells, MDA-MB-231 WT, MDA-MB-231 CDK4, MDA-MB-231 FLNA and MDA-MB-231 CDK4 siRNA FLNA cell lines were seeded at densities of 30,000 cells per well in a μ-slide 8 well chamber (Ibidi) and grew in 10% FBS DMEM medium overnight. After PBS rinsing, cells were incubated with 200 nM Flutax-2 (paclitaxel, Oregon Green 488 conjugate) (Invitrogen, #P22310) and 25 μM verapamil (Sigma-Aldrich, #152-11-4) in 10% FBS Hank's Balanced Salt Solution (HBBS) medium (Sigma-Aldrich, #H6648) for 48 h. Images were taken every 15 min during 48 h using a Leica-TCS SP5X confocal microscope, with a HCX PL APO 20X objective using LAS AF version 2.5.1 software. Quantitation of green fluorescence was performed in live cells using the Definiens Developer XD software with a customized script for detection and quantification of green area.

## Kinase assay

Kinase reactions were carried out at 30 °C for 30 min using specific kinases buffer. Human recombinant CDK4/Cyclin D1 (Sigma-Aldrich, #C0620) was incubated with human recombinant Filamin A (Origene, #TP326488) in kinase assay buffer (20 mM MOPS, pH 7.2, 12.5 mM glycerol 2-phosphate, 25 mM $MgCl_2$, 5 mM EGTA, 2 mM EDTA and DTT 0.25 mM) in presence of 50 μM cold ATP and 1.5 uCi [32 P] ATP. Recombinant RB (Santa Cruz Biotechnology, #sc-4112) was used as a positive phosphorylatable substrate for CDK4/Cyclin D1. Reactions were stopped by addition of Laemmli sample buffer. Radioactive samples were subject to acrylamide gel electrophoresis, followed by gel drying and autoradiography.

## Statistics

Clinical and demographic characteristics of the study patients by treatment arm were compared by the Mann–Whitney, Chi-Squared or Fisher's test as appropriate.

In order to study the relationship with response to paclitaxel of the candidate kinases or phosphoproteins resulting from the analysis of the discovery sets in Set 1 and Set 2, an immunohistochemical H-score was calculated for each candidate. The H-scores were then categorized as follows: for each candidate marker, patients with a H-score above the 75th percentile were encoded as High, whereas those with H-score below the 75th percentile were categorized as Low. In order to find the hazard ratio of obtaining or not a response in case of belonging to the High or Low categories, a univariate logistic regression model was run for each candidate marker in Sets 1 and 2. Then, for each marker with a positive association in the regression model (p-P70S6K, CDK4, p-vimentin, filamin A and HMG-Coa reductase), the proportion of patients achieving a pCR among those with high staining was compared with the proportion of patients achieving a pCR among those with low staining. The proportion of pCR for each category (i.e., High or Low) for each candidate marker was compared with a Chi-square test. For each breast cancer subtype (or the whole Set 1), 5 comparisons were run (one per marker) in Set 1. In Set 2, 5 comparisons were run as well; thus, the Bonferroni correction was applied to the type-1 error. P-values below 0.01 were considered significant, whereas P-values between 0.01 and 0.1 were considered borderline significant.

CDK4 and Filamin A correlation, or their correlation with Ki67 replicative fraction, were investigated with the Pearson´s coefficient in a pairwise manner. Drug sensitivity, mRNA levels and acetylated tubulin levels were compared with two-sided unpaired t-tests. All tests were performed with the SPSS Statistics V.19.0 software.

Unsupervised hierarchical biclustering was performed using Morpheus software (https://software.broadinstitute.org/morpheus). Consensus clustering and pvclust clustering methods were used to assess the robustness of the unsupervised clustering obtained by UPGMA. The R libraries ConsensusClusterPlus[59] and pclust[24] were used for this purpose.

A Gene Set Enrichment Analysis (GSEA)[60] was used to define sets of kinase substrate motifs that shows statistically significant, concordant differences between categories of interest, such as responders or nonresponders to paclitaxel, herein termed Kinase Set Enrichment Analysis (KSEA). To this end, KSEA was applied using annotations for

motifs extracted from Perseus software. Leading proteins data matrix was ranked based on their t-test statistic. KSEA scoring scheme was classic. The kinase-set size limits were established at >5 and <1000. After Kolmogorov-Smirnoff testing with 1000 permutations, those kinase-sets showing FDR < 0.20 were selected as significant.

In order to compare phosphopeptides up- or downregulated in responders or nonresponders in the paclitaxel arm, the paclitaxel plus nintedanib arm, or in the whole trial, normality of the phosphopeptide intensity distribution was tested with the Shapiro Wilk normality test. H0 was that the distributions were normal, and H1 was that they were not. H0 was rejected in the three cases ($P < 2.2 \times 10^{-16}$ in the three cases), and thus, the assumption of non-normality of the data distribution was adopted. According to this the median value for each phosphopeptide was calculated for each condition (i.e., responders or nonresponders, in Arm A, Arm B, or whole trial). The median values were compared with the non-parametric Mann–Whitney Wilcoxon test using 100,000 permutations; thus, $P$-values were calculated with 5 decimal places. The obtained $P$-values were adjusted by FDR using the Benjamini–Hochberg method to account for multiple testing.

Volcano-plots were depicted using the median log fold-change intensity (X-axis) values and raw $P$-values (Y-axis). The plots were generated with the GraphPad Prism software version 5.04. In order to avoid divisions by 0 in the phospho-peptide ratio calculation when two classes were compared (i.e., responders and nonresponders) and any given phospho-peptide was highly abundant in one class and undetectable in the other, the value "0" in spectral intensity was switched to "1" (a change in 1 spectral intensity when phospho-peptides are found in the range of $2^{10}$–$2^{20}$ intensity is biologically negligible, but allows the calculation of the ratios).

### Reporting summary

Further information on research design is available in the Nature Portfolio Reporting Summary linked to this article.

## Data availability

All the data supporting the findings of this study are available within the article and its supplementary information files. The mass spectrometry proteomics data are publicly available in the ProteomeXchage Consortium via the PRIDE partner repository with the dataset identifier PXD034355. Source data are provided with this paper.

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

## Acknowledgements

M.Q.F. is a recipient of the following grants: AES – PI16/00354 and AES – PI 19/00454 funded by the ISCIII and co-funded by the European Regional Development Fund (ERDF), and B2017/BMD3733 (Immunothercan-CM) – Call for Coordinated Research Groups from the Madrid Region – Madrid Regional Government – ERDF funds. R.C. is a recipient of the ISCIII grant PI17/01865, funded by the ISCIII and co-funded by the European Regional Development Fund (ERDF). Boehringer-Ingelheim contributed with a research grant to this project. This study was also funded by a donation from CRIS Contra el Cancer Foundation.

## Author contributions

Conceptual design was conceived by S.M., M.J.B., M.M., R.C. and M.Q.F. S.M., M.J.B., E.C. and D.M. performed experimental work. A.L.L., L.M., I.C., J.C., J.A.G.-S., N.M.-J., J.V.A., L.G.-C., R.M., R.S.-B., J.M.-C., M.M., A.M., R.C. and M.Q.F. managed the patient and samples accrual. Bioinformatic analysis was performed by G.G.-L. and C.F.-T. Proteomic experiments were carried out by P.X.-.E and J.M. Statistical analysis was done by S.S.-L. and N.M. All authors contributed to data interpretation and manuscript writing and have approved the final manuscript.

## Competing interests

The authors declare no competing interests.

## Additional information

[1]Breast Cancer Clinical Research Unit Centro Nacional de Investigaciones Oncológicas – CNIO, Madrid, Spain. [2]Medical Oncology Department, Hospital Clínico Universitario, Valencia, Spain. [3]Medical Oncology Department, Hospital Universitario 12 de Octubre, Madrid, Spain. [4]Medical Oncology Department MD, Anderson Cancer Center Madrid, Madrid, Spain. [5]International Breast Cancer Center Quiron Group, Barcelona, Spain. [6]Vall d'Hebron Institute of Oncology, Vall d'Hebron Hospital, Barcelona, Spain. [7]Medical Oncology Department, Hospital Clinico San Carlos, Madrid, Spain. [8]Medical Oncoogy Department Institut, Catala d'Oncologia-IDIBELL L'Hospitalet de, Llobregat, Spain. [9]Medical Oncology Department, Hospital Universitario Ramon y Cajal, Madrid, Spain. [10]Histopathology Unit Centro Nacional de Investigaciones Oncológicas – CNIO, Madrid, Spain. [11]Proteomics Unit Centro Nacional de Investigaciones Oncológicas – CNIO, Madrid, Spain. [12]Medical Oncology Department, Hospital Universitario Quironsalud, Madrid, Spain. [13]Pathology Department, Hospital Universitario Quironsalud, Madrid, Spain. [14]Bioinformatics Unit Centro Nacional de Investigaciones Oncológicas – CNIO, Madrid, Spain. [15]Genetic & Molecular Epidemiology Group Centro Nacional de Investigaciones Oncológicas – CNIO, Madrid, Spain. [16]Pathology Department, Hospital Universitario 12 de Octubre, Madrid, Spain. [17]Pathology Department, Hospital Universitario de Fuenlabrada, Madrid, Spain. [18]Confocal Microscopy Unit Centro Nacional de Investigaciones Oncológicas – CNIO, Madrid, Spain. [19]Cell Division and Cancer Group Centro Nacional de Investigaciones Oncológicas – CNIO, Madrid, Spain. [20]Medical Oncology Department, Hospital Universitario La Princesa, Madrid, Spain. [21]Endowed Chair of Personalized Precision Medicine Universidad Autonoma de Madrid (UAM) – Fundacion Instituto Roche, Madrid, Spain. ✉e-mail: mquintela@cnio.es

