## [Peer Review File · Nature Communications]

Reviewer #2 (Remarks to the Author): expertise in paclitaxel resistance and clinical relevance in breast cancer

This work has identified that high co-expression of CDK4 and Filamin-A predict for high sensitivity to preoperative paclitaxel in HER2 negative breast cancers.

1. Definition of pCR is not clear. PCR is RCB=0. Are analysis confined to RCB=0 patients?
2. Lack of microdissection of the breast cancer samples does not allow for uniform analysis of high tumor content samples and confounds interpretation of quantitative protein expression in the cancers.
3. Need for multivariate analysis of CDK4 and Filamin levels with other known predictors for pCR including grade, Ki-67, degree of ER and PR expression.
4. Insufficient information is provided about the 2 validation tissue sets, i.e., patient and tumor characteristics, preoperative treatments, definition of response to therapy, etc.
5. Sometimes difficult to follow and understand the data in Results because not consistently and clearly stated which biopsy results are being described, i.e., responder vs non-responder; pre- vs post- Nintedanib. Also sometimes not clear if specific proteins were over-expressed or up-or-down regulated in parts of the manuscript.
6. In patients who had residual disease post pre-op paclitaxel sufficient for mass spectra analysis, was CDK4 and Filamin A expression analyzed?

Reviewer #3 (Remarks to the Author): expertise in paclitaxel mechanism of action

In this manuscript, "Phosphoproteomic analysis of neoadjuvant breast cancer reveals a cluster of patients with increased sensitivity to paclitaxel driven by CDK4 and Filamin A", Mouron et al present phosphoproteomic analyses of breast cancer patient tumor data sets from neoadjuvant trials. Identifying markers of chemotherapy sensitivity is important as it could spare non-responders the toxicities associated with ineffective therapies. Specifically with their methodology they discovered several candidate markers of Paclitaxel sensitivity of patient cancers. They also present some functional insight into the mechanisms responsible for regulation of Paclitaxel sensitivity by these candidate molecules. Overall, it is an intriguing, well done study but several questions need to be addressed

- 1) The manuscript as presented is somewhat disjointed with the first half focused on nintedanib sensitivity (with unrevealing results) then transitioning to paclitaxel sensitivity. Why not focus the manuscript on Paclitaxel sensitivity with more validation sets?
- 2) For the methodology of the phosphoproteomic analysis, it was not clear whether there was any normalization to total levels of a candidate molecule
- 3) For all patient tumors, unlike cell lines, there is a significant degree of heterogeneity in levels of staining for any marker. Did the authors/pathologists observe heterogeneity in staining for CDK4 or Filamin in patient tumor sections, and if so, how was the overall intensity of staining determined (leading to classification into quartiles). Was the pattern of staining distribution relevant at all?
- 4) In Figure 6D, the association between Filamin A and CLIP7 diminishes with overexpression of CDK4 or Filamin A. This seems to contradict the text and needs to be explained.
- 5) In Fig 7A, the signals on the Western blot need to be quantified as higher levels of acetylated tubulin on CDK4 or Filamin A overexpressing cells is difficult to appreciate.
- 6) Table S3 (patient characteristics for the data set) seems to be missing? It is not common for ER+ cancers to have complete pathological complete response and it is more common for patients with Her2+ disease to undergo simultaneous neoadjuvant treatment with chemotherapy+Her2 targeted therapy. Knowing the baseline patient and tumor characteristics is important.
- 7) How specific is CDK4 and Filamin as markers of sensitivity for Paclitaxel? Are they general markers for response of proliferative tumors to many types of chemotherapy?

Reviewer #4 (Remarks to the Author): expert in proteomics

In this manuscript by Mouron S et al., entitled "Phosphoproteomic analysis of neoadjuvant breast cancer reveals a cluster of patients with increased sensitivity to paclitaxel driven by CDK4 and Filamin A", authors aimed to define a set of biomarkers for a cluster of patients with breast cancer and propose high levels of CDK4 and Filamin-A, linked to a 90% chance of achieving a pCR in response to paclitaxel. However, I have major concern with the most important experiment used to build the hypothesis, detection of phospho peptides was used to infer the involvement of specific kinase and other target proteins, however, no direct measure of these proteins/phosphoprotein using mass spectrometry was carried out to confirm the dysregulation of these kinase and/or other target proteins in the patient's tissue samples. With additional suggested experiments and revisions this manuscript may be considered for publication in Nature Communications but in its current state the manuscript is premature and not ready.

Major comments:

1. In Figure 1B, please clarify what data was used for clustering analysis, in figure legend it says "phosphoproteomic spectra" and in text it says, "on the basis of the nature and abundance of phosphopeptides found on each sample". Please clarify what does "nature" of phosphopeptides means here? How was the peptide abundance normalized? Also, please write the number of patient (n=?) shown in dendrogram.
2. For 1352 distinct phospho proteins identified, how many minimum numbers of total and unique peptides were detected? Please provide the numbers of peptides and proteins detected in each condition.
3. In Figure 2, the Kinases driving the phosphoprofiles of responders to paclitaxel or nintedanib is shown. This is an indirect analysis based on detection of phospho peptides, however, it would be important to show and validate the identification and quantification of kinases directly by mass spectrometry.
4. In Figure 2A-2D, for same kinase different number of phospho peptides (vertical black lines) are shown, e.g., for CDK4, in 2A many more number of peptides are shown compared to 2B, 2C and 2D, it would be important 1) to know how different number of detected peptides influence the analysis and the enrichment score 2) how many unique phospho peptides and thereby corresponding proteins are present in different comparison and if there is any common peptides detected across all conditions and if yes has any biological significance 3) for a peptide to be used in such an analysis was there a minimum number of samples in which the peptides needs to be detected or in other words whether or not peptide patients frequency was used in the analysis.
5. For volcano plot (figure 3), what was the minimum number of samples in each group in which a peptide needs to be identified/quantified to be included in the analysis? Proteins like CDK4, Filamin-A, Vimentin and others as listed in line 300-303 are phosphorylated for their physiological activity and I am wondering how many "if any" phosphorylated peptides from these proteins are identified and if can be labelled in volcano plot?
6. I agree with authors that "Mass spectrometry is not an over-the-counter technology" and immunohistochemistry technique is easily accessible, however, for the validation of the proposed potential biomarkers it would be important to show the identification and quantification of 15 potential biomarkers of sensitivity to paclitaxel (p-P70S6K (Thr389), CDK4, Filamin A, HMG-CoA Reductase, p-Vimentin (Ser56), p-AMPK1/2 (Thr172), p-Pan-PKC (Thr497), p-CaMKIV (Thr196/200), p-YAP1 (Ser127), p-40SRPS6 (Ser235), NEK1, p-Filamin A (Ser2512), RAPGEF6, Plectin and METTL14).
7. In Figure 5A, I am wondering why 0.6nM paclitaxel did not show any significance difference between MDA-MB-231 and MDA-MB-231 CDK4 cells, whereas all others lower (0.2 and 0.4nM) and higher (0.8nM) concentration showed the difference?
8. In Figure 6, please clarify whether the quantification is based on spectral counting or intensity? And what does sum of the average intensities (in spectral counts) means? Also, why the scale is different for y-axis (log10) and for x-axis (log2)?
9. For the pull-down experiment, the way data is presented (Figure 6B, 6C) it is hard to judge if there is any significant change in the quantity of target proteins (Filamin A and alpha tubulin) between two conditions. Also, so many different proteins were pulled down along with target proteins which raises the questions about quality of the pull-down experiment. So many proteins are showing similar change, so what was the rationale for selecting CLIP-170 protein for follow up experiment?

10. For figure 7A, 7B, please show quantification of α -tubulin and acetylated α -tubulin by mass spec. Also, which sites are modified and if there is any other PTMs observed?

REVIEWER COMMENTS

Reviewer #1 (Remarks to the Author): expertise in biostatistics

This review is specific to the data and statistical analysis and its related discussion as presented in the manuscript and does not attempt to critique the scientific methods or findings.

Summary of statistical analysis steps: Using biopsies from a subset of 130 breast cancer patients in a previous clinical trial, 15 candidate biomarkers were identified using Kinase Set Enrichment Analysis, which is a variant of GSEA but applied to peptides and kinases instead of gene sets. In a subsequent cohort of 117 early breast cancer patients of 3 subtypes all treated with paclitaxel, these 15 markers underwent investigation with IHC and 5 were found to be associated with response to paclitaxel therapy. And in subsequent cohort of 101 triple-negative patients all treated with paclitaxel, 2 of these 5 biomarkers exhibited associations with pCR. The remainder of the manuscript focuses on investigations into mechanistic aspects of these 2 biomarkers and the role they may play in response to treatment.

Comments on the data: The primary results are the consequence of multiple stages of analysis in separate datasets/cohorts used to narrow down the field of candidate markers. The initial step identified 15 candidate biomarkers using samples from a clinical trial [21] in which 130 patients were randomized to treatment on nintedanib-plus-paclitaxel versus paclitaxel-only treatment; the original trial aimed at testing the treatment effect on the endpoint of residual cancer burden. The current analysis, however, is focused on 85 (“valid”) biopsy samples taken at baseline in that trial, and another 45 (“valid”) samples taken from a subset of these 85 patients in the experimental arm following two weeks of Nintedanib treatment (Figure 1 and line 179). Since these data are exclusively from the experimental arm, the fact that they come from a randomized clinical trial is incidental to this research. The authors, however, focus on that randomization: “*we conducted the largest phosphoproteomic screening in samples from a randomized clinical trial to date*” (Line 149), and “*Taking advantage of the randomized trial design, and the quantitative treatment outcome (RCB), KSEA allowed finding the following potential associations*” (line 209). Even if the two-arm randomization comes into play in the analysis, it would be non-standard to use, for the purpose of biomarker discovery, a cohort that was randomized to treatment, and then use subsequent single-arm treatment cohorts for confirmation. As used here, the clinical-trial randomized design is not related to the findings. Although there is nothing wrong with what was done, it is misleading to describe the findings as if they are in any way the result of a large randomized clinical trial. Such a claim would require a new study specifically designed to test the endpoint of response to paclitaxel among patients exhibiting (or not) the proposed markers.

It is asserted (lines 184-186) that “*Hierarchical clustering showed that ... phosphopeptides found on each sample, patients were not significantly grouped by the hospital in which they were harvested or by trial arm (Figure 1B)*”. However, Figure 1B shows a distinct clustering of phosphopeptide profiles from the standard arm (red) on the left of the dendrogram.

Figure 1A (and the text in line 1066) says 45 samples from the experimental arm (nintedanib-only) had sufficient protein material (i.e., were “valid”), but in Figure 1B, I only count 39 samples (yellow) in the Experimental Arm, and 46 (red) in the Standard Arm.

General Comments:

1. The **Title** and text refer to “*a cluster of patients with increased sensitivity to paclitaxel driven by CDK4 and Filamin A*”. I am not able to see exactly what “cluster” this refers to. It also appears in the subsection with this title (line 280), and apparently the “cluster” being referred to is those patients exhibiting high levels of both Filamin A and CDK4 (line 351). This is not really a “cluster” in the usual sense of the word. Rather, it would be clearer to simply identify this as a phenotype or subset of the patient population that appears to respond to therapy.
2. Another general comment about **terminology**: the term “validation” is used repeatedly; in particular, it is used to describe the second two cohorts as validation sets. In the area of biomarker research and statistical leaning, this word has a specific connotation. Here, however, these cohorts were used in subsequent filtering steps, follow-up confirmation, and continued investigation of the biomarkers in different ways. My suggestion is to refrain from using “validation” to describe the uses of these datasets.
3. Overall, it was difficult to discern the steps that were taken to reach the manuscript’s conclusions. The **Abstract** (line 100) refers to a “screening” step of “*130 patients treated with single-agent paclitaxel*”. Then the Introduction (line 149) refers to conducting “*phosphoproteomic screening in samples from a randomized clinical trial (paclitaxel versus paclitaxel plus nintedanib)*”. This seems to be a contradiction, but once again it seems that the (superfluous) concept of a *randomized clinical trial* gets in the way of the description.

It is additionally stated that the screening step aimed to “*define sensitivity biomarkers and understand the biology behind them.*” This again may be just a linguistic issue, but it seems atypical to “screen” for markers in an initial discovery dataset for the purpose of understanding the biology. Rather, a training or screening dataset may point toward candidate markers, while subsequent investigation and validation uncovers their biological role.

The next step is described (line 101) as: “*Filtering through two independent validation sets (N=218) allowed defining a cluster of patients with high levels of CDK4 and Filamin-A*”. I may be misunderstanding the big picture, but this is also an odd use of “independent validation” data, since seeking clusters of patients is a discovery process, and using these data to “define” a cluster of patients expressing these markers is backwards. (See also the comments above on terminology.)

4. The **Introduction** is not transparent for understanding the steps that lead to the authors conclusions. It focuses on some background and the authors’ expectations, but only in the last paragraph (line 149) is there discussion of this work, which is a “*phosphoproteomic screening*” that leads to a “*predictive factor for paclitaxel, what allowed us to understand the mechanism of increased sensitivity to this drug, driven by an axis [jargon] linking CDK4, Filamin A and CLIP170*”. However, it reads as if conclusions are being made here on the discovery dataset without mention of how this is confirmed in the two “independent validation” datasets.

Line 153: “*mechanisms of resistance against the antiangiogenic agent nintedanib, previously tackled in the preclinical setting, were validated in this study*”. Here, If the aim is to recapitulate and validate a previous finding, the specific hypotheses need to be clarified at the outset, along with specification of the statistical testing being used in the new study.

5. The first page of the **Results** section contains detailed discussions about cohorts and methods and advantages of phosphoproteomic techniques, all of which distract from the primary findings. As with the Abstract and Intro, it is difficult to find a focus on the primary results.

I suggest moving the subsection on “Patient cohort, Samples and Mass Spectra” out of the **Results** section to provide more focus to this discussion. Additionally, it may help to provide a simple flow chart of cohorts, discovery steps and confirmation (“validation”) steps. The two paragraphs beginning at lines 211 and 234 describe two inquiries, but it is confusing to understand how the various subsets of samples (within single-arm, responder vs nonresponders, whole trial) contribute to the conclusions.

Line 172: The notation “*T1/T2 and N0/N1 patients*” needs clarification.

Lines 226-233: KSEA was applied to find Kinase sets that differed between the 85 baseline biopsy samples and the 45 post-nintedanib therapy biopsies. These 45 samples are, apparently, paired with 45 of the 85 baseline samples (Figure 1A), although the samples are treated as independent. However, as noted on the GSEA FAQ website, “GSEA software does not provide paired-sample analysis”. The paired-samples aspect is not mentioned in the manuscript.

Line 352: “*A patient with high levels of both biomarkers had, according to the logistic regression analysis, a 16-fold higher chance of achieving a pCR in response to paclitaxel-based chemotherapy when compared to other patients. Although this study is retrospective, ...*”. Presumably this is meant to say “Patients with high levels of both biomarkers ...” since one cannot perform regression on a single patient. There are no details about the logistic regression model that was fit (sample size, covariates, errors, odds ratio estimates) and it seems to have been fit on a marker that was already deemed to be significantly association with the response. I suggest removing this “retrospective” regression analysis.

6. The “**Data Analysis**” section seems more appropriately labelled as “Data Processing”. The exception is lines 680-689, which seem to belong in the Statistics Section, with an expanded description:
7. The **Statistics** section needs a more complete description of the hypotheses and tests with any parameters identified, something like:

“KSEA [citation] was applied to 500 kinase sets in the discovery cohort of 128 samples [cite the paper/software and identify any parameters used in GSEA software]. Using an FDR cutoff of 0.25, 15 candidate biomarkers were chosen. ...”

Lines 913-915: This sentence is confusing. What is being tested? Is it the pCR rate, as the sentence indicates, or candidate marker expression between categories of pCR?

Line 915: The Yates correction is cited as the authors method to correct for performing multiple T-tests. This is incorrect. Yates is a continuity correction for a chi-squared test, which is not mentioned anywhere. Also not mentioned are how many tests were performed, what the test statistic is, nor how the type-1 error was actually corrected for multiple comparisons.

Line 918: “co-linearity” is used to describe the correlation between two markers. It would be clearer to simply refer to correlation.

Related to this is the discrepancy between the values in Supplementary Figure S5 ($R=0.41$, $p=0.0006$) versus the text in the caption ($R\text{-squared}=0.28$, $p=0.022$).

Figure 3: The volcano plots are not completely described. I struggle to understand why there are large gaps in fold-change and also why there seems to be saturation (flat line) of p-values in each of the “outer” groups (those with huge fold-changes) but this does not appear in the center groups. Can this artifact be explained?

RESPONSE TO REVIEWERS' COMMENTS

Color code

-Black: Reviewers' queries

-Blue: Authors' answers

-Green: Text from the main manuscript quoted in the answers

-Orange: Text from one Reviewer query quoted in the answer to a query issued by other Reviewer.

Comments from Reviewer #1. Expertise in statistics.

This review is specific to the data and statistical analysis and its related discussion as presented in the manuscript and does not attempt to critique the scientific methods or findings.

Summary of statistical analysis steps: Using biopsies from a subset of 130 breast cancer patients in a previous clinical trial, 15 candidate biomarkers were identified using Kinase Set Enrichment Analysis, which is a variant of GSEA but applied to peptides and kinases instead of gene sets. In a subsequent cohort of 117 early breast cancer patients of 3 subtypes all treated with paclitaxel, these 15 markers underwent investigation with IHC and 5 were found to be associated with response to paclitaxel therapy. And in subsequent cohort of 101 triple- negative patients all treated with paclitaxel, 2 of these 5 biomarkers exhibited associations with pCR. The remainder of the manuscript focuses on investigations into mechanistic aspects of these 2 biomarkers and the role they may play in response to treatment.

Query #1: Comments on the data: The primary results are the consequence of multiple stages of analysis in separate datasets/cohorts used to narrow down the field of

candidate markers. The initial step identified 15 candidate biomarkers using samples from a clinical trial [21] in which 130 patients were randomized to treatment on nintedanib-plus-paclitaxel versus paclitaxel-only treatment; the original trial aimed at testing the treatment effect on the endpoint of residual cancer burden. The current analysis, however, is focused on 85 (“valid”) biopsy samples taken at baseline in that trial, and another 45 (“valid”) samples taken from a subset of these 85 patients in the experimental arm following two weeks of Nintedanib treatment (Figure 1 and line 179). Since these data are exclusively from the experimental arm, the fact that they come from a randomized clinical trial is incidental to this research. The authors, however, focus on that randomization: “*we conducted the largest phosphoproteomic screening in samples from a randomized clinical trial to date*” (Line 149), and “*Taking advantage of the randomized trial design, and the quantitative treatment outcome (RCB), KSEA allowed finding the following potential associations*” (line 209). Even if the two-arm randomization comes into play in the analysis, it would be non-standard to use, for the purpose of biomarker discovery, a cohort that was randomized to treatment, and then use subsequent single-arm treatment cohorts for confirmation. As used here, the clinical-trial randomized design is not related to the findings. Although there is nothing wrong with what was done, it is misleading to describe the findings as if they are in any way the result of a large randomized clinical trial. Such a claim would require a new study specifically designed to test the endpoint of response to paclitaxel among patients exhibiting (or not) the proposed markers.

We thank the Reviewer for these insights and the detailed review. Our statistics knowledge is much smaller than the Reviewer’s and thus we have adapted the manuscript to all these observations to adhere to statistical reporting standards.

First of all, in line with Reviewers 2 and 3's questions, we have decided to remove from the manuscript the data related to nintedanib sensitivity or resistance. The trial was originally designed to test whether nintedanib added some efficacy to single-agent paclitaxel in a randomized phase II setting. However, with the advent of immunotherapies and the small effects observed with antiangiogenics such as nintedanib in different trials, attention was switched away from this drug class and most agents (including nintedanib) were discontinued. Since we had some preclinical data about acquired resistance against nintedanib, and the KSEA data suggested that those mechanisms were preserved in patients, we thought that they should be reported; unfortunately, given the low interest in these compounds, and with the lack of subsequent experimental insights or external validation, nintedanib data do not support a standalone manuscript. This is what made us include these data in this manuscript. However, the current manuscript is focused on paclitaxel biomarkers and mechanisms of action; paclitaxel is a key drug in breast cancer, and nintedanib data add little but distraction to this main focus. Thus, eliminating the reported results about this drug makes the manuscript cleaner, more "linear", and also avoids making references to the randomized nature of the trial as an "strength" to the discovery part of the study.

The summary of changes related to the elimination of nintedanib data and reference to the randomized nature is as follows:

-Introduction section:

In the second paragraph: "...understand the basic kinome landscapes underlying the response to the cytotoxic agent paclitaxel and the multitargeted antiangiogenic nintedanib in HER2-negative breast cancer..." → "...understand the basic kinome landscapes underlying the response to the cytotoxic agent paclitaxel in HER2-negative breast cancer."

In the third paragraph: “To that end, we conducted the largest phosphoproteomic screening in samples from a randomized clinical trial to date (paclitaxel versus paclitaxel plus nintedanib in early HER2-negative breast cancer” → “To that end, we conducted a phosphoproteomic screening in samples from a clinical trial that compared single-agent paclitaxel against paclitaxel plus the antiangiogenic nintedanib in early HER2-negative breast cancer”

Also in the third paragraph: “In addition, mechanisms of resistance against the antiangiogenic agent nintedanib, previously tackled in the preclinical setting, were validated in this study” → This text has been deleted.

-Results section, “Patient cohort, samples and mass spectra” sub-section.

In the second paragraph: “...at least 1352 distinct phospho-proteins were identified (Table S1)” → “...at least 1352 distinct phospho-proteins were identified (Table S1). In this study, we will refer exclusively to the baseline samples and their relationship with response to paclitaxel; the effects of nintedanib in the phospho-proteome and their relationship with response or resistance against this agent will be reported separately”.

-Results section, “Kinases and phosphopeptides enrichment among responders or non-responders to paclitaxel or nintedanib” sub-section: The title of this section has been changed, removing “or nintedanib”.

“Taking advantage of the randomized trial design, and the quantitative treatment outcome (RCB), KSEA allowed finding the following potential associations” → “In our sample set, KSEA allowed finding the following potential associations”.

The whole paragraph “First, since a biopsy was obtained before and after nintedanib.....
... against nintedanib are able to up-regulate PKA and AMPK, and sensitive tumors are not” has been deleted.

The introductory sentence of the second paragraph (“The second question of interest was to investigate potential markers of sensitivity or resistance to paclitaxel”) has been removed.

In the third paragraph: the sentence “Phosphorylated sites up-regulated in post-nintedanib versus pre-nintedanib samples in non-responder patients are displayed in Figure 3A, whereas the same comparison in responder patients is shown in Figure 3B” has been deleted.

-Discussion section

In the third paragraph, the sentence “We took advantage of the samples of a randomized clinical trial....” has been modified: “We took advantage of the samples of a clinical trial that compared paclitaxel monotherapy in treatment-naïve early breast cancer patients versus paclitaxel plus the multikinase inhibitor nintedanib”.

The whole fifth paragraph, discussing on the nintedanib results (“In addition, our study has allowed us to confirm that the up-regulation of the nutrient stress... These results could be useful for other tumor types where antiangiogenics are the treatment cornerstone such as liver, kidney or colorectal carcinoma”) has been deleted.

Figures

Accordingly, figures 2A and 2B have been deleted. The title of Figure 2 has also been modified (we have deleted “..or nintedanib”). All references to nintedanib in the figure legend have been removed.

Figures 3A and 3B have been deleted; similarly, “...or nintedanib” or any reference to nintedanib has been removed from Figure 3 legend title and text, respectively. The

accompanying table S3 stays, but sheets 1 and 2 (corresponding to peptide IDs from Figures 3A and 3B) have been deleted.

Query #2: It is asserted (lines 184-186) that “*Hierarchical clustering showed that ... phosphopeptides found on each sample, patients were not significantly grouped by the hospital in which they were harvested or by trial arm (Figure 1B)*”. However, Figure 1B shows is a distinct clustering of phosphopeptide profiles from the standard arm (red) on the left of the dendrogram.

Thank you for the comment. The referee is right. In addition to the UPGMA hierarchical cluster shown in the article, we performed *consensus clustering*¹ to evaluate the robustness of the UPGMA cluster (see below). Consensus clustering considers $k=3$ as the optimal number of groups, however, as can be seen in the consensus matrix the separation is due to only two outlier samples. In other words, we found no robust clusters within the total group of samples consequently we rejected the possibility of significant clustering associated with the trial arm.

(A) This panel shows the Cumulative Distribution Function (CDF) curve under different values of k . At optimal k , the area under the CDF curve will not significantly increase with the increase of k value.

(B) This panel shows the relative change in the area under the CDF curve under different values of k .

(C) Finally, this panel shows the consensus matrix. Consistency values range from 0 to 1; 0 means never clustering together (white) and 1 means always clustering together (dark blue). The consensus clustering¹ considered $k=3$ as the optimal number of groups as shown by the consensus CDF (A) and delta area (B). However, as shown by the consensus matrix here, this separation was due to two outlier samples. Therefore, we found no robust clusters associated with the trial arm.

Furthermore, we also check the previous results applying $pvclust^2$ to our dataset (below). In this case, the algorithm has not been able to locate any cluster within the total group

of samples, only a large one that groups them all together (red square in the figure below).

In this panel, the values on the dendrogram correspond to approximately unbiased (AU) probability p-values (red, left) and Bootstrap Probability (BP) values (green, right) and cluster labels (grey, bottom). Clusters with AU >95 are considered to be significant and highlighted in a box. In agreement to the consensus clustering approach, the pvclust algorithm only found one significant cluster. No significance was associated with the trial arm cluster.

We have clarified this point in the manuscript. We have added the following text in the clustering description: “Consensus clustering¹ did not find robustness in the cluster associated with the trial arm (Figures S1A, S1B and S1C). In agreement to consensus clustering approach, pvclust algorithm² did not find significance associated with the trial arm cluster (Figure S1D).”

The figures shown above are now Supplementary Figures S1A, S1B, S1C and S1D, explaining the lack of significant clustering.

Also, because of adding a flow chart (now Figure 1B) as suggested by the Reviewer in Question #8, Figure 1B is now Figure 1C.

Query #3: Figure 1^a (and the text in line 1066) says 45 samples from the experimental arm (nintedanib-only) had sufficient protein material (i.e., were “valid”), but in Figure 1B, I only count 39 samples (yellow) in the Experimental Arm, and 46 (red) in the Standard Arm.

We apologize for the confusion; it was originated by the slight changes in sample numbers as the sampling processes “flow forward” (i.e., 130 patients; only 123 consent for the baseline biopsy; of those, only 85 have sufficient material, etc).

In addition, we have detected a mistake in the figure labels: the yellow tag actually corresponds to the Standard Arm and the red tag to the Experimental Arm.

The sample count is as follows: One hundred and twenty-three patients out of the 130 that were enrolled in the trial consented for the baseline biopsy (62 in the Experimental Arm and 61 in the Standard Arm). Of those, 46 and 39 samples yielded sufficient protein material respectively; also, 45 of the 58 post-nintedanib biopsies obtained from the patients from the Experimental Arm after nintedanib monotherapy were valid as well.

However, as recommended by the Reviewer below in Question #8, we have explained these points now in a flow chart (new Figure 1B) and in the Figure legend, in order to avoid excessive distractions in the Results section.

Figure 1 has been re-labeled; sample numbers have been added by trial branch so that the number of total and processed samples is better depicted. Figure 1 legend has been changed accordingly. As mentioned above, the samples clustering is now shown in Figure 1C.

Query #4: The Title and text refer to “a cluster of patients with increased sensitivity to paclitaxel driven by CDK4 and Filamin A”. I am not able to see exactly what “cluster” this refers so. It also appears in the subsection with this title (line 280) and apparently the “cluster” being referred to in those patients exhibiting high levels of both Filamin A and CDK4 (line 351). This is not really a “cluster” in the usual sense of the word. Rather, it would be clearer to simply identify this as a phenotype or subset of the patient population that appears to respond to therapy.

We apologize for using the wrong word (we are not native English speakers, hence the confusion). We refer, indeed, to a subgroup or subset of patients, that can be detected by displaying high levels of Fil-A and/or CDK4. We have changed it all along the manuscript (now “subgroup” or “subset”), with the exceptions in which the word “cluster” applies (“consensus clustering”, “hierarchical clustering”, etc).

Importantly, we have changed the title. The title is now: “Phosphoproteomic analysis of neoadjuvant breast cancer reveals a subset of patients with increased sensitivity to paclitaxel driven by CDK4 and Filamin A”.

Query #5: Another general comment about terminology: the term “validation” is used repeatedly; in particular, it is used to describe the second two cohorts as validation sets. In the area of biomarker research and statistical leaning, this word has a specific connotation. Here, however, these cohorts were used in subsequent filtering steps,

follow-up confirmation, and continued investigation of the biomarkers in different ways. My suggestion is to refrain from using “validation” to describe the uses of these datasets.

We thank the Reviewer for this insight. We have reworded the text every time we used the word “validation” or “validate”.

Query #6: Overall, it was difficult to discern the steps that were taken to reach the manuscript’s conclusions. The **Abstract** (line 100) refers to a “screening” step of “130 patients treated with single-agent paclitaxel”. Then the Introduction (line 149) refers to conducting “phosphoproteomic screening in samples from a randomized clinical trial (paclitaxel versus paclitaxel plus nintedanib)”. This seems to be a contradiction, but once again it seems that the (superfluous) concept of a *randomized clinical trial* gets in the way of the description.

We thank the Reviewer for detecting the less clear point of our manuscript. We have rephrased the abstract (now reading “....Here, we conducted a phosphoproteomic screening in a neoadjuvant trial of HER2-negative breast cancer patients (N=130) treated with paclitaxel or paclitaxel plus nintedanib, aiming to find candidate biomarkers of paclitaxel sensitivity...”) in order to clarify the trial design from which the samples were obtained. Now, it is concordant with what its described in the introduction (discussed in Query #1, please see above), where the reference to the randomized nature of the trial has also been withdrawn because of its lack of significance in this context.

It is additionally stated that the screening step aimed to “*define sensitivity biomarkers and understand the biology behind them.*” This again may be just a linguistic issue, but it seems atypical to “screen” for markers in an initial discovery dataset for the purpose of understanding the biology. Rather, a training or screening dataset may point toward candidate markers, while subsequent investigation and validation uncovers their biological role.

Thank you for pointing this out. As mentioned in the previous paragraph, the whole sentence in the abstract has been changed. The new wording is “..Here, we conducted a phosphoproteomic screening in a neoadjuvant trial of HER2-negative breast cancer patients (N=130) treated with paclitaxel or paclitaxel plus nintedanib, aiming to find candidate biomarkers of paclitaxel sensitivity...”); we hope to have corrected both issues with the new narrative.

The next step is described (line 101) as: “*Filtering through two independent validation sets (N=218) allowed defining a cluster of patients with high levels of CDK4 and Filamin-A*”. I may be misunderstanding the big picture, but this is also an odd use of “independent validation” data, since seeking clusters of patients is a discovery process, and using these data to “define” a cluster of patients expressing these markers is backwards. (See also the comments above on terminology.)

Continuing with the abstract rewording, that sentence is now written as follows: “...Filtering 15 candidate biomarkers through 2 independent patient sets (N=218) allowed finding a subgroup of patients characterized by high levels of CDK4 and Filamin-A, who had a 90% chance of achieving a pCR in response to paclitaxel....”.

We believe that now the study flow is more clearly explained than in the previous version.

Query #7: The **Introduction** is not transparent for understanding the steps that lead to the authors conclusions. It focuses on some background and the authors’ expectations, but only in the last paragraph (line 149) is there discussion of this work, which is a “*phosphoproteomic screening*” that leads to a “*predictive factor for paclitaxel, what allowed us to understand the mechanism of increased sensitivity to this drug, driven by an axis [jargon] linking CDK4, Filamin A and CLIP170*”. However, it reads as if

conclusions are being made here on the discovery dataset without mention of how this is confirmed in the two “independent validation” datasets.

We have extended and clarified the explanations of what was made in the last paragraph of the introduction section. The new text is as follows:

“To that end, we conducted a phosphoproteomic screening in samples from a clinical trial that compared paclitaxel against paclitaxel plus the antiangiogenic agent nintedanib in early HER2-negative breast cancer³. The screening led us to 15 candidate biomarkers (kinases and phosphorylated proteins). In order to transfer these results to potential biomarkers assessable with routine clinical tools, candidate biomarkers were then tested in two independent early HER2-negative breast cancer datasets by immunohistochemistry. Two of them preserved predictive association with response to paclitaxel (Filamin A and CDK4). Subsequent experimental work allowed us to understand the mechanism of sensitization, which was driven by a functional axis linking CDK4, Filamin A and CLIP170; this mechanism enhanced sensitivity to paclitaxel but not to other chemotherapeutics”

The last few words (“... to paclitaxel but not to other chemotherapeutics”) is the result of a few experiments requested by other Reviewers, who asked us to prove if the mechanism was specific for paclitaxel, which it was (now shown in Figure S9)

Line 153: “*mechanisms of resistance against the antiangiogenic agent nintedanib, previously tackled in the preclinical setting, were validated in this study*”. Here, If the aim is to recapitulate and validate a previous finding, the specific hypotheses need to be clarified at the outset, along with specification of the statistical testing being used in the new study.

As mentioned above, and also according to the suggestions from other Reviewers of this manuscript, we have removed all data related to nintedanib and have opted to report them elsewhere.

Query #8: The first page of the **Results** section contains detailed discussions about cohorts and methods and advantages of phosphoproteomic techniques, all of which distract from the primary findings. As with the Abstract and Intro, it is difficult to find a focus on the primary results.

I suggest moving the subsection on “Patient cohort, Samples and Mass Spectra” out of the **Results** section to provide more focus to this discussion. Additionally, it may help to provide a simple flow chart of cohorts, discovery steps and confirmation (“validation”) steps.

Following these suggestions, we have considerably shortened the initial text of the result sections.

The first sub-section has been renamed (now “Discovery patient set for phosphoproteomic analysis”).

Although we believe that a minimum information about the trial and samples must be shown here (otherwise readers might be lost about what was done, in which patients and why does this predict sensitivity to paclitaxel if we do not describe that they come from a trial where patients were treated with this drug) and thus we have not simply removed the section, we just mention now the trial registration number, treatment regimen and the endpoint description in 8 lines of text. We reference the full trial report for interested readers, and we explain the rest of relevant data in Figure 1, which includes now the flow chart. The flow chart, among other data, also depicts the number of obtained samples and valid samples for the proteomic analysis.

The minimal description about the obtained mass spectra has been left unchanged. This is common in all major studies reporting on proteomics results in clinical samples such as the CPTAC manuscripts^{4, 5, 6, 7, 8, 9, 10, 11, 12}; the minimum information that is usually shown is the number of mass spectra, the number of peptides and proteins identified, and in case the phospho-fraction was determined, also the number and type of phosphorylation events detected. Since these data are described in just 4.5 lines of text, we think that readers will not lose the main focus of the study. We believe that reporting this minimal set of results is mandatory in a study of this type, since it can attract the attention of general oncologists, experimental scientists, or experts in proteomics; all of them will need some explanation in the areas of the manuscript that fall beyond their area of expertise.

The two paragraphs beginning at lines 211 and 234 describe two inquiries, but it is confusing to understand how the various subsets of samples (within single-arm, responder vs nonresponders, whole trial) contribute to the conclusions.

We apologize for the lack of clarity. The whole section has been simplified: first, the narrative regarding sensitivity markers for nintedanib has been eliminated as explained above; second, the text regarding the comparisons between responders and non-responders has been re-written, simplified, and clarified.

Line 172: The notation "*T1/T2 and N0/N1 patients*" needs clarification.

T1, T2, N0 or N1 corresponds to tumor staging terminology (T1: tumors smaller than 2 cm; T2: tumors between 2 and 5 cm; N0: tumors with no axillary nodes invaded by metastases; N1: tumors with at least one but no more than three invaded axillary nodes).

Since the narrative about the trial samples and phosphoproteomic spectra has been considerably trimmed, as suggested in Query #8, this sentence is no longer in the manuscript. Instead, we cite the main trial report, published a few years ago, for interested readers that want to learn more about the original trial dataset; we simply mention that these patients were all “early HER2-negative breast cancer that received neoadjuvant treatment”.

Lines 226-233: KSEA was applied to find Kinase sets that differed between the 85 baseline biopsy samples and the 45 post-nintedanib therapy biopsies. These 45 samples are, apparently, paired with 45 of the 85 baseline samples (Figure 1A), although the samples are treated as independent. However, as noted on the GSEA FAQ website, “GSEA software does not provide paired-sample analysis”. The paired-samples aspect is not mentioned in the manuscript.

Thank you for the constructive comment. The referee is right. Indeed, GSEA does not incorporate paired-sample analysis although you can generate rankings based on a statistical (e.g. t-statistic) that takes into account the pairing of samples (e.g. paired-t statistic) and work with those rankings. In our case, all samples were treated as independent. We could have generated a ranking for KSEA that took into account the pairing of samples, but this implied losing sample size (~52% samples would have been left out, since a number of patients that had baseline valid sample did not have valid post-nintedanib sample and vice-versa); thus, we decided to include all the samples we had and treat them as independent. However, since the nintedanib samples analysis is no longer in the manuscript, we have not included further discussion about this point in the main text or methods (i.e., the problem of paired samples is no longer an issue).

Line 352: “A patient with high levels of both biomarkers had, according to the logistic regression analysis, a 16-fold higher chance of achieving a pCR in response to paclitaxel-based chemotherapy when compared to other patients. Although this study is retrospective, ...”. Presumably this is meant to say “Patients with high levels of both biomarkers ...” since one cannot perform regression on a single patient. There are no details about the logistic regression model that was fit (sample size, covariates, errors, odds ratio estimates) and it seems to have been fit on a marker that was already deemed to be significantly association with the response. I suggest removing this “retrospective” regression analysis.

We deeply thank the Reviewer for this comment, since we were not aware that this could constitute an analytic problem – now that we think twice about it we agree with the Reviewer, it is true that it is redundant and can lead to “over-fitting”. We have removed the comment, which was only aiming to stress the point that patients with high levels of both markers were highly likely to achieve a complete response, a point that is clear enough by showing the 90% pCR rate.

Query #9: The “**Data Analysis**” section seems more appropriately labelled as “Data Processing”.

Thank you for the suggestion. We have changed the labelling.

The exception is lines 680-689, which seem to belong in the Statistics Section, with an expanded description

We have moved that text to the Statistics Section (further explanations in the next query).

Query #10: The **Statistics** section needs a more complete description of the hypotheses and tests with any parameters identified, something like:

“KSEA [citation] was applied to 500 kinase sets in the discovery cohort of 128 samples [cite the paper/software and identify any parameters used in GSEA software]. Using an FDR cutoff of 0.25, 15 candidate biomarkers were chosen. ...”

We thank the Reviewer for this constructive suggestion. We have re-written the Statistics section.

Lines 913-915: This sentence is confusing. What is being tested? Is it the pCR rate, as the sentence indicates, or candidate marker expression between categories of pCR?

We apologize for the bad narrative of the previous statistics section; when I reviewed the queries I (the corresponding author) realized that the student had dragged the text from a previous manuscript and adapted the changes, incurring in several mistakes.

In that sentence what was aimed to describe was that, for each marker, the proportion of patients achieving a pCR among those patients with high staining levels was compared with the proportion of patients achieving a pCR among those with low staining levels (candidate marker A: % of pCRs in patients with H-score >75th percentile vs % of pCRs in patients with H-score <75th percentile; candidate marker B: % of pCRs in patients H-score >75th percentile vs % pCRs in H-score <75th ; ... etc). That was done for all (15) candidate markers in Set 1, and only for 5 candidate markers in Set 2.

We have re-written the paragraph in the statistics section; we hope that now is clearer.

Line 915: The Yates correction is cited as the authors method to correct for performing multiple T-tests. This is incorrect. Yates is a continuity correction for a chi-squared test, which is not mentioned anywhere. Also not mentioned are how many tests were performed, what the test statistic is, nor how the type-1 error was actually corrected for multiple comparisons.

The Reviewer is right, and we apologize for the mistake in transcribing what was done. What was done was multiple T-testing. In set 1, 15 T-tests were run, one per each candidate biomarker. The type-1 error (alpha) was divided by 15, and the values shown in the manuscript are the adjusted P-values (or q-values). Such adjustment is known as the Bonferroni test, as the Reviewer knows, and not as the “Yates correction”, as it was erroneously written. I (corresponding author) went through the text and realized that the student dragged some text from a previous manuscript – it is still my fault for not reviewing it thoroughly enough.

In set 2, only 5 T-tests were run, one per each of the 5 candidate biomarkers that “passed” set-1 filter. The same adjustment was applied here (5).

The statistic section is now correct and the mistake has been deleted. We have also mentioned the correction in Figure 4 legend.

Line 918: “co-linearity” is used to describe the correlation between two markers. It would be clearer to simply refer to correlation.

Thank you for the correction. We have changed the word.

Related to this is the discrepancy between the values in Supplementary Figure S5 ($R=0.41$, $p=0.0006$) versus the text in the caption ($R\text{-squared}=0.28$, $p=0.022$).

Thank you for detecting this inconsistency, that I overlooked during the final manuscript checkup. The correct values are those included in the Figure. We have eliminated the text in the legend to avoid confusions (Figure S5 is now S8). The reason the wrong number was in the figure legend is because it corresponded to other correlations that were studied (regarding Vimentin, which was another potential biomarker that finally did not “pass” the second sample set filter).

Figure 3: The volcano plots are not completely described. I struggle to understand why there are large gaps in fold-change and also why there seems to be saturation (flat line) of p-values in each of the “outer” groups (those with huge fold-changes) but this does not appear in the center groups. Can this artifact be explained?

We apologize for the insufficient description. We have tried to correct this problem:

-First, we have included in the statistics section the methodology conducted to generate the plots.

-Second, regarding the large gaps in fold-change: as opposed to other more common -omic techniques in translational oncology (e.g., gene expression), it is not unusual to observe phospho-peptides with 1 million-fold regulation or even more. Although a significant part of phospho-peptides are present in both compared conditions, and experience 2-20-fold regulation (central clouds), it is common to observe that a large portion of the identified peptides are regulated 10^6 -fold. In gene expression experiments it is rare to observe such regulations and thus there are not “lateral clouds” in the typical volcano plots; at most, a few dots with no statistical significance, that quite often are deleted from the charts since they add little or no biological information. Here, however, the amount of highly regulated phospho-peptides is very large and thus we believe that it is very important to depict them. Many of these highly regulated phospho-peptides are in fact present in one condition (at high amount) and undetectable in one another. Since the representation of each dot (i.e., each phospho-peptide) requires calculating a ratio (i.e., spectral intensity in condition A / spectral intensity in condition B), those peptides with “0” intensity in one condition (e.g., “responders”) but highly abundant in another (e.g., “non-responders”) were transformed to “1”. Biologically speaking, a value of 1 in spectral intensity is negligible, but allows performing the ratio calculation. This way, peptides with values in the range of millions (or even billions, since some of the “hits” are in the 2^{20} -fold regulation range) in one condition (responders or non-responders) and absent in the other, end up in the lateral clouds. We have added explanation about this

normalization strategy in the statistics section when we explain how the ratios were calculated.

It might be asked that why there is not a continuum in the peptides regulation – we do not really have an answer to this; it may just be the nature of the phenotype regulation in this particular scenario (responders or non-responders to paclitaxel) – many of the signaling proteins implicated in response/lack of it are regulated several dozens of times, whereas others are regulated by a factor of 10^6 .

-Lastly, unfortunately we do not have an answer to the question why some of the dots are grouped in “flat lines” in the lateral clouds. We put the words between quotations because we have gone through the adjusted *P*-values of the phospho-peptides in those clouds and some of them only differ by some changes in the 6th decimal place or even further. Although it looks like they have the exact same adjusted *P*-value to the “human eye”, they have different adjusted *P*-values (although not very different) but they cause the visual effect of a flat line. That is the “reason”, but, being completely honest, we do not have a biological explanation to why several peptides among those with dramatic regulation have a very close adjusted *P*-value (what, strictly speaking, it is not an artifact, it is just the way it is).

Comments from Reviewer #2. Expertise in paclitaxel resistance and clinical relevance in breast cancer.

This work has identified that high co-expression of CDK4 and Filamin-A predict for high sensitivity to preoperative paclitaxel in HER2 negative breast cancers.

Query #1. Definition of pCR is not clear. PCR is RCB=0. Are analysis confined to RCB=0 patients?

We apologize for the confusion we have generated. Our intention was to refer only to patients that had no residual disease after neoadjuvant therapy, or at most, *in situ* residual component in the breast plus no residual cancer in the lymph nodes. We had correctly attributed the first class (no residual disease) to RCB=0, and the second class erroneously (only *in situ* component in the breast with no residual disease in the lymph nodes) to RCB = 1. We have reviewed in depth the two landmark articles about this matter^{13, 14}, including the formula for calculating RCB ($RCB = 1.4 * (F_{inv} * D_{prim})^{0.17} + [4 * (1 - 0.75^{LN}) * D_{met}]$), where F_{inv} is the proportion of the residual tumor that contains invasive carcinoma, D_{prim} is the diameter of the residual primary tumor bed, LN is the number of axillary lymph nodes containing metastatic carcinoma, and D_{met} is the diameter of the largest metastasis in an axillary lymph node; an online calculator is also available at the following [link](http://www3.mdanderson.org/app/medcalc/index.cfm?pagename=jsconvert3) <http://www3.mdanderson.org/app/medcalc/index.cfm?pagename=jsconvert3>). After checking all our samples, we realize that we were talking about pCR or RCB = 0 all the time, since in all patients with no residual invasive tumor and no residual axillary lymph nodes tumor the calculation always yields 0. ($1.4 * 0 * [any\ diameter]^{0.17} + 4 * (1 - 0.75^0) * 0 = 0$).

In the first version we erroneously state that since we consider “responders” patients that had no residual invasive tumor or residual *in situ* carcinoma (in the breast), such patients corresponded to RCB = 0 and RCB = 1, respectively; also, we stated that since patients with RCB = 0 and RCB = 1 have similar long-term prognosis, we included both in the same category (“responders”). However, we were talking all the time about RCB=0 patients.

In the 2022 RCB follow-on publication¹⁴ it is shown that RCB = 1 patients, albeit good, display a significantly worse prognosis than patients with RCB = 0 across all breast cancer subtypes. However, this point is now irrelevant for our study, since we never included patients with RCB = 1. We had included only patients with RCB = 0, although

we erroneously labelled them as RCB = 0 (for those with no residual disease) and RCB = 1 (for those with residual *in situ* carcinoma).

We have corrected and clarified that we only included RCB = 0 patients in our study along the whole manuscript. We thank the Reviewer for detecting this inconsistency.

Query #2. Lack of microdissection of the breast cancer samples does not allow for uniform analysis of high tumor content samples and confounds interpretation of quantitative protein expression in the cancers.

This is an important point in translational studies. Indeed, for example in genomic experiments, “contamination” by non-tumor tissue can lead to inaccurate results, and for mutations with low VAF and depth below 100X, it is possible that such mutations remain overlooked. A similar statement could be valid for proteomic studies, or gene expression studies, where “regulation” with respect to non-tumor tissue may not be enormous.

Our study simply analyzed the phospho-fraction in the discovery set. Unfortunately, most reagents used in the microdissection protocols available at our institute, deteriorate the protein quality and reduce the final amount of protein that can be recovered (approximately a 100-fold higher amount of protein is needed to analyze the phospho-fraction compared to studies where the total protein is analyzed); this process also interferes with the phospho-purification protocols. In our hands, the results are much worse when we micro-dissect the samples.

However, we believe that this has not been an issue in our studies for two reasons:

-First, because the regulation of phospho-events (at least in our hands) can be in the 10^2 - 10^4 range (or even higher) across tissues of different nature (i.e., tumor and non-tumor). We macro-dissected tumors, warranting that all samples from which we extracted protein had at least a 75% tumor content (>90% in most of the cases, actually), led by an expert pathologist that examined an H&E section before and after cutting

(author E. Caleiras). Thus, even if we were macrodissecting in a less careful manner (say, 50% tumor purity), highly regulated events such as phospho-events would still be detectable regardless of contamination by non-tumor tissue.

-The second reason is that the candidate biomarkers were not “accepted” as predictive factors just out of the proteomic experiments. As “perfect” or “faulty” as proteomics techniques (or our technical execution thereof) may be, in our view as translational scientist, they are not useful unless the “hits” are confirmed externally. It is possible that because of some degree of normal tissue contamination some potential biomarkers were overlooked and not discovered; however, what was detected, was subsequently validated in a quite astringent manner, both in external patient series and experimentally. Thus, we do not think that a 10-25% normal-tissue contamination can negatively affect the conclusions of our study, since, at best, they only affect the ability to detect potential biomarkers in the training set.

We have added a paragraph about the potential limitations of our study in the discussion section (second-last paragraph). The potential consequences of micro/macro-dissection have been added.

Query #3. Need for multivariate analysis of CDK4 and Filamin levels with other known predictors for pCR including grade, Ki-67, degree of ER and PR expression.

We thank the Reviewer for pointing this out. In the previous version of the manuscript we had made several tests, adjusted and un-adjusted by the usual co-variates. We acknowledge the fact that this was not properly described/reported in the previous version.

However, an expert biostatistician has reviewed the manuscript (Reviewer #1) and he/she has recommended us to remove the multivariate analysis. We have consulted with the Editorial Office, and since Rev. #1 was a technical reviewer for the statistical

aspects, they have recommended us to follow his/her advice in this matter. Thus, we have removed the mentions to the multivariate analysis all along the manuscript.

We copy below his/her comment so that the Reviewer understands why Rev. #1 recommended to remove this analysis.

“Line 352: “A patient with high levels of both biomarkers had, according to the logistic regression analysis, a 16-fold higher chance of achieving a pCR in response to paclitaxel-based chemotherapy when compared to other patients. Although this study is retrospective, ...”. Presumably this is meant to say “Patients with high levels of both biomarkers ...” since one cannot perform regression on a single patient. There are no details about the logistic regression model that was fit (sample size, covariates, errors, odds ratio estimates) and it seems to have been fit on a marker that was already deemed to be significantly association with the response. I suggest removing this “retrospective” regression analysis”

Query #4. Insufficient information is provided about the 2 validation tissue sets, i.e., patient and tumor characteristics, preoperative treatments, definition of response to therapy, etc.

According to this question, and another question from Reviewer #3 (Query #6), it seems that Supplementary Tables 3 and 4 were not merged in the PDF for external review in the previous version. Those Tables reported all the characteristics about the 2 validation sets. We will make sure that this time both tables are correctly merged in the manuscript.

The definition of response was not added in the previous version, but we have added it now. In both validation sets, we consider “response” achieving a pCR in response to neoadjuvant chemotherapy (no residual tumor in breast and axilla, or , at most, residual *in situ* carcinoma in the breast). This definition is included in the Patients and Methods Section, “Patients and tumor samples” sub-section. The definition applies both for the training and validation sets, and complements the information answered to your Query #1.

Query #5. Sometimes difficult to follow and understand the data in Results because not consistently and clearly stated which biopsy results are being described, i.e., responder vs non-responder; pre- vs post- Nintedanib. Also sometimes not clear if specific proteins were over-expressed or up-or-down regulated in parts of the manuscript.

We thank the Reviewer for this constructive comment. We have modified partially the manuscript and the narrative, aiming to improve clarity:

-First, following this Query, and others from Refs. #1 and #3, we have decided to eliminate the manuscript sections/figures related to nintedanib, so that the main message is not mixed with a non-relevant one. In the clinical trial where the “training set” samples were obtained³, several hypotheses were tested: whether a “priming” monotherapy course of antiangiogenic would improve paclitaxel response rate, whether the combination was better than the monotherapy arm, the role of a molecular imaging probe (18F fluoromisonidazole) in monitoring vascular normalization, and also predictive factors for paclitaxel and/or nintedanib. Back in the day when this trial was designed (2011), antiangiogenics were still a “hot topic” in breast cancer, and nintedanib seemed like a potentially very good drug candidate (at that moment, it had shown positive results in ovarian carcinoma and lung cancer). In parallel, we published a manuscript reporting

on acquired mechanisms of resistance against antiangiogenics that were proficient in causing vascular normalization, of which nintedanib was a paradigmatic drug. However, time passed and antiangiogenics were basically withdrawn from all indications in advanced breast cancer. Immunotherapies came, and research in antiangiogenics in breast cancer was mostly abandoned. Although our proteomic data suggested that the preclinical results about acquired resistance to nintedanib were validated in this patient series, the fact is that such results are no longer relevant – this agent never gained approval and never will, neither in breast cancer nor in other tumor types. Since we did not even validate those results in external patient sets, or experimentally, akin what was done with paclitaxel, the data add nothing to the main point of the manuscript, which focuses on a key drug in the management of breast cancer: paclitaxel.

Clearly, reporting on predicting factors for an antiangiogenic and a chemotherapeutic drug should be subject of two different manuscripts, since they are unrelated topics. However, since we did not follow-on the nintedanib data, when we prepared the first version of the current manuscript, and such data probably did not support a stand-alone manuscript, we merged both datasets on it and waited for editorial feedback. Now, with the Reviewers' and editorial feedback it is clear to us that nintedanib data add nothing but confusion to the manuscript and thus we have decided to remove them.

-Second: we have gone through the whole manuscript trying to clarify the narrative about which samples were being tested and which kinases or proteins were up- or down-regulated. The second sub-section (*“Kinases and phosphopeptides enrichment among responders or non-responders to paclitaxel”*) has been re-written. The third sub-section (*“Elevated CDK4 and Filamin A levels narrow down a subset of patients with increased sensitivity to paclitaxel”*) has been significantly edited as well.

-Third, in the Intro section (in the last paragraph), we have added a brief overall description of what was done in this study, in order to guide the reader through the steps taken along this research, since we acknowledge that this is a complex topic.

We hope that the Reviewer now finds the manuscript more focused and clearer.

Query #6. In patients who had residual disease post pre-op paclitaxel sufficient for mass spectra analysis, was CDK4 and Filamin A expression analyzed?

This is a very interesting question that can open research lines regarding potential mechanisms of acquired resistance against paclitaxel.

First of all, we would like to clarify that proteomics was only done in the pre-treatments samples from the discovery-set trial. This was a multi-centric trial and the study, consent form, and ethics approval only considered sending to our group the samples from the pre-treatment biopsies (and the post- 2-week nintedanib). All the post-treatment (surgical) specimens remained at the investigation sites; although the samples were sent to a central lab for assessing RCB, tissue left-overs stayed at the investigation sites. Sample exhaustion for proteomics was not allowed by the ethics board, since the surgical samples might be important for further clinical decisions, and thus were kept and not sent to our lab for obtaining more mass spectra.

However, we have post-treatment sample slides from the two external sets in which we confirmed the candidate biomarkers. This tissue is not sufficient for mass spectrometry, and its paraffinized nature would not be the best vehicle for proteomics (all the discovery set samples were snap-frozen samples, the best vehicle for proteomics). But since our aim was to define biomarkers that could be determined in daily routine – i.e. ideally by IHC – we have sufficient tissue leftovers to determine also CDK4 and Filamin A in the post-treatment samples.

We have compared by IHC the pre- and the post-neoadjuvant treatment CDK4 and Filamin A staining levels. Obviously, since all the samples that achieved pCR had, on average, higher levels of CDK4 and Filamin A than those samples that did not achieve pCR, we have removed from the pre-post comparison the samples from patients with pCR – not doing so would constitute an artifact biasing towards a decrease in the post-treatment samples, since all those with high CDK4 and Filamin A would disappear and

don't have a paired post-treatment samples. Thus, we compared the evolution of CDK4 and Filamin A during the neoadjuvant treatment in the patients that did not achieve pCR. The following figures illustrate the changes

-Figure 1: Pre- post- samples in three subtypes samples (Set 1) – CDK4

-Figure 2: pre- post samples in three subtype samples (Set 1) – Filamin A

-Figure 3: Pre- post samples in TNBC only (Set 2) – CDK4

-Figure 4: pre- post samples in TNBC only (Set 2) – Filamin A.

As the Reviewer can appreciate, it seems that during the exposure to neoadjuvant chemotherapy, resistant tumors tend to down-regulate CDK4. Such effect can be appreciated in both cohorts. Regarding Filamin A, no clear conclusions can be extracted: in the first validation cohort, Filamin A levels do not experience significant changes, while in the TNBC cohort, it seems that Filamin A levels tend to increase. Down-regulating CDK4 could be a potential mechanism of resistance to avoid the effect of cytotoxic drugs, since, if the cell cycle is stalled, the action of cytotoxics (which is cell-cycle dependent) does not take place¹⁵; however, both this conclusion, and studying why some tumors regulate Filamin A in response to paclitaxel, would require an entire new research project.

We are not sure of whether and where to place these results in the manuscript. We believe that since our manuscript verses on predictive factor for response, and mechanisms of sensitization to it, opening a new research story about acquired resistance would be somehow unrelated to the main focus. We request Reviewer's and Editorial advice of whether this new line of research and results should be included (and if so, where, since we do not really find a clear place to start talking about acquired resistance) in the main manuscript or supplemental figures.

Comments from Reviewer #3. Expertise in paclitaxel mechanism of action

In this manuscript, "Phosphoproteomic analysis of neoadjuvant breast cancer reveals a cluster of patients with increased sensitivity to paclitaxel driven by CDK4 and Filamin A", Mouron et al present phosphoproteomic analyses of breast cancer patient tumor data sets from neoadjuvant trials. Identifying markers of chemotherapy sensitivity is important as it could spare non-responders the toxicities associated with ineffective therapies. Specifically with their methodology they discovered several candidate markers of Paclitaxel sensitivity of patient cancers. They also present some functional insight into the mechanisms responsible for regulation of Paclitaxel sensitivity by these candidate molecules. Overall, it is an intriguing, well done study but several questions need to be addressed

Query #1. The manuscript as presented is somewhat disjointed with the first half focused on nintedanib sensitivity (with unrevealing results) then transitioning to paclitaxel sensitivity. Why not focus the manuscript on Paclitaxel sensitivity with more validation sets?

We thank the Reviewer for this suggestion, which was also made by Reviewers #1 and #2. Because of this unanimous request, we have removed all the manuscript sections related with nintedanib.

We actually agree with the Reviewers, although there is a reason why we included those data in the first version: back in the day when the trial from which the samples for the training set were extracted, antiangiogenics were still a "hot topic" in breast cancer. In fact, we started in parallel the trial and also preclinical work about acquired resistance to nintedanib, which was published in 2016. However, around the same period immunotherapies gained traction in non-melanoma tumors, while it started becoming

clear that antiangiogenics added little, if anything, in breast cancer. Thus, we focused our attention in using the samples for understanding the mechanisms of sensitivity to paclitaxel, since we had a very “clean” patient cohort. Regardless, we run all the samples, including the post-nintedanib ones, and found that, at least in the KSEAs, data seemed to confirm what we found in the preclinical data (up-regulation of PKA and AMPK as a potential mechanism of acquired resistance). Since the story was never followed, such proteomic data would never warrant a stand-alone manuscript, both due to the lack of completion and due to the lack of interest in this drug class now in breast cancer. Thus, we opted for including the data and seeking for editorial advice. Now, it is clear to us that both the editorial office and the Reviewers share our same opinion and these data add nothing to the main manuscript but distraction from the main focus.

The Introduction, the second sub-section of the Results section (now “*Kinases and phosphopeptides enrichment among responders or non-responders to paclitaxel*”), and the discussion section have been re-written to eliminate the text related to antiangiogenics and/or results related to nintedanib samples. In addition, Figures 2 and 3 have been shortened, since several panels belonged to nintedanib results. A brief explanation of the whole study is provided as well at the end of the Intro section in order to “guide” the average reader.

We hope that the Reviewer finds now the manuscript more focused, clearer and sharper.

Regarding potential addition of validation sets – we think that, for now, with two external validation sets and preclinical experimentation, together with the data that we have added in response to your 7th Query (i.e., whether the mechanism is specific for paclitaxel or just a non-specific mechanism for cytotoxics; the answer is that it is, indeed, specific), we have proved reasonably well that these two biomarkers are related with sensitivity to paclitaxel therapy (in fact, in the discovery set, patients were treated only with paclitaxel). It is true that now external validations should follow, akin with other biomarkers. The moment at which these biomarkers are strong candidates to incorporate them in the

clinical decision tree, we do not really believe in adding further small validation sets – we think that the real value comes from launching a “definitive” merely clinical study.

We will add our contribution in this sense too: in collaboration with the Spanish GEICAM group, we will explore the role of these two biomarkers in the overall survival impact in two important studies: GEICAM 9805 and GEICAM 9906. These two studies were adjuvant studies that compared FAC versus TAC (docetaxel instead of fluorouracil) and FEC versus FEC followed by 8 weeks of paclitaxel, in early breast cancer^{16, 17}. Together, both studies add up more than 2500 patients, and >15 years of median follow-up. In this translational study we aim to confirm their role as biomarkers, in the definitive outcomes: disease-free survival and overall survival, both un-adjusted, and adjusted by breast cancer subtype, and drug (paclitaxel versus docetaxel). Such results would aim for registration of these biomarkers as tests to be included in future guidelines for deciding escalation or de-escalation of treatment in early breast cancer, in a cheaper manner than current genomic platforms. However, we think that a study of this type should be reported as an independent manuscript, and not as an ancillary figure to the current one. This study is currently in its preliminary steps (Ethics board, sample gathering, statistical design, etc); it may take approximately 18 months from now. We hope that the Reviewer agrees with our opinion.

Query #2. For the methodology of the phosphoproteomic analysis, it was not clear whether there was any normalization to total levels of a candidate molecule.

This is a relevant point and we thank the Reviewer to give us the opportunity to clarify it. Our intention in the discovery set is only to find as many potential candidate markers as possible. That is the reason why for the KSEA we apply a relax FDR boundary (0.2), and why we do not apply any normalization strategy. Other manuscripts report (phospho)-proteomic findings in large sample cohorts, and all the conclusions that those

manuscripts reach are based on the -omic techniques, such as those from the CPTAC consortium^{4, 5, 6, 7, 8, 9, 10, 11, 12}. In such a case, it is mandatory to apply highly stringent filters; otherwise, the conclusions would be weak (and in our opinion, although techniques such as gene-expression or NGS probably do not require further external validation due its robustness, even state-of-the-art proteomic findings require external validation with other techniques and in other sample sets, due to the stochastic nature of the technique). However, our focus is different – in the discovery set we look for candidate markers, and then, stringent filters are applied in external validation sets; plus, experimental validation of the hits is provided to add robustness (since “biological plausibility” is one of the most important causality criteria). Once a given “hit” is validated in several external sets, and experimental work demonstrate its mechanism, the origin of the candidate hit, if the Reviewer allows us to use this expression, does not really matter: it can be a candidate marker from the literature, a candidate marker from a -omic experiment (performed with more or less stringent filters depending on whether the researchers want to do more or less downstream validation work), or just a candidate coming from intuitive thought. As mentioned, since we have experienced the limitations of mass spectrometry to find robust markers as a standalone technique in the past^{18, 19}, we have opted for applying as less stringent filters in this first step as possible.

Why not applying a normalization by the total amount of protein for each identified phosphopeptide? Just for practical reasons: gene expression (and for the matter, protein levels), for example, display several-fold regulation from normal tissue to cancer tissue or from one tumor to another (although there are cases of extreme differences). Similarly, VAF of mutations of interest may vary 20, 30, or 50% from one tumor to another, but again, there is a limit to the variation. However, phosphorylation events, at least in our hands, vary from nothing to literally billions (spectral intensity) – several orders of magnitude. If we were comparing just a few samples or a few experimental models, it might be important to compare as well the levels of the native proteins and not just the phospho-fractions, as long as phosphoproteomics was a “finalist” technique (i.e., not a

discovery technique). However, when a large number of samples are compared, the potential differences in native protein levels evens out among groups, although the phospho-levels can vary radically. KSEAs detect differences in enrichment in kinases activities among sets of samples regardless of normalization, if the sample number is large. When the sample set is small, either a very large difference between groups or strict normalization techniques are required.

Thus, the reason for not running an extra set of samples (running the phospho and the total protein fraction requires 2X runs, material and costs) is merely practical: in the past, we have successfully found and validated biomarkers without normalization in the discovery set; the strategy was accepted by peer reviewers and published¹⁹. Similarly, we discovered the kinases implicated in the adaptation to hypoxia-correcting antiangiogenics relying in the same technique, which were subsequently validated in independent experiments¹⁸; also, this strategy was accepted by peers and published. Here we are applying the same strategy and we actually support it as a valid and practical one in order to translate markers to the clinical setting (for a “pure-proteomics” work it may be a different story).

In a similar line of thought – what would have happened in case a normalization by total protein strategy was adopted? We can think of three scenarios:

-In the first one, we find the same candidate biomarkers. Still, we would have needed to proceed to confirmation in external series and provide experimental validation. Thus, same results, but twice as much work, expenses and sample exhaustion.

-In the second one: applying more strict boundaries and normalization, some of the candidate markers in the discovery set are discarded. Then, the work ends there. Does that invalidate our markers? We do not think so, since, regardless of which is the origin of the candidate biomarkers (literature, -omics, or simply previous knowledge), we show their role in external series with other techniques and provide experimental validation.

-In the last scenario, normalization strategies make other markers emerge, that have been overlooked with our strategy (unlikely, since that would imply that in those cases the regulation would not be in the phosphorylation of the marker but in the total levels, making the ratio phospho/non-phospho “pop-up” in one or another group). That is fine – future researchers may find other mechanisms and markers of sensitivity to paclitaxel. The “perfect” study (in this case, the one that finds all the markers, discriminating 100% of the patients, and explains all the sensitivity and resistance mechanisms at the cellular and sub-cellular levels) never exists, and the projects have to be wrapped-up at some point - again, that would not invalidate our proposed markers.

We have added discussion about this point in the study limitations paragraph in the Discussion section (second-last paragraph). However, we would like to clarify that we would see this as a limitation for biomarker definition if the conclusions were supported just by proteomics techniques. In a study like this one, aiming to define biomarkers for the clinical setting, that can be determined with a routine technique available at hospitals (and mass spectrometry is not one of those), our position is that the discovery set is as “relaxed” as possible so that many candidate biomarkers are available for the downstream validation experiments. Then, only the validation sets and experimental validation can confirm or reject those candidate biomarkers – we think that those are the real strengths of our work

Query #3. For all patient tumors, unlike cell lines, there is a significant degree of heterogeneity in levels of staining for any marker. Did the authors/pathologists observe heterogeneity in staining for CDK4 or Filamin in patient tumor sections, and if so, how was the overall intensity of staining determined (leading to classification into quartiles). Was the pattern of staining distribution relevant at all?.

We hope to understand well the question – it seems that the Reviewer asks about how the overall intensity led to classify each patient into one or another quartile. However, the degree of staining heterogeneity is also mentioned. Since they are not exactly the same thing, we will answer to both, hoping to answer any potential doubt:

-First, it was checked whether the staining patterns were heterogeneous in two ways:

A) whether, for each marker, there was changes in the staining distribution in tumor cells and/or stroma from patient to patient, in order to evaluate if this led to predictive different

B) whether, for each marker, there was differences in terms of nuclear or cytoplasmic staining, to ascertain if this could also imply differences in response to paclitaxel.

An expert human pathologist (author E.C.) reviewed the scanned images and, first of all, he mentioned that the task of addressing heterogeneity in this tumor material is somehow limited by the fact that the samples were mounted in a TMA – thus, the total area evaluated by the pathologist was relatively small in all cases.

Regarding Filamin A, the staining signal was always cytoplasmic in tumor cells. The staining was very low in the stroma and was not deemed as a potential source of variability (not a single patient had a meaningful staining “by eye”). CDK4, conversely, was distributed along the nucleus and cytoplasmic in all cases. CDK4 was also very low and homogeneous across patients in stroma. Thus, stroma was not quantified; only tumor areas were quantified for CDK4 and Filamin A.

We have added a Supplementary Figure (new S6) that shows tumor/stromal and nuclear/cytoplasmic staining patterns for both markers. We did not proceed to further divide patient sets in “predominantly-nuclear CDK” or “predominantly-cytoplasmic CDK” in order to avoid “data torturing”. Maybe in the future, with larger datasets, where only CDK4 is stained, patient subgroups can be established in this basis.

-Second, since the only source of significant patient-to-patient variability was cytoplasmic staining (for Filamin A) and cellular staining (indistinctly distributed along nuclear and

cytoplasm, but no cases were found of “only nuclear” or “only cytoplasmic” staining), a computer-aided H-score was calculated for each marker in order to assess overall staining intensity, what yields a continuous quantitative variable from 0 to 3 (0 : no staining; 3: maximum staining). For each marker, according to the H-score distribution in each of the 2 validation sets (from 0 to whichever was the maximum H-score), the 3 quartiles were found. The cut-off values determining the top-quartile for each marker in each of the 2 sets are shown in Supplementary Tables 5 (Set 1) and 6 (Set 2). According to the H-score of each patient for each marker, she was classified above or below the top-quartile staining. Since the stainings were done in different days, by different technicians, and with samples originated in different hospitals, the H-score distribution for any given marker in one sample set can be slightly different from that in another sample set. The digitalized acquisition and automatic quantitation methods are described in the last paragraph of the “Antibody setup and immunohistochemistry staining” sub-section of the Methods section. The classification of patients in high- or low-quartiles is also described in the Statistics sub-section.

We hope to have clarified the potential sources of staining heterogeneity, and, more importantly, how patients were classified in quartiles.

Query #4. In Figure 6D, the association between Filamin A and CLIP7 diminishes with overexpression of CDK4 or Filamin A. This seems to contradict the text and needs to be explained.

The Reviewer is right regarding how the figure is not very representative. However, co-immuno-precipitation is not, strictly speaking, a quantitative technique; the quantitative information about the amount of CLIP170 binding to Filamin or Tubulin mentioned in the text (higher in MDA-MB-CDK4 cells) was extracted from the mass spectrometry data shown in Figure 6C (extended data in Tables S7 and S8). Regardless, we have included

a new version of Figure 6D that we hope the readers will find more congruent with the narrative.

Since the binding between CLIP170 and Filamin had not been previously described, the intention of showing Figure 6D was to prove that point – that CLIP170 and Filamin can bind *de facto*; the “metrics” are provided with the experiment shown in Figure 6C, Table S7 and S8. Tubulin and CLIP170 were already known to bind each other²⁰.

Query #5. In Fig 7A, the signals on the Western blot need to be quantified as higher levels of acetylated tubulin on CDK4 or Filamin A overexpressing cells is difficult to appreciate.

Thank you for the suggestion – we have added in the right-hand side of the panel a bar chart with the quantification of three experimental replicas. The figure legend has been modified accordingly, and the WB quantification methods have been added in the Methods Section.

Query #6. Table S3 (patient characteristics for the data set) seems to be missing? It is not common for ER+ cancers to have complete pathological complete response and it is more common for patients with Her2+ disease to undergo simultaneous neoadjuvant treatment with chemotherapy+Her2 targeted therapy. Knowing the baseline patient and tumor characteristics is important.

According to this question, and another question from Reviewer #2 (Query #6), it seems that Supplementary Tables 3 and 4 were not merged in the PDF for external review in the previous version. Those Tables reported all the characteristics about the 2 validation sets. We will make sure that this time both tables are correctly included in the manuscript. We apologize for overlooking this in the previous version.

Query #7. How specific is CDK4 and Filamin as markers of sensitivity for Paclitaxel? Are they general markers for response of proliferative tumors to many types of chemotherapy?

Although the candidate biomarkers came from a discovery set where patients received single-agent paclitaxel, the question is very important, since the external confirmation sets were constituted by patients that received standard poly-chemotherapy regimens. As the Reviewer adequately points out, highly proliferative tumors have been associated to non-specific sensitization to cytotoxics in general.

We tested whether MDA-MB-231-CDK4 or -Filamin A cells were differentially sensitized to other cytotoxics or only to paclitaxel. Patients from the validation sets received the other two key agents used in TNBC: anthracyclines and platins. Thus, we checked whether overexpression of CDK4 and Filamin A sensitized cells to either of those drugs. The IC50 for cisplatin or adriamycin remained stable regardless of overexpression of CDK4 or Filamin A, and were not statistically significantly different, as opposed to the case of paclitaxel, where each overexpression decreased the IC50 by 3-fold. These results are now shown in Figure S9.

Although the activity of CDK4 is required for the progression of the cell cycle, correlations with Ki67 are not linear. Such correlation has been commonly reported in hormone-positive breast cancer; however, TNBC has high levels of Ki67 in most cases and the correlation with CDK4 levels is less studied. In our case, we studied the correlation between CDK4 and Filamin A staining levels and we found the following Pearson's coefficients: CDK4 and Ki67: $R=0.27$; $P=0.010$ (low positive correlation, statistically significant); Filamin A and Ki67: $R=0.11$; $P=0.37$ (no correlation, non-significant).

Given the following facts: 1) CDK4 and Filamin A were selected as candidate biomarkers in a patient group that received single-agent paclitaxel; 2) CDK4 and Filamin A

overexpression sensitize cells to paclitaxel, but not anthracyclines/platins (the drugs that also received patients in the validation sets); 3) mechanistic experiments suggest specific accumulation of paclitaxel in the microtubules elicited by the Filamin A-mediated CLIP170 binding; and 4) CDK4 and Filamin A showed only low and no correlation respectively with Ki67 in the TNBC series, but both are strongly implicated in the sensitization mechanism, we believe that we can conclude that the sensitization is specific, and not the result of non-specific sensitization because of high cell replication. We have added the colony assays and the CDK4/Filamin A – Ki67 correlation charts in Figure S9, and added text in the results and discussion section in this regard. This observation noted by the Reviewer adds a great value to the manuscript.

Comments from Reviewer #4. Expert in proteomics

In this manuscript by Mouron S et al., entitled “Phosphoproteomic analysis of neoadjuvant breast cancer reveals a cluster of patients with increased sensitivity to paclitaxel driven by CDK4 and Filamin A”, authors aimed to define a set of biomarkers for a cluster of patients with breast cancer and propose high levels of CDK4 and Filamin-A, linked to a 90% chance of achieving a pCR in response to paclitaxel. However, I have major concern with the most important experiment used to build the hypothesis, detection of phospho peptides was used to infer the involvement of specific kinase and other target proteins, however, no direct measure of these proteins/phosphoprotein using mass spectrometry was carried out to confirm the dysregulation of these kinase and/or other target proteins in the patient’s tissue samples.

With additional suggested experiments and revisions this manuscript may be considered for publication in Nature Communications but in its current state the manuscript is premature and not ready.

Major comments:

Query #1. In Figure 1B, please clarify what data was used for clustering analysis, in figure legend it says “phosphoproteomic spectra” and in text it says, “on the basis of the nature and abundance of phosphopeptides found on each sample”. Please clarify what does “nature” of phosphopeptides means here?

We thank the Reviewer for detecting this inconsistency, which is an example of not clear narrative. We apologize for the mistake. After consulting with an English-native computer scientist from our institution, he has allowed us to clarify what we really meant: the clustering was done with the phosphopeptides intensity data. By “abundance”, we meant intensity; by “nature”, we meant the ID of each peptide.

We have re-written the text in both locations. The new text is “Phospho-peptides intensity data matrix was used for clustering analysis”.

Query #2. How was the peptide abundance normalized?

We thank the Reviewer for bringing up this point, since it is worth discussion and actually an important difference between “pure” proteomic studies and translational studies. We have added now text in the discussion section (detailed below) since agreement between researchers in this matter will have to be reached in the future.

First of all, as the Reviewer knows, proteomics is not a technique that is widely used in the clinical setting (i.e., as opposed to gene expression or gene sequencing; because these techniques are easy to perform, consume little material, are robust – i.e., not stochastic detection – , inexpensive,... among other advantages for clinical application). Gene expression or gene sequencing have led to defining clean-cut, binomial biomarkers for clinical implementation (that is, “yes/no” markers that are easy to

determine and, to some extent, unequivocal; a paradigmatic example can be the detection of EGFR mutation in lung cancer for allocating a patient to an EGFR inhibitor, or the Oncotype score in breast cancer for deciding chemotherapy versus hormones in early breast cancer). Proteomics has not reached this point yet, and thus, many existing studies are “far” from the clinical application sphere. Few proteomics studies are in areas closer to the clinical setting, and thus, beyond the discovery steps, in these studies validation steps follow, with different approaches that could A) prove that the biomarker is actually a biomarker in the clinical setting in independent datasets, and B) somehow determine the biomarker in the clinical setting with a different, more accessible technique.

Well-known recent studies are those produced by the CPTAC consortium^{4, 5, 6, 7, 8, 9, 10, 11, 12}. These studies constitute huge efforts and uncover novel biological insights of different tumor types (breast, lung, ovarian,...), of how genomic aberrations “re-wire” signaling pathways, and how proteo-genomic clusters can relate to sensitivity or resistance (or define biologic subtypes within a tumor type) to one or another treatment. However, in all cases, these studies lack independent validation in clinical series where the candidate biomarkers are confirmed (with proteomics or another techniques), or functional studies or clinical efficacy studies confirm sensitivity or resistance to the proposed agents. Results, and conclusions, are descriptive. This type of studies is similar to the initial TCGA studies, which provided “encyclopedic” knowledge about potential driver mutations, novel targets, or novel disease subtypes; clinical trials followed proving one or another concept. We hope that soon proteomics technologies can penetrate the clinical setting in a similar way genomics did, and similar validation studies can implement for good proteomics in the clinical decision tree. However, this is not yet the case, and, currently, a clear framework of how potential proteomics biomarkers can be incorporated in the clinical setting is still lacking, as opposed to genomics or gene-expression markers.

In this sense, the way researchers try to discover/validate biomarkers in this field, is still creative and designed the best way the researchers can think of, since walking the path “from proteomics to the clinics” is still quite complicated.

If a study bases its conclusions solely in proteomics studies, we understand that proteomic scientists agree in how the experiments have to be performed and samples processed; procedures have to be state-of-the-art in order to draw any conclusion. That is the case, for example, of the CPTAC studies. But, from the clinical point of view, still, such conclusions would not be worth of much clinical use unless they are validated, ideally, with a clinically-available technique. What we aim to convey with this statement is that, in the oncology biomarkers field, it is not highly relevant whether the candidate biomarkers come from existing literature, plain hypothesis, serendipitous results of experiments of different kinds, state-of-the-art discovery experiments, or just previous knowledge. What it is really important is that the candidate biomarkers are validated in independent datasets (if possible, with techniques accessible in the clinical setting). Additionally, mechanistic insights can help establishing unequivocal relationships between the biomarker and the outcome (as opposed to spurious associations, such as the known example of “yellow-stained finger as a cause of lung cancer”). This is what we have done: define a set of potential biomarkers, validate them, and confirm the specific nature of the biomarker with extensive experimental work. In this type of studies, if the source of candidate biomarkers is a discovery experiment (as opposed to literature search, serendipitous findings,...etc), it is up to the researchers to design the discovery experiment with such a set of characteristics and detection boundaries that ensures a larger or smaller set of candidates with higher or lower chance of validation. That will determine the amount of work that has to be done in the validation steps, and the chance of success of it, but at the end of the day, is a decision taken by the investigators.

Our previous experience^{18, 19} tells us that most markers lose significance once in the validation sets or validation experiments – thus, we applied relaxed statistical boundaries and detection techniques in order to have as many biomarker candidates as possible.

Since phosphorylation events, as opposed to for example gene-expression events, experience thousands or millionfold (or even higher) regulation among different samples, we aimed to capture as much of these differences as possible, **not normalizing by total levels of proteins in purpose**. Given the clinical nature of these samples, even if we wanted to do it now, it would not be possible since the samples are now exhausted (as the Reviewer knows, the amount of tissue required for obtaining good phospho-readouts is considerably high); however, in the past, we did not do it because of two reasons: first, in order to maximize the capture of phosphorylation differences; and second, in order to decrease the cost and duration of the experiments by half. The second reason is quite obvious, but the first may not be so – thus, we provide further explanation: even after normalization by total native protein, the main source of differential regulation in a phospho-event from one sample to another is the phosphorylation itself; i.e., an event that is detected with 10^6 -fold regulation in one sample compared to another is more likely to be caused by a 10^5 - 10^6 -fold regulation in the phosphorylation, not in the expression levels in the native protein (which may vary 1-20 fold, but commonly not more). Thus, whether the true regulation value is 10^6 or 10^5 is not highly relevant unless we want to specifically conclude how many times is an event regulated in one sample versus another (i.e., a “pure proteomics” experiment). Here, given the number of phospho-events, and the number of samples, at the end of the day what we want to pinpoint is, from a qualitative point of view (not quantitative) which are the potential kinases implicated in the enrichment of the p-sites observed in one group (i.e, “responders”) versus another (i.e., “non-responders”). P-sites that only emerge or are only cleared out after normalization, with the size of the data matrix, would probably be regulated in insufficient magnitude to be preserved as biomarkers in subsequent validation steps. It would be a different story if the aim was to compare among two cell lines (i.e., very few samples) the degree of regulation of a specific event (i.e., the conclusion wants to be delivered out of a proteomics experiment in a quantitative manner).

The real point of interest in our work is, after a discovery step, which biomarkers actually hold after astringent and state-of-the-art validation filters. We not only did that, but also did so translating the biomarkers to a technique that could be used in the clinics right away, and provided experimental proof of the mechanistic specificity of the biomarkers. Having reached this point, it is not relevant whether the candidate biomarkers came from literature, previous knowledge, or a state-of-the-art proteomics experiment; what gives validity to a biomarker is not its discovery but its validation process. If a candidate biomarker comes from “bad” literature or experiments, validation will discard it regardless.

Nevertheless, we want to illustrate which would be the three potential scenarios if the total protein levels had been used to normalize the p-peptides:

-In the first scenario, normalization does not change anything, and the same candidate biomarkers “pass” the first filter (discovery): given the aims of our study, we still would have had to proceed to external astringent validation and experimental validation (globally, no change).

-In the second scenario, normalization makes that none of the candidate markers are found in the discovery experiment. This would have ended the study here. How does that affect to our study and conclusions? In our opinion, in no way: whichever was the origin of CDK4 and Filamin A as candidate biomarkers, our study proves that they are, in fact, biomarkers that sensitize triple-negative breast cancer to paclitaxel. In fact, what the results in Set 1, Set 2, and Figures 5-7 do, among other things, is proving that the proteomic experiments were not so unrevealing (because some of the markers withstood the validation steps). So, globally speaking, in this scenario, normalization would have hampered the possibility of finding these 2 new biomarkers.

-In a third scenario, normalization suggests additional candidate biomarkers. That would be great, and we hope that more studies are conducted until this one and other biomarker problems are solved completely – not a single work resolves a pathologic situation in 100% of its aspects. Yet, that would not have changed the conclusions of our study.

Taken together, we hope the Reviewer understands that we are not reporting a proteomics study, but a clinical oncology biomarker study, and that in this scenario, the discovery-validation framework is somehow less clear than in other fields. In fact, we are pioneering this field and we are not aware of many other groups that are walking the “proteomics to the clinics” transition. In this sense, our previous studies were performed with a similar approach and were understood and accepted by peer-reviewers and well-published^{18, 19}; somehow, this should constitute an, at least, acceptable landmark.

The situation would have been different if we were aiming to just issue our conclusions out of the proteomics experiments, and we hope that the Reviewer shares our opinion. Using proteomics as a discovery step can be as accurate or as faulty as any other -omic experiments; what we are trying to say is not that our technique was not good, but that, “puristic” or not, results from a discovery set within this study context are only worth something if they are subsequently validated. From a patient perspective, a study with “puristic” discovery steps with no further validation is less useful than a “less puristic” study with full validation

We have added the following text in the discussion section:

“... In those studies, conclusions are based in state-of-the-art multi-omic experiments. In ours, proteomic screening was just a tool for discovering potential biomarkers mostly based in KSEAs emerging out of different phospho-profiles, with the aim of defining specific predictive factors for a single drug (paclitaxel). Relaxed statistical boundaries were applied in the initial study steps in order to maximize the number of biomarker candidates; the strength of our study is the combination of astringent external validation in external well-annotated patient series with mechanistic experiments showing biomarker specificity...”

And below:

“.... The second limitation also concerns the discovery experiment: the lack of normalization of phosphopeptide intensities by total protein intensity could be perceived as a shortcoming. Although the native protein can experience regulation from one sample to another, this is always of much smaller magnitude than the changes in phosphorylation. Thus, it is unlikely that many of the detected events with 10^5 - 10^7 -fold regulation had not been detected if normalization techniques were used. Regardless, our strategy was to maximize the number of candidate biomarkers found in the proteomics experiment, applying low FDR boundaries for the KSEAs and, deliberately, not normalizing phosphoprotein intensity by total protein intensity (since this would have duplicated the cost and duration of the discovery step): no matter how reliable these candidate markers would have been because of the boundaries set in the discovery experiment, they still would have had to undergo confirmation in external sets and experimental mechanistic validation. Studies that base their conclusions exclusively in proteomic techniques (or other -omic) obviously require more astringent boundaries and different experimental processes^{5, 7, 8, 9, 21}. However, in a clinical oncology biomarker study, what matters most is whether the biomarkers can be confirmed and actually hold a mechanistic relationship with the feature under study regardless of the origin of the candidate markers (literature, serendipitous experimental results, or discovery experiments, just to name a few), which we believe to have achieved. We have successfully applied the same strategy in the past^{18, 19} and we think that it can be safely stated that this is an acceptable discovery biomarker strategy in clinical proteomics, particularly for difficult-to-solve translational oncology problems....”

Also, in the introduction section, we have added a short text in the last paragraph explaining what was the workflow of this study, and clarifying that proteomics was just a discovery tool.

Finally, we take the liberty of pasting here the introductory comments and Query#1 from Reviewer #1, an expert in biostatistics, who is probably well familiarized with this type of studies (biomarker validation), that illustrates well the overall orientation of our study, which, as we say, it is not a proteomics study:

***“Summary of statistical analysis steps:** Using biopsies from a subset of 130 breast cancer patients in a previous clinical trial, 15 candidate biomarkers were identified using Kinase Set Enrichment Analysis, which is a variant of GSEA but applied to peptides and kinases instead of gene sets. In a subsequent cohort of 117 early breast cancer patients of 3 subtypes all treated with paclitaxel, these 15 markers underwent investigation with IHC and 5 were found to be associated with response to paclitaxel therapy. And in subsequent cohort of 101 triple- negative patients all treated with paclitaxel, 2 of these 5 biomarkers exhibited associations with pCR. The remainder of the manuscript focuses on investigations into mechanistic aspects of these 2 biomarkers and the role they may play in response to treatment.*

***Query #1: Comments on the data:** The primary results are the consequence of multiple stages of analysis in separate datasets/cohorts used to narrow down the field of candidate markers. The initial step identified 15 candidate biomarkers using samples from a clinical trial [21] in which 130 patients were randomized to treatment on nintedanib-plus-paclitaxel versus paclitaxel-only treatment; the original trial aimed at testing the treatment effect on the endpoint of residual cancer burden. The current analysis, however, is focused on 85 (“valid”) biopsy samples taken at baseline in that trial, and another 45 (“valid”) samples taken from a subset of these 85 patients in the experimental arm following two weeks of Nintedanib treatment (Figure 1 and line 179). Since these data are exclusively from the experimental arm, the fact that they come from a randomized clinical trial is incidental to this research. The authors, however, focus on that randomization: “we conducted the largest phosphoproteomic screening in samples from a randomized clinical trial to date” (Line 149), and “Taking advantage of the randomized trial design, and the quantitative treatment outcome (RCB), KSEA allowed finding the following potential associations” (line 209). Even if the two-arm randomization comes into play in the analysis, it would be non-standard to use, for the purpose of biomarker discovery, a cohort that was randomized to treatment, and then use subsequent single-arm treatment cohorts for confirmation. As used here, the clinical-trial randomized design is not related to the findings. Although there is nothing wrong with what was*

done, it is misleading to describe the findings as if they are in any way the result of a large randomized clinical trial”

And later:

“... However, these cohorts were used in subsequent filtering steps, follow-up confirmation, and continued investigation of the biomarkers in different ways..”

Query #3. Also, please write the number of patient (n=?) shown in dendrogram.

We thank the Reviewer for the suggestion. The number of samples is now depicted in the dendrogram, and hope that now the readers find the information more easily than in the previous version.

Query #4. For 1352 distinct phospho proteins identified, how many minimum numbers of total and unique peptides were detected? Please provide the numbers of peptides and proteins detected in each condition.

First of all, it is worth mentioning that Reviewers #1, #2 and #3 recommended eliminating the data related to nintedanib and focus only on paclitaxel. Thus, results concerning the changes in the proteome from pre- to post-nintedanib samples have been removed.

Following the Reviewer's recommendation, we have also expanded Table S1, including now the PSMs, MS/MS, phospho-PSMs, phosphoproteins, Etc by treatment arm as well. Since we do no longer report about nintedanib results, the “post-treatment” condition has not been included, and thus, adding the numbers in the second and third column yields always a lower number than in the first column (since the first lists the results for the whole study, which included the runs of baseline samples from the

standard arm, baseline samples from the experimental arm, and post-nintedanib samples from the experimental arm).

Query #5.In Figure 2, the Kinases driving the phosphoprofiles of responders to paclitaxel or nintedanib is shown. This is an indirect analysis based on detection of phosphopeptides, however, it would be important to show and validate the identification and quantification of kinases directly by mass spectrometry.

We thank the Reviewer for this suggestion. The kinases are now shown in Figure S2, but we believe that it is very important to make an introductory explanation. This query takes us back to the issue of how candidate biomarkers are detected and the need for validation. It also evidences to what extent perspectives behind the same study can differ and highlights that many bridges have yet to be built between proteomic scientists and clinician-scientists to implement proteomics in the clinics

Also, first of all, it is important to tell the Reviewer that all nintedanib data have been removed from the manuscript as suggested by the other 3 Reviewers, in order to focus just on paclitaxel biomarkers.

We understand that, from the view of a proteomic scientist, the logical way to conclude something about detecting or not a kinase in some sample, is to look for that kinase in the mass spectra and show it. What we were trying to do in our study is not direct kinase detection, and this point may require some clarification from our side: what we tried to do since the beginning is to compare phosphoprofiles among two different sets of samples: samples from responders, and samples from non-responders to paclitaxel. Within those phospho-profiles, we wanted to answer if (globally, not at the single-patient level) the differential amount of some phosphopeptides across the sets was potentially

caused by the increased or decreased activity of some specific kinase(s), in order to establish potential mechanistic relationships between some kinase(s)' activity and response to paclitaxel (beyond the fact of identifying biomarkers). To that end, we did, in purpose, KSEA and not kinase detection (in case we wanted to perform direct kinase detection we would have used bead-based assays, RPPAs, or other techniques).

Should we have used a direct kinase detection technique for our study, we are not sure how we could have ascribed the biological differences (profiles) between the two sample sets (responders, non-responders) to one or another kinases, since we would have had the kinases detected at the individual, and not collective, sample level. This is an important difference, since KSEAs detect the kinase activation pattern across sample sets, which is what we wanted to detect (not describe inter-individual differences). Once these (KSEAs-predicted) potential kinases that, in theory, drive the phospho-profiles of responders versus non-responders, validation is required in two aspects:

- 1) Are these kinases detected, directly, in independent sample sets, in responders, more preferentially than in non-responders? (answered in Figure 4)
- 2) How are these kinases related, mechanistically, to paclitaxel sensitivity? (answered in Figures 5-6-7)

Since KSEAs detect activity enrichment of potential kinases behind phosphoproteomes (the logical strategy for our study workflow), the kinases themselves may or may be not detected in individual samples due to the following reasons:

- A) We determined only the phospho-fraction. Some kinases are regulated by phosphorylation and others are not (please see below). Thus, kinases accounting for some phospho-profiles may not be present in the samples that were run
- B) Kinases regulated in tiny amounts may cause important differences in phosphoproteomes. This could be evident in the p-profile analysis (KSEAs). However, regarding the detection of the kinase itself, since the detection

probability is related to the relative amount within the samples, again, kinases driving a found phosphoprofile may not be detected with this strategy.

- C) KSEAs predict the enrichment of one kinase's activity out of several samples' p-profiles, not individual samples. Thus, the putative kinase may be detected in some samples and non-detected in others.

What the KSEAs are telling us is that some kinases may be behind the determined phosphoprofiles in responders comparatively to non-responders. Regardless of whether we find those kinases, or not, in the mass spectra, such statement requires validation, and validation is what is provided in the rest of the manuscript. We show that CDK4 and Filamin A are related to response to paclitaxel, and that is independent of what else we can show in our mass spectra now. Direct kinase-detection techniques would have given us different pieces of information, and maybe we would have come up with several potential candidates for validation – there is no way to know that now – whichever was the case, it would have still required validation. Another reason that made us chose phospho-profile analysis is that potential markers can be extracted from the top-regulated phospho-peptides, such as we did in the case of Filamin A; this piece of information would have been missed if we had gone for direct kinase detection.

Having clarified this, we have gone back to the discovery dataset and this is what was found in the mass spectra:

The candidate biomarkers are depicted in Figure S4 (previous Figure S2). As the Reviewer can appreciate, of those biomarkers, 6 (left-hand side) come from predictions made by the KSEA. Of those, 4 of them have a clear and detectable (by immunohistochemistry) phosphorylation site that accounts for their activation (P70S6K, PKC, AMPK and CaMKIV), and thus, in principle, they could be present in the phospho-fraction that was run in the mass spectrometry. The other two (CDK4 and HMG-CoA reductase kinase) are a different case: for example, the main regulatory mechanism of CDK4 is transcriptional; thus, a phosphoprofile predicted to be driven by high CDK4

activity would imply that such sample has high levels of CDK4, but not necessarily phosphorylated CDK4; thus, it is unlikely that phosphorylated peptides mapping to CDK4 are found in the phospho-fraction of that sample, even if the KSEA suggest that it is caused by increased CDK4 activity. Similarly, HMG-CoA reductase kinase, have complex regulatory mechanisms involving allosteric regulation and others. Finally, in the right-hand side of Figure S4, the Reviewer can see that some other candidate biomarkers were chosen from the regulated phospho-peptides, and among them, one maps to a kinase: pNEK1 (Ser1052).

Thus, it can be reasonable to expect that, at least in some samples, we should be able to find phosphorylated peptides mapping to P70S6K, PKC, AMPK, CaMKIV and NEK1. With the exception of P70S6K, we found phosphorylated peptides mapping to the regulatory regions of those kinases in our experiment. Phospho-peptide intensity was higher in responders comparatively to non-responders in all cases. Regarding P70S6K: we did not find phosphorylated peptides mapping to P70S6K in the whole data matrix; however, S235/236 of S6K was phosphorylated in greater amount among responders compared to non-responders. Since this is a canonic substrate of P70S6K that is the reason why the KSEA shows statistically significant enrichment of P70S6K... even in this case, where the kinase could not be detected. That is, in our view, the advantage or proceeding with KSEAs – they give us reasonably strong candidates for the rest of the study.

We have included these data in a new figure (according to the position in the main text, it is the new Supplementary Figure 2).

We have added the following text in the results section, mentioning the direct detection of kinases:

“Of note, on top of the indirect detection of P70S6K, CDK4, PKC, AMPK1/2 and CaMK-IV activity enrichment by KSEA, a higher amount of phosphorylated peptides mapping to the regulatory regions of these kinases was detected in samples from responders to

paclitaxel comparatively to non-responders (Figure S2). Since the detected phosphorylation sites of these kinases are known to be involved in kinase activation, this finding further suggests the involvement of those kinases in the “responder phenotype”.

Query #6. In Figure 2A-2D, for same kinase different number of phospho peptides (vertical black lines) are shown, e.g., for CDK4, in 2A many more number of peptides are shown compared to 2B, 2C and 2D, it would be important 1) to know how different number of detected peptides influence the analysis and the enrichment score 2) how many unique phospho peptides and thereby corresponding proteins are present in different comparison and if there is any common peptides detected across all conditions and if yes has any biological significance 3) for a peptide to be used in such an analysis was there a minimum number of samples in which the peptides needs to be detected or in other words whether or not peptide patients frequency was used in the analysis.

Thank you for these comments – we can provide further information about the KSEA methodology; however, this concerns again to the fact that KSEA is only a tool used for generation of hypotheses; these hypotheses then have to be validated or rejected. No matter how more or less “relaxed” boundaries are used for KSEA predictions, these predictions are never “valid” or “invalid” – it is only the validation workflow what will confirm some of them and reject the others.

-Regarding point #1: This referee has hit on a key point. Yes, KSEA considers the size of the kinase sets. Thus, only those kinase sets with >5 and <1000 matches in the ranking generated from the peptide data matrix are analyzed. We are aware that this is a liberal cut-off since the default cut-off in GSEA is >15 and <500 . This is because KSEA was run on rankings of small size (~12k peptides) compared to the rankings normally

used in GSEA (~22k genes). At this point we would like to stress the fact that KSEA was used as an *in silico* hypothesis generator in such a way that it served to guide the experimental validations.

-Concerning point #2: Currently, in the manuscript, only 3 comparisons remain: responders versus non-responders in the standard arm; responders versus non-responders in the experimental arm, and responders versus non-responders combining both trial arms. The comparisons involved, respectively, 2151/1027, 2594/1212 and 2757/1252 peptides/unique proteins, respectively. Obviously, many phosphopeptides were detected in patients from both conditions (responders and non-responders) – the differences in their amount (combining all the peptide data matrix) is what leads to the KSEAs to suggest, with more or less statistical significance, that one kinase in particular accounts for such differences. Other peptides are detected just in one of the two conditions, and that also contributes to each KSEA to be, or not, significant. The number of peptides/proteins included in each comparison has been added to the main text

-Finally, with respect to point #3: This is a good point. We did not take into account the patient's frequency since low frequency peptides have mostly zeros in the peptides intensity data matrix (i.e. flat patterns). To perform Kolmogorov-Smirnoff, KSEA generates a two-tailed ranking based on t-statistic obtained from the comparison of peptide intensity values between phenotypes. Thus, peptides with little change (i.e. flat patterns) between conditions are located in the middle of the ranking (i.e. $t \sim 0$) and hardly contribute (little weight) to the calculation of the normalized enrichment score (NES). In contrast, peptides with consistent and relevant intensities changes (i.e. higher and lower values of t) between the conditions compared contribute more weight in the calculation of the NES.

Query #7.For volcano plot (figure 3), what was the minimum number of samples in each group in which a peptide needs to be identified/quantified to be included in the analysis?

Proteins like CDK4, Filamin-A, Vimentin and others as listed in line 300-303 are phosphorylated for their physiological activity and I am wondering how many "if any" phosphorylated peptides from these proteins are identified and if can be labelled in volcano plot?

All those peptides were phosphorylated and detected, and were already highlighted in the previous version (peptides highlighted in Figure 3, listed in Table S2, from which, according to the criteria discussed in Figure S4 legend, a few of them were selected for subsequent validation). It is possible that the Reviewer has overlooked this point; thus, we are happy to explain the process of biomarker selection again:

Both the potential biomarkers identified by KSEAs and those that in the volcano plots have a significant FDR (adjusted $P < 10^{-2.5}$) and >16-fold up-regulation belong to the same class: candidate biomarkers that need to be validated. Regarding the peptides shown in the volcanos, there was no pre-filtering: all peptides, regardless being detected in just one sample or all samples, are used in the analysis. Whether they are present in more or less samples, at higher or lower amounts, or differentially present in responders and non-responders is what determines their FDR. Together with whether something is known about the protein or not that could make us think its relationship with response, and whether it is possible to detect them by immunohistochemistry, is what determines their potential role as "candidate biomarkers" for our subsequent validation steps.

As explained in Figure S4 (Figure S2 in the previous version), out of all the potential biomarkers, we selected those that either had the highest statistical significance, highest regulation, and/or could be determined by immunohistochemistry in the validation set. Sometimes, we can not detect the phosphorylated form (i.e., because no antibody exists) and we go for indirect detection (i.e., we go for total protein; such is the case for example for NEK1 or METTL14). Similarly, sometimes there are not available antibodies against a candidate kinase but there are antibodies against the substrate and thus we search for

the substrate in the validation set (i.e., akin the case of HMG-CoA Reductase). We chose to do it this way because we want these biomarkers to be detected in the clinical setting with the currently available materials. As the Reviewer can appreciate, the left-hand side of the workflow depicted in Figure S4 concerns the candidate biomarkers originated from the KSEAs, and the right-hand side concerns the candidate biomarkers originated from the volcano plots.

All those phosphorylated peptides (the volcanos include only phosphorylated peptides) were already labelled: in Figure legend 3, we mention that those p-peptides that have an adjusted $P < 10^{-2.5}$ and >16-fold regulation are listed in Supplementary Table 3, and are highlighted with red color (present only in responders, not in non-responders) or orange color (present both in responders and non-responders, but in higher amount in the former). Their IDs are listed in Table S3. These tables include all the peptides the Reviewer asks for: P-vimentin, p-Plectin, p-YAP, etc.. .and many more. Out of the considerable list of peptides listed in Table S2 (peptides highlighted in Figure 3 volcanos), we chose to try to validate only a few based on their potential relationship with paclitaxel sensitivity (literature- and knowledge-based).

Query #8. I agree with authors that “Mass spectrometry is not an over-the-counter technology” and immunohistochemistry technique is easily accessible, however, for the validation of the proposed potential biomarkers it would be important to show the identification and quantification of 15 potential biomarkers of sensitivity to paclitaxel (p-P70S6K (Thr389), CDK4, Filamin A, HMG-CoA Reductase, p-Vimentin (Ser56), p-AMPK1/2 (Thr172), p-Pan-PKC (Thr497), p-CaMKIV (Thr196/200), p-YAP1 (Ser127), p-40SRPS6 (Ser235), NEK1, p-Filamin A (Ser2512), RAPGEF6, Plectin and METTL14.

We may be wrong, but this question seems to us like a mix of query #5 and query #7: identification and quantitation of the kinases predicted by the KSEAs in the phospho-peptide data matrix, and identification and quantitation of the phospho-peptides selected from the volcano plots for subsequent validation. The kinases are shown in Figure S2, and the explanation is provided in response to Query #5. Regarding the other candidate biomarkers, please check the answer to the previous Query.

Query #9. In Figure 5A, I am wondering why 0.6nM paclitaxel did not show any significance difference between MDA-MB-231 and MDA-MB-231 CDK4 cells, whereas all others lower (0.2 and 0.4nM) and higher (0.8nM) concentration showed the difference?

The reason is that for some data points the number of experimental replicas is sufficient to show statistical significance whereas in others (i.e., 0.6 nM) is not, according to the amount of cells living and dying at each drug concentration. Regardless, what matters is the comparison of the different IC50, which is significant. We have added in the main text the P-values for all the statistical comparisons between IC50s among the different cell lines.

Nevertheless, we have increased the number of experimental replicas so that the comparisons are now significant at all datapoints. The new figures replace the previous IC50 curves shown in Figure 5.

Query #10. In Figure 6, please clarify whether the quantification is based on spectral

counting or intensity? And what does sum of the average intensities (in spectral counts) means? Also, why the scale is different for y-axis (log10) and for x-axis (log2)?

Protein quantification was performed with MaxQuant which computes intensity-based values. Protein intensities are calculated using the extracted ion current (XIC) values of all isotopic peaks associated with the identified peptide sequences. For the Y-axis, the sum of protein intensity in the bait and in the control were calculated.

Regarding the X-axis and Y-axis scale: Protein intensities have a wide dynamic range due to large differences in their expression/abundance levels and consequently are normally represented in the log10 scale (see for instance PMID 30777892²²). Differential protein ratios on the other hand do not show such large distributions and are frequently transformed to log2 scale. These transformations improve the visualization of the data distribution and that is the reason of choosing each axis scale.

Query #11. For the pull-down experiment, the way data is presented (Figure 6B, 6C) it is hard to judge if there is any significant change in the quantity of target proteins (Filamin A and alpha tubulin) between two conditions. Also, so many different proteins were pulled down along with target proteins which raises the questions about quality of the pull-down experiment. So many proteins are showing similar change, so what was the rationale for selecting CLIP-170 protein for follow up experiment?

The answer to each of these points are provided below. Also, we thank the Reviewer for giving us the opportunity to clarify, once again, the difference between screening experiments and confirmation experiments.

This pull-down is simply a screening experiment in where we search for potential changes in protein partners that can explain the observed phenotype (i.e., increase sensitivity to paclitaxel). The representation is just one more standardized way of presenting pull-down experiments (reviewed elsewhere²³). Since we anticipated little change, because of the fact that the cells in which the pull-downs were performed are isogenic and thus in principle identical except for the overexpression of CDK4 or Filamin A, that is the reason why we provided in Tables S7 and S8 all the intensities and ratios. We invite the Reviewer to check the intensities in those tables. The changes (small or large) are the result of the intrinsic (limited) difference in each cell's biology, but illustrate the point that we were pursuing: in presence of increased CDK4, there is more CLIP-170 bound to Filamin A. We have also added some small changes to the chart in order to add “understandability” to them – we have added a dark bar in the middle to separate the proteins enriched in bait versus non-specific binding, and also added the Log2-fold limit (dotted vertical bar) that represents the lower limit for enrichment set for considering proteins potential candidates.

Regarding the amount of proteins that were pulled-down, in our experience and also for what we found out in existing literature, cytoskeletal proteins are quite sticky; a report published not long ago in a major journal (Current Biology) which aimed to study tubulin-associated proteins identified >3000 proteins in different experiments²⁴; thus, our numbers should did not particularly call our attention as especially high. Furthermore, back in 2013, the paper about the “CRAPome” (proteins non-specifically bound as contaminants across different experiments) seem to increase the number of proteins normally pulled-down in any experiment²⁵; however, since our search for candidate proteins was knowledge-based among those with regulation and statistical significance, we did not proceed to clean-up the data eliminating background contaminants, it was not necessary to find CLIP-170 or MAP4 (see 2 paragraphs below). Of note, CLIP and MAP are hits as well in the 2016 tubulin paper.

As in other experiments in this manuscript, we are not trying to conclude anything out of a proteomic analysis of a pull-down. That would be a faulty and unrevealing approach. We do not believe that any change in the experiment, representation, interpretation, number of replicas, ... or any other factor would change that and would allow establishing any firm conclusion about what leads to changes in paclitaxel sensitivity. That is the reason why, with the screening results, we proceed to confirmatory experiments: Figure 6D, Figure 6F, Figures 7A-D, S10 and S11 are the ones that confirm the role of CLIP-170.

Then, why choosing CLIP-170? As in any other study, when a researcher gets the results from a screening experiment, there are several ways to proceed to the next steps; what determines if the chosen approach was the (a) right one is if what was found at the end makes sense and fits in the full picture. What we chose to do was checking, one by one, the top hits (Tables S7 and S8) in the literature first. That led us finding CLIP-170, a protein of which we were completely unaware. Since there was previous literature showing that it can enhance the amount of paclitaxel bound to tubulin²⁶, the next obvious step was to check if the increase in Filamin A led to increased CLIP-170 binding and increased paclitaxel accumulation – that was the case, and thus, there was nothing left to do.

We also found another potential candidate to explain the mechanism in the hit list – MAP4, a protein that is supposed to modulate the equilibrium of polymerization and depolymerization of microtubules. However, we did not find MAP4 changes (total, phosphorylated, or bound to Filamin or Tubulin). Thus, we did not pursue further mechanistic insights of this protein (data not shown).

It could have happened that 200 candidates made sense in terms of implications in the findings according to the available literature – or, similarly, that no candidate made sense. In such situations, we would have proceeded, one by one, with gain-of-function

and loss-of-function studies, starting from those with greater regulation and/or statistical significance. Fortunately, we did not have to do it this way.

In some situations, even after doing that, it is impossible to tackle the mechanism in a reasonable timeframe, and thus, we would have written a totally different manuscript.

Coming to this point, at the end of the day it is all about how each research group decides their own research methodology to advance through their projects. But the relevant point, again, is that the pull-down experiment is just a tool to find potential candidates to explain the mechanism – the candidate markers could have been extracted from literature instead of a pull-down. It is the validation of at least one of the candidate proteins what constitutes a useful result.

Query #12. For figure 7A, 7B, please show quantification of α -tubulin and acetylated α -tubulin by mass spec. Also, which sites are modified and if there is any other PTMs observed?

Our goal was, in that figure, to determine the specific levels of Lys-40 acetylated tubulin to prove the increase in microtubule stability. Determining the acetylation levels of Lys-40 by western blot or immunohistochemistry is the standardized way of measuring microtubule stability: Tubulin acetylation was described in 1985²⁷. Monoclonal antibodies against acetylated tubulin were already available that year²⁸. The effects of this PTM were already described a year later (increased microtubule stability, decreasing the polymerization and de-polymerization equilibrium)^{29, 30}. Since then, >2100 references have been published about this matter, using monoclonal antibodies as standardized detection method. As long ago as 1995 (27 years ago) manuscripts start reporting using monoclonal antibodies against Lys-40 acetylated tubulin as the standardized method for detection (by western blot)³¹. A quick search in www.biocompare.com reveals >100

available antibodies against acetylated tubulin from different vendors. With all the literature available about this acetylation site, we are not sure of what does it add determining it with a more complex, time- and resource consuming technique such as mass spectrometry. Running the experiment to show the same results by mass spectrometry would increase the experimental time by 5 more weeks and 11000 Euro in reagents and costs. The results shown in Figure 7A and 7B are definitive according to current standards for Tubulin acetylation reporting experiments, and conclusive about the answer to the questions that we were looking to answer (i.e., is there an increase in Tubulin acetylation levels – in other words, decrease polymerization rate – in response to paclitaxel in presence of CDK4 and/or Filamin A overexpression? And, do acetylated Tubulin levels go down when Filamin A is knocked down in CDK4-overexpressing cells?). Then, what is the reason for repeating the experiment with other technique? The technique is more resource-, time- and cost-consuming, and will add nothing to the results or conclusions for the paper. Of note, the identification of TUBA1A-K40 acetylation by mass spectrometry is not trivial. Trypsin is the protease of choice for mass spectrometry-based proteomics due to its high specificity and robustness; it also generates short peptides with a basic Arg or Lys at the C-terminus, ideally amenable for the current proteomics workflow in terms of chromatographic separation, peptide fragmentation and search algorithm-based identification methods. According to the TUBA1A protein sequence, *in silico* digestion in the vicinity of the acetylated K40 site yields a very long tryptic peptide, containing between 58 and 60 aminoacids. This number of aminoacids far exceeds the optimal mass range of 7-35 aminoacids³² and may explain why in repositories such as proteomicsDB (www.proteomicsdb.org) there are only 7 psm of acetylated K40 out of 155,437,510 experimental spectra. Therefore, the first step in the quantification of the acetylated TUBA1A K40 site will require to test other enzymes that provide a peptide within the optimal mass range³³.

We have not looked for any other PTM in any other position in alpha tubulin. This is the PTM that we were looking for since it is known that it translates microtubule stabilization in response to paclitaxel, which is the subject of the manuscript. Surely other PTMs are involved in other aspects of microtubule (or other) biology, but we do not see the point in exploring opening different and unrelated areas of research within the manuscript. We hope that the Reviewer understands this point. We could understand, if the Reviewer is aware of other specific PTM that could affect the conclusions of our study, that he/she thinks that may have experienced a change that should be looked for under our experimental conditions, that we were asked to look for the PTM XYZ, which is known to do XXX, XXX, because of the reasons XXX and XXX. Other than that, we think that an untargeted search looking for other PTMs unrelated to the direction of the question that was being answered, would not add to current results or conclusions of the manuscript.

References

1. Monti S, Tamayo P, Mesirov J, Golub T. Consensus Clustering: A Resampling-Based Method for Class Discovery and Visualization of Gene Expression Microarray Data. *Machine Learning* **52**, 91-118 (2003).
2. Suzuki R, Shimodaira H. Pvclust: an R package for assessing the uncertainty in hierarchical clustering. *Bioinformatics* **22**, 1540-1542 (2006).
3. Quintela-Fandino M, *et al.* 18F-fluoromisonidazole PET and Activity of Neoadjuvant Nintedanib in Early HER2-Negative Breast Cancer: A Window-of-Opportunity Randomized Trial. *Clin Cancer Res* **23**, 1432-1441 (2017).

4. Cao L, *et al.* Proteogenomic characterization of pancreatic ductal adenocarcinoma. *Cell* **184**, 5031-5052 e5026 (2021).
5. Mertins P, *et al.* Proteogenomics connects somatic mutations to signalling in breast cancer. *Nature* **534**, 55-62 (2016).
6. Dou Y, *et al.* Proteogenomic Characterization of Endometrial Carcinoma. *Cell* **180**, 729-748 e726 (2020).
7. Gillette MA, *et al.* Proteogenomic Characterization Reveals Therapeutic Vulnerabilities in Lung Adenocarcinoma. *Cell* **182**, 200-225 e235 (2020).
8. Krug K, *et al.* Proteogenomic Landscape of Breast Cancer Tumorigenesis and Targeted Therapy. *Cell* **183**, 1436-1456 e1431 (2020).
9. Huang C, *et al.* Proteogenomic insights into the biology and treatment of HPV-negative head and neck squamous cell carcinoma. *Cancer Cell* **39**, 361-379 e316 (2021).
10. Wang LB, *et al.* Proteogenomic and metabolomic characterization of human glioblastoma. *Cancer Cell* **39**, 509-528 e520 (2021).
11. Chen YJ, *et al.* Proteogenomics of Non-smoking Lung Cancer in East Asia Delineates Molecular Signatures of Pathogenesis and Progression. *Cell* **182**, 226-244 e217 (2020).

12. Xu JY, *et al.* Integrative Proteomic Characterization of Human Lung Adenocarcinoma. *Cell* **182**, 245-261 e217 (2020).
13. Symmans WF, *et al.* Measurement of residual breast cancer burden to predict survival after neoadjuvant chemotherapy. *J Clin Oncol* **25**, 4414-4422 (2007).
14. Yau C, *et al.* Residual cancer burden after neoadjuvant chemotherapy and long-term survival outcomes in breast cancer: a multicentre pooled analysis of 5161 patients. *Lancet Oncol* **23**, 149-160 (2022).
15. Salvador-Barbero B, *et al.* CDK4/6 Inhibitors Impair Recovery from Cytotoxic Chemotherapy in Pancreatic Adenocarcinoma. *Cancer Cell* **37**, 340-353 e346 (2020).
16. Martin M, *et al.* Adjuvant docetaxel for node-positive breast cancer. *N Engl J Med* **352**, 2302-2313 (2005).
17. Martin M, *et al.* Adjuvant docetaxel for high-risk, node-negative breast cancer. *N Engl J Med* **363**, 2200-2210 (2010).
18. Navarro P, *et al.* Targeting Tumor Mitochondrial Metabolism Overcomes Resistance to Antiangiogenics. *Cell reports* **15**, 2705-2718 (2016).
19. Zagorac I, *et al.* In vivo phosphoproteomics reveals kinase activity profiles that predict treatment outcome in triple-negative breast cancer. *Nat Commun* **9**, 3501 (2018).

20. Pierre P, Scheel J, Rickard JE, Kreis TE. CLIP-170 links endocytic vesicles to microtubules. *Cell* **70**, 887-900 (1992).
21. Huang KL, *et al.* Proteogenomic integration reveals therapeutic targets in breast cancer xenografts. *Nat Commun* **8**, 14864 (2017).
22. Wang D, *et al.* A deep proteome and transcriptome abundance atlas of 29 healthy human tissues. *Mol Syst Biol* **15**, e8503 (2019).
23. Li M, *et al.* MAP: model-based analysis of proteomic data to detect proteins with significant abundance changes. *Cell Discov* **5**, 40 (2019).
24. Yu N, *et al.* Isolation of Functional Tubulin Dimers and of Tubulin-Associated Proteins from Mammalian Cells. *Curr Biol* **26**, 1728-1736 (2016).
25. Mellacheruvu D, *et al.* The CRAPome: a contaminant repository for affinity purification-mass spectrometry data. *Nat Methods* **10**, 730-736 (2013).
26. Sun X, *et al.* Microtubule-binding protein CLIP-170 is a mediator of paclitaxel sensitivity. *J Pathol* **226**, 666-673 (2012).
27. L'Hernault SW, Rosenbaum JL. Chlamydomonas alpha-tubulin is posttranslationally modified by acetylation on the epsilon-amino group of a lysine. *Biochemistry* **24**, 473-478 (1985).
28. Piperno G, Fuller MT. Monoclonal antibodies specific for an acetylated form of alpha-tubulin recognize the antigen in cilia and flagella from a variety of organisms. *J Cell Biol* **101**, 2085-2094 (1985).

29. LeDizet M, Piperno G. Cytoplasmic microtubules containing acetylated alpha-tubulin in *Chlamydomonas reinhardtii*: spatial arrangement and properties. *J Cell Biol* **103**, 13-22 (1986).
30. Maruta H, Greer K, Rosenbaum JL. The acetylation of alpha-tubulin and its relationship to the assembly and disassembly of microtubules. *J Cell Biol* **103**, 571-579 (1986).
31. Gaertig J, Cruz MA, Bowen J, Gu L, Pennock DG, Gorovsky MA. Acetylation of lysine 40 in alpha-tubulin is not essential in *Tetrahymena thermophila*. *J Cell Biol* **129**, 1301-1310 (1995).
32. Swaney DL, Wenger CD, Coon JJ. Value of using multiple proteases for large-scale mass spectrometry-based proteomics. *J Proteome Res* **9**, 1323-1329 (2010).
33. Shah AK, Wali G, Sue CM, Mackay-Sim A, Hill MM. Antibody-Free Targeted Proteomics Assay for Absolute Measurement of alpha-Tubulin Acetylation. *Anal Chem* **92**, 11204-11212 (2020).

REVIEWER COMMENTS

Reviewer #1 (Remarks to the Author): expertise in biostatistics

1. The title refers to “*a subset of patients with increased sensitivity to paclitaxel*”. I am still not clear on what subset this refers to or what cohort this is a subset of. There is no analysis that specifically identifies this “subset of patients”. The **Statement of Significance** claims “*These data can allow personalized treatment decisions*”. Given the claims, the authors need to answer the question: how would one identify such a patient in the general population?

Lacking this, the title should be changed to: “**Phosphoproteomic analysis of neoadjuvant breast cancer suggests that increased sensitivity to paclitaxel is driven by CDK4 and Filamin A**”. The word “suggests” (rather than “reveals”) is preferable because the results being reported appear to be suggestive; the paper is not truly revealing any validated findings. In the authors’ own words:

(line 510) “*our data suggest that CDK4 increases Filamin A levels ...[and] leads to ... increased paclitaxel binding to microtubules. These effects ... lead to mitotic catastrophe, explaining the increased sensitivity to this drug in tumors with elevated CDK4.*”

Indeed, the authors describe their work as “*aiming to find candidate biomarkers of paclitaxel sensitivity*” (line 120), not identifying a specific subset of patients.

2. (line 562) “*the strength of our study is the combination of astringent [sic] external validation in external well-annotated patient series with mechanistic experiments showing biomarker specificity*”. This is a misleading use of the concept of “external validation”. The authors first identified some biomarkers and then performed in vitro experiments showing their specificity. This is indeed a valuable corroboration of the conjectures, but the wording “external validation” as used here is not consistent with the typical usage in the biomarker literature.

3. The authors continue to describe their research as being based on “*the largest phosphoproteomic screening conducted within a clinical trial*”. This is misleading since although the samples originated from a clinical trial, the aspects that defined the trial seem to be irrelevant.

4. The authors have not adequately addressed the fact that the samples from the experimental versus standard arm exhibit distinct properties. The claim is made (line 221) that “*patients allocated to the Experimental Arm were not intrinsically different from the biologic point of view to those allocated to the Standard Arm,...*” However, according to their own analysis, there are at least 2 or 3 distinct groups, and these clusters seem to be at least partially associated with study Arm. The new plot shown in Figure S1A implies the optimal number of clusters is $k=2$ (Note: the authors say *consensus clustering estimates $k=3$ groups*, which is inconsistent with their own interpretation of this figure).

The hierarchical clustering in Figure 1C clearly shows that 22/46 experimental-arm samples are grouped together, so it is a non-sequitur to say this is the consequence of “**two** outlier samples” (as claimed on line 16 of the supplementary figures description file). Further, one cannot “*reject the possibility of significant clusters*” seen in one clustering method by performing a different type of clustering and failing to see the same clusters---each clustering method has its own definition of dissimilarity between samples; none are “right” but each may provide its own perspective. This also applies to consensus clustering: *common implementations of CC perform poorly in identifying the true K* (www.nature.com/articles/srep06207). Clustering algorithms are subjective; there are many choices of distances, linkage and/or numbers of groups affect the results. Related to this, the software `pvclust` appears to be applied to a different distance (correlation) than what is shown in the paper.

5. Figure 3 still makes me uncomfortable. I can accept the description of the large gaps in fold change, but there are still mysteries: the lower limit (flat lines) of $-\log_{10}(p)$ values; why these lower bounds differ between up- versus down-regulated peptides; and to what extent the differences in 3A and 3C might be due to potential biases in the two study arms. The authors do not have a biological explanation, so more digging into the data needs to be done to explain these anomalies---including model assumptions, distributions of the data, and appropriateness of each statistical test. The authors surely feel it is important to understand **all** aspects of the results being reported, so consulting a statistician at this point would be helpful.

Reviewer #2 (Remarks to the Author): expertise in paclitaxel resistance and clinical relevance in breast cancer

I've reviewed the responses to my comments and the changes made to the manuscript and I am satisfied with the responses. I think the manuscript is now acceptable for publication in Nature Communications.

Reviewer #3 (Remarks to the Author): expertise in paclitaxel mechanism of action

All of my questions and concerns were adequately addressed, as best as technically feasible.

Reviewer #4 (Remarks to the Author): expert in proteomics

The authors have addressed my previous points. I have no further comments.

RESPONSE TO REVIEWERS' COMMENTS:

Answers to Statistical Reviewer.

Dear Reviewer,

First of all, please let us apologize for our delay in answering your questions – we got them by the end of July and, unfortunately, Spain virtually shuts down in August. It was not easy to get any statistician “on board” during that month, and we only started working in this manuscript after September 1st. Fortunately, two statisticians from our institution were back short ago and have helped us to solve all your queries, that, again, have helped improving the overall level of the manuscript. We have added those two authors to the author list due to their involvement in the new statistical analyses (Nuria Malats and Sergio Sabroso).

A manuscript in a journal of such prestige as Nature Communication has to be correct in all aspects of the data generation and analysis, as the Reviewer conveys. We agree with this point, and with the facts that the arm imbalances and the statistical tests used to generate the volcano plots in the previous version had not been properly assessed. Now, with the aids of the statisticians, we can confidently state that we have corrected those mistakes. There was indeed an imbalance between treatment arms (although the trial was randomized 1:1, we could extract phosphoproteomic data from only 85 patients, what generated imbalances (with the tumors from the experimental arm being larger, with more advanced nodal status, hormone-positive in a larger percentage, and of higher grade) although non-statistically significant (borderline for tumor size). Also, we did not perform a normality test for the phosphoproteomic intensity distribution; we had previously applied a test assuming normality (T-test) for the comparisons that resulted in the volcano plots. Now, we have confirmed the non-normality of the data distribution, and accordingly, data were compared calculating the median value for each peptide ID

and condition, and Mann-Whitney Wilcoxon test. Also, FDR data are shown on top of raw P - values. Interestingly, this testing “cleans” the screening and the number of shortlisted “hits” is smaller than before.

The great thing is that none of the hits that were subsequently confirmed has changed (i.e., Filamin and Vimentin are still there), and thus no experimental changes or confirmation in external series has been required. Thus, the results and the conclusions of the manuscript have not changed, although now all data and tests are correctly analyzed and plotted, which has been achieved thanks to the thoroughness of the Reviewer. Regardless, we agree with the fact that even if the main results and conclusions do not change, the methods – and all aspects – of the manuscript have to be flawless, and now we think that we have approached that point.

Color code

-Black: Reviewers' queries

-Blue: Authors' answers

-Green: Text from the main manuscript quoted in the answers

-Red: computer code pasted in the answers

List of Queries

Query #1. The title refers to “*a subset of patients with increased sensitivity to paclitaxel*”.

I am still not clear on what subset this refers to or what cohort this is a subset of. There is no analysis that specifically identifies this “subset of patients”. The **Statement of Significance** claims “*These data can allow personalized treatment decisions*”. Given the

claims, the authors need to answer the question: how would one identify such a patient in the general population?

Lacking this, the title should be changed to: **“Phosphoproteomic analysis of neoadjuvant breast cancer suggests that increased sensitivity to paclitaxel is driven by CDK4 and Filamin A”**. The word “suggests” (rather than “reveals”) is preferable because the results being reported appear to be suggestive; the paper is not truly revealing any validated findings. In the authors’ own words:

(line 510) *“our data suggest that CDK4 increases Filamin A levels ...[and] leads to ... increased paclitaxel binding to microtubules. These effects ... lead to mitotic catastrophe, explaining the increased sensitivity to this drug in tumors with elevated CDK4.”*

Indeed, the authors describe their work as *“aiming to find candidate biomarkers of paclitaxel sensitivity”* (line 120), not identifying a specific subset of patients.

We thank the reviewer for making this point. Somehow, we were trying to convey the message that by determining CDK4 and Filamin levels in tumor samples from patients, other physicians could predict the sensitivity to paclitaxel of their future patients. But, as the Reviewer adequately points out, this is not what has been done in this manuscript, and the specific way of how to do it is not reported here.

We have recently started a study in collaboration with the GEICAM breast cancer collaborative group that will specifically aim to answer those points – whether prospective patients can have their drug regimen chosen, or not, in the basis of CDK4 and Filamin immunohistochemical determinations. Which levels, cut-off points, staining protocols, positive and negative predictive values, and other clinical parameters of interest of this intervention will be studied and addressed in this current project; indeed, when such

project is finished, the statement of using CDK4 and/or Filamin to define a subset of patients (characterized by these or those CDK4/Filamin levels) among the general breast cancer population in light of their enhanced sensitivity to paclitaxel will be possible to issue. Until then, as they Reviewer points out, we can only say that these two proteins seem to be implicated in sensitivity to paclitaxel, which is what has been studied in this work.

The title has been changed – the new one is the one suggested by the Reviewer, and we thank him/her for this suggestion: “Phosphoproteomic analysis of neoadjuvant breast cancer suggests that increased sensitivity to paclitaxel is driven by CDK4 and Filamin A”.

We have also changed the statement of significance, the new text being “...these data could be implemented in future studies aiming to personalize chemotherapy treatment in breast cancer,...”

Query #2. (Line 562) “*the strength of our study is the combination of astringent [sic] external validation in external well-annotated patient series with mechanistic experiments showing biomarker specificity*”. This is a misleading use of the concept of “external validation”. The authors first identified some biomarkers and then performed in vitro experiments showing their specificity. This is indeed a valuable corroboration of the conjectures, but the wording “external validation” as used here is not consistent with the typical usage in the biomarker literature.

We thank the Reviewer for detecting this error. In the first “revision round” we were instructed to change all allusions to “validation” to more adequate terms. We totally agreed with the suggestion then, and obviously still agree – it seems that we missed one change in line 562.

We have reworded the sentence. The text is now: "...of our study is the combination of testing the candidate biomarkers in two additional clinically-annotated patient series with mechanistic experiments...".

Query #3. The authors continue to describe their research as being based on "*the largest phosphoproteomic screening conducted within a clinical trial*". This is misleading since although the samples originated from a clinical trial, the aspects that defined the trial seem to be irrelevant.

We thank the reviewer for his/her thoroughness – this was also something that we try to correct all along the text in the previous version. We have modified the text in the discussion, now reading "First, large tumor sample collections have been profiled already taking advantage of phosphoproteomics, such as those included in the studies published by the Clinical Proteomic Tumor Analysis Consortium (CPTAC) ^{1, 2, 3, 4, 5}. Although ours has a considerably large sample size as well, the results are not overlapping and the studies have different objectives: in the CPTAC..."

We have also changed a mention to "the trial" in the abstract – the new text is "... screening in samples from HER2-negative breast cancer patients randomized to neoadjuvant paclitaxel or paclitaxel plus nintedanib (N=130), aiming to..."

Query #4. The authors have not adequately addressed the fact that the samples from the experimental versus standard arm exhibit distinct properties. The claim is made (line 221) that "patients allocated to the Experimental Arm were not intrinsically different from the biologic point of view to those allocated to the Standard Arm,..." However, according to their own analysis, there are at least 2 or 3 distinct groups, and these clusters seem to be at least partially associated with study Arm. The new plot shown in Figure S1A

implies the optimal number of clusters is $k=2$ (Note: the authors say consensus clustering estimates $k=3$ groups, which is inconsistent with their own interpretation of this figure).

The hierarchical clustering in Figure 1C clearly shows that 22/46 experimental-arm samples are grouped together, so it is a non-sequitur to say this is the consequence of “two outlier samples” (as claimed on line 16 of the supplementary figures description file). Further, one cannot “reject the possibility of significant clusters” seen in one clustering method by performing a different type of clustering and failing to see the same clusters---each clustering method has its own definition of dissimilarity between samples; none are “right” but each may provide its own perspective. This also applies to consensusclustering: common implementations of CC perform poorly in identifying the true K (www.nature.com/articles/srep06207). Clustering algorithms are subjective; there are many choices of distances, linkage and/or numbers of groups affect the results. Related to this, the software pvclust appears to be applied to a different distance (correlation) than what is shown in the paper.

We thank the Reviewer for bringing up this issue, and indeed, we agree with the point that our previous interpretation was not accurate, and that Figure 1C shows two clusters. There were, indeed, some imbalances between study arms in some clinical characteristics, resulting from the fact that although the trial randomized patients 1:1 and the main characteristics were balanced among arms, only 39/65 and 46/65 samples were successfully profiled in the standard and experimental arm, respectively. The patient cohort resulting from the profiled patients is now described in Table S1 and displays imbalances in tumor size, nodal status, and other features, although none of them statistically significant. We have incorporated a number of changes in the manuscript because of that, and provide explanations below. Also, the data shown in Figure S1 simply show that by using other methodologies we are unable to find the 2 clusters shown in Figure 1C. This does not mean that there are not clusters, only that

we were unable to find them using other methodologies and thus the variables behind that cluster may not be robust (probably reflecting imbalances between arms, but not statistically significant). However, whether the samples cluster or not, as we explain below, has little effect in the study results (although our interpretation of the findings was wrong). We provide at the end of this query a brief explanation about why, in our view, it is unlikely that treatment arm imbalances lead to any significant changes in the main results or conclusions of the manuscript.

Introduction

-The main reason we performed unsupervised hierarchical clustering was to detect whether we observed clear differences in the samples according to the hospital in which they were obtained. This is a common problem in translational oncology – since most studies are multi-centric, it is important to warrant that sample harvesting and preservation are as homogeneous as possible, in order to maximize the chances of obtaining good results from the screening. Even in worst case scenario (i.e., detecting that each hospital applied different procedures, what translated into different biological properties and signals in each sample batch, such as for example hypoxia hallmarks in kinase signaling observed in hospitals in which the procedure of harvesting and snap-freezing took longer), this would have only impact if we aimed to draw the study conclusions just in the basis of our screening (which we did not – we proceeded to study the screening hits in two additional patient series and also with experimental work, Figures 4-7). If the samples are well collected, homogeneously across hospitals, then, most likely the number of hits that will need external confirmation will be low; if the samples are not so good, the additional work will be higher. It is not that there is a “landmark reference” of how many hits out of the screening hits are finally confirmed that can deem a study as “good” or “bad”; rather, the fact that some hits are actually useful and show a potential role as biomarkers in external series with *in vitro* experimental support can serve the purpose of saying that the research approach was successful. In

our case, there didn't seem to be major differences among samples according to the hospital of origin, and, it may have played a role or not in it, but this led to the definition of two potential biomarkers and their potential mechanism of action. Would we have obtained the same conclusions starting from a not so-well preserved, or an even better preserved and homogeneously harvested set of samples? Maybe yes, maybe no; it is not possible to know now, but it possibly does not matter. Whether the samples were biased or not by the hospital of origin is probably not very relevant to the study or even the reader of this manuscript, since – and forgive us for being so reiterative – at the end we were able to detect some biomarkers out of the screening data. But we think it is good that the readers have access to this piece of information. We are open to Reviewer and Editorial feedback in case they consider that we should remove Figures 1C and S1 in case they are not deemed particularly informative for the reader.

-The second aspect of the clustering regards to the point of sample grouping according to the study arm. Often, in large randomized clinical trials, we can observe study sub-analysis aiming to define prognostic or predictive markers. When such sub-studies are performed, since most of the times they rely on biological samples (blood, tumor, etc) obtained from the study participants, it is common that the investigators have not obtained 100% of the planned samples or that they have lost some samples because of sample quality or samples procedures. In that case, it is mandatory to check whether the samples from the sub-study are similar to the whole study cohort, so that robust conclusions can be drawn. Say a clinical trial with 50% of the patients being hormone-positive and 50% hormone-negative; if the sub-study is biased and the investigators lose a significant proportion of hormone-positive samples (i.e, the sub-study is done with samples that are for example hormone-negative in 75% of the cases), the conclusions of the sub-study will not be valid for the whole trial. Why is this a problem? Because in those cases, it is common that the conclusions of the sub-studies are issued without any further testing in other series, and are issued as conclusions of the main trial. That is

why, in order to at least conclude that the sub-study is representative of the principal study, they make sure that there are not biases in the sub-study cohort.

We have tried to determine whether the sample collection that we used for our research was, or was not, representative from the CNIO-BR-003/GEICAM-2010-10 trial (<https://pubmed.ncbi.nlm.nih.gov/27587436/>), because it is usual practice to do so in clinical studies. It seems that we cannot conclude such a thing for sure in light of the presented data in Figures 1 and S1, and the new table presenting the data of the study sub-cohort with “valid” samples (Table S1), but it is not particularly relevant for the work, because of one main reason and three secondary reasons. The main reason is that we are not trying to issue any classification of the samples, and thus, whether they cluster or not, in two or more groups, is not very relevant; showing the clustering is only aiming to provide descriptive parameters of the sample collection. The secondary reasons are: 1) we do not try to conclude anything right out of the screening; 2) we do not try to conclude anything out of the screening for the CNIO-BR-003/GEICAM-2010-10 trial, but for the general triple-negative breast cancer population, and that is why we performed our studies in additional patient series and *in vitro* models; 3) and, as the reviewer mentions in the previous and present Letter, it is irrelevant that our discovery collection was constituted by samples from a clinical trial. Representative from the CNIO-BR-003/GEICAM-2010-10 trial or not, this sample collection led us to obtain a few candidate hits, filter two of them as potential biomarkers and do subsequent preclinical work that was consistent with the clinical phenotype (increased chance of response to paclitaxel in tumors with high CDK4 and/or Filamin A).

Going to the specific points issued by the Reviewer (specific changes in the manuscript are detailed below this sub-section):

- 1) Trial arms imbalance and clustering: We have included what we consider a relevant piece of data: since the CNIO-BR-003/GEICAM-2010-10 enrolled 130 patients, but this study only managed to obtain mass spectra from 85 of them,

and, as the Reviewer adequately points out, it is irrelevant whether the samples came from a clinical trial or not, we have provided a supplementary table (new Table S1) where we include the clinical and demographic characteristics of these 85 patients, instead of referring to our previous publication reporting the clinical/demographic characteristics of the whole trial. Although the trial randomized patients in a 1:1 manner, the “sample dropout” alters this ratio, and thus this may explain why there is now imbalance among several patient characteristics such as tumor size, nodal status, menopausal status or tumor grade (although not statistically significant) across both arms, what may explain why the clustering seems to suggest differences according to study arms (please see below).

- 2) Also, we agree with the fact that the data shown in Figure 1C suggests the grouping of 22 of the 46 run samples from the experimental arm. This was studied by unsupervised hierarchical clustering. In order to further study this potential grouping, we analyzed the data using two additional methods: 1) Consensus Clustering, and 2) pvclust. We did not observe again the same grouping, what probably means that this aggrupation is not robust. That is the only thing we wanted to state – maybe we used the wrong words, and we have changed the narrative about it. It is true that the optimal number of clusters by CC, according to Monti et al. seems somehow “diffuse”. The shape of the CDF in Figure S1A, shows gradual jumps between 0 and 1, reflecting the lack of stability in cluster membership. Figure S1B shows that the largest relative change in the area under the CDF curve occurs at $k=3$ (above a 100% increase) but remains significantly different from 0 as the number of clusters increases, again reflecting poor stability. The same is true for Figure S1C, where there does not seem to be any clear clustering pattern. We enclose here what happens when we vary the value of k : as it can be seen from the full Consensus Clustering results, increasing k

does not reveal any consistent clustering beyond $k=3$. We include below the C panel of Figure S1 for the Reviewer's convenience.

In fact, the relative change in the area under the CDF curve decreases up to $k=10$, which is the maximum k value we employed in the study. In case the two clusters observed in Figure 1C were robust, we would have expected to observe them consistently when we change k , but we didn't. Three "clusters" may be the most "robust" number of clusters but that does not mean that $k=3$ is an optimum

number (two clusters are composed of one sample each, and the remaining samples do not seem to have any further “grouping”). Testing the clustering in different ways may add little to the interpretation of the clustering (since, as the Reviewer adequately points out, “clustering algorithms are subjective; there are many choices of distances, linkages, and/or number of groups that affect the results”), but, most importantly, to the results or conclusions of our work. The two reasons behind performing the clustering, as mentioned, were to test, to a reasonable extent, whether there seemed to be, or not, any factor that suggested that samples from different hospitals were different, and whether the samples from both arms were very different or similar; but, whether there were homogeneous or not across hospitals, or across trial arms, both options are valid to conduct the screening and are just informing the reader about general aspects of the samples. They are just minor pieces of information. They would only invalidate conclusions if we had non-homogeneous samples and we were trying to convey something directly out of the results of the screening, but we have not done such a thing.

We do not want to turn this into a dissertation about how to make clusters, the best methodology for clustering, or how clusters should be interpreted, because it is far from the objectives of the manuscript (find biomarkers for paclitaxel and make some sense of them from the preclinical point of view). Clustering methods are just tools that we use in our study, but we do not research about clustering methodologies. We use them, and try to interpret them within the framework of our study, but whether our samples cluster or not is not relevant to the results or conclusions of the study. What we have tried, though, is to provide readers with the open data (raw data already uploaded in the repository) and a basic analysis of the screening sample set, adjusting our interpretation of the clustering to that one recommended by the Reviewer, acknowledging that the interpretation of clusterings is somehow subjective, that different clustering methods may yield

different results, that clustering methods may have limitations as discussed in the manuscript of Consensus Clustering limitations (www.nature.com/articles/srep06207) and that the screening sample set may not be 100% representative of the CNIO-BR-003/GEICAM-2010-10 clinical trial, but this is not highly relevant to the results or the conclusions of the study. We agree with the fact that hierarchical clustering suggests 2 clusters, but then, faulty consensus clustering method, or pvclust, do not show robust clustering. Probably this disagreement stems from the fact that although there are imbalances between the two treatment arms, they are not statistically significant. We did not elaborate further about clustering methodology since we think that it is a tangential matter to our study.

- 3) Distances in clustering algorithms: The reviewer is correct: the different clustering algorithms depend completely on the chosen parameters. In our case, all the clusters have been done using the same distance (correlation), both the cluster shown in Figure 1C and the ones used to check the consistency between groups (the plots shown in Figure S1 and obtained by consensus clustering and pvclust). We use correlation in all three clustering methodologies that we used. The unsupervised UPGMA was ran in Morpheus using Pearson correlation (<https://software.broadinstitute.org/morpheus/>). Consensus Clustering and pvclust were executed in R. We include below the lines of R code executed for Consensus clustering and pvclust. Both functions employ correlation:

#Consensus clustering

```
res <- ConsensusClusterPlus(exprs(eset),  
maxK=10,  
reps=500,  
pItem=0.8,  
pFeature=1,  
title=title,  
innerLinkage="average",  
finalLinkage="average",  
clusterAlg="hc",  
distance="pearson",
```

```
seed=1262118388.71279,  
plot="pdf",  
verbose=T)
```

```
#pvclust
```

```
res<- pvclust(data,  
method.hclust="average",  
method.dist="correlation",  
use.cor="pairwise.complete.obs",  
nboot=1000,  
parallel=FALSE,  
r=seq(.5,1.4,by=.1),  
store=FALSE,  
weight=FALSE,  
iseed=NULL,  
quiet=FALSE)
```

- 4) Concluding that we “reject the possibility of significant cluster” seen in one clustering method by performing a different type of clustering: Of course, from a purely “logical” point of view, we agree that by not observing a pattern with methods B and C does not reject the fact that a pattern was observed with method A. We simply aimed to state that we are not fully convinced that there is a strong biological signal behind the 22/46 sample grouping observed with method A, and that the results performed obtained with methods B and C seemed to support such statement. Now, we have included Table S1, where we show that there is certain imbalance between the treatment arms, and we narrate how one clustering method suggests clustering while other two methods do not (previous claims were not sufficiently supported, but now we think that Table S1 data serve as the basis for supporting our narrative). We hope that the Reviewer finds the new narrative less conflicting.

In summary, we agree with the Reviewer in the fact that hierarchical clustering suggests arm imbalance; we have explored in more depth such imbalance (Table S1), and

elaborated further about the fact that the lack of statistical significance in the imbalanced characteristics may account for the disagreement between clustering methods. However, we do not believe that this imbalance impacts the results of our study; at the end of the day, the candidate hits resulting from the screening are the candidate hits of the screening, may them have come from a “perfect” or a “no-so-perfect” screening sample set. It is the subsequent experiments (Figures 4, 5, 6, 7 and Supplementary Figures 2 to 11) what supports or rejects their role as potential biomarkers. A “not so perfect” screening may increase the number of potential hits (i.e., noise), and, if it is “bad enough”, it may even occur that no hit is confirmed, but that only impacts the amount of work that has to be done downstream of the screening (i.e., more “false hits” equals “more work”). But, if the downstream work is state-of-the-art (and three independent reviewers have not issued any further queries about it) then the conclusions (acceptance or rejection of the hits as potential biomarkers) should be accepted as robust.

Summary of changes

-Results section, “Discovery patient set for phosphoproteomic analysis” sub-section.

We have added text mentioning how the patient cohort that yielded valid samples was somehow different to the whole NCT01484080 trial, leading to certain imbalances between some clinical characteristics of the patients that received paclitaxel monotherapy and paclitaxel+nintedanib.

“.... described elsewhere⁶. Although the NCT01484080 randomized patients 1:1, and thus the main clinical and pathologic characteristics of the patients were well-balanced among both treatment arms, the patient cohort resulting from the valid samples was slightly imbalanced (Table S1): tumors with valid samples from patients that received paclitaxel monotherapy (N=39) were somehow smaller, with less involved axillary nodes, and of lower grade and replicative fraction than those from patients that received the

combination (N=46); however, these differences did not reach statistical significance. The flow chart in Figure....”

Also, below, we have modified the text concerning our interpretation of the clusterings, highlighting now that different groups can be appreciated according to the treatment arm.

“In order to search for potential differences in sample handling and/or preservation procedures across the different hospitals in which the samples were harvested that could result in signaling alterations (e.g., signaling changes in stress kinase pathways caused by tumor ischemia⁷), hierarchical clustering was performed with the phospho-peptides intensity data matrix (Figure 1C). Hierarchical clustering did not show that the samples seemed to be significantly grouped by the hospital in which they were harvested (Figure 1C). Given the imbalance in certain patients’ characteristics when they were classified by treatment type (Table S1), it was not surprising to observe clustering of the samples by treatment arm (Figure 1C). In order to explore further these clusters, we studied the sample clustering relying in other methodologies: Consensus clustering⁸ (Figures S1A, S1B and S1C) and pvclust algorithm⁹ (Figure S1D), however, did not find robustness in the clustering associated with the trial arm (Figures S1A, S1B and S1C). ”

-Discussion section

We have added a long paragraph summarizing all that has been described above about how both treatment arms are somehow imbalanced, although not statistically significantly, and how this may have impacted in sample clustering. Also, the limitations of the clustering methods are discussed and referenced; finally, we add our discussion about how we do not think that the results of the study are much affected by those facts.

“Regarding the discovery experiment, another point of criticism could be raised, which is the imbalance in certain clinical/pathologic characteristics between the patients that

received paclitaxel monotherapy or combination treatment (Table S1). Correlative studies are often performed with the samples originated in large randomized trials, and, usually, it is uncommon to observe a 100% success in sample retrieval or sample validity. Thus, the question of whether the patient characteristics of the patient sub-cohort constituted by those with valid samples resembles the full trial, or not, is normally addressed in such studies in order to be able to conclude whether the obtained results apply, or not, to the full trial. We can't confidently conclude that the non-statistically significant differences in tumor size, grade, nodal status or Ki67 resulted in meaningful biases in the discovery set experiment, since although hierarchical clustering showed sample clustering by the treatment arm (Figure 1C), other clustering techniques (Figure S1) did not. Clustering algorithms have limitations¹⁰ and are subjective, since there are many choices of distances, linkage and number of groups that can affect the results. The differences obtained when the three clustering methods were applied may stem from the fact that the imbalances between treatment arms were not statistically significant, or from the cluster methodologies *per se*. Nevertheless, potential biases in the discovery patient set and disagreement between clustering results when different methodologies are used would be relevant if we were aiming to establish conclusions just in the basis of the screening, or attempting to extract conclusions about the biomarker role of Filamin and CDK4 just for the NCT01484080 trial based only in this screening, or trying to classify patient subgroups in the basis of the discovery set. An imbalanced discovery set can impact as well in the percentage of "hits" that are confirmed out of the screening candidates (introducing noise; i.e., yielding many hits that are not subsequently confirmed; in our study only 2 out of 11 were so). However, since our objectives were to determine potential biomarkers for the general TNBC population treated with paclitaxel, and understand them from the biological point of view, filtering the candidate biomarkers through two additional external patient cohorts and preclinical experimentation can serve those purposes, and it is probably not very relevant whether Filamin A and CDK4 were originated from a larger or shorter list of candidate biomarkers resulting from the

discovery set. Thus, we think it is unlikely that such imbalance affects in any manner our results.”

-Methods Section, Statistics sub-section

We have added the description of the statistical tests used for Table S1.

“Clinical and demographic characteristics of the study patients by treatment arm were compared by the Mann-Whitney, Chi-Squared or Fisher’s test as appropriate”

-Supplementary material

We have added Table S1, which includes the comparison of the main clinical/demographic patients’ characteristics of those whose tumor yielded valid phosphoproteomic spectra (N=39 and N=46; paclitaxel monotherapy and combination arms, respectively)

Supplementary Table 1: General clinical and demographic characteristics of the patients from the NCT01484080 trial that yielded valid samples for the study.

	Standard Arm (N=39)	Experimental Arm (N=46)	P value
Treatment	Paclitaxel	Paclitaxel + nintedanib	N/A
Age (median, range)	48.8 (30.6 – 64.2)	47.0 (31.1 – 79.2)	0.816 (Mann-Whitney- Wilcoxon)
ECOG 0/1	39 (100%)	46 (100%)	N/A
Menopausal status			0.280

Pre-menop.	20 (51.3%)	30 (65.2%)	(Chi-Square)
Menopausal	19 (48.7%)	16 (34.8%)	
Hormonal receptors			0.508 (Chi-Square)
ER and/or PR +	29 (74.3%)	38 (82.6%)	
TNBC	10 (25.7%)	8 (17.4%)	
Nodal status			0.220 (Fisher's test)
N0	19 (48.7%)	19 (41.3%)	
N1	20 (51.3%)	22 (47.8%)	
N2	0 (0%)	4 (8.7%)	
N3	0 (0%)	1 (2.2%)	
Tumor size			0.092 (Fisher's test)
T1	0 (0%)	0 (0%)	
T2	32 (82.1%)	29 (63.0%)	
T3	7 (17.9%)	15 (32.6%)	
T4	0 (0%)	2 (4.4%)	
Grade			0.551 (Chi-square)
G1	7 (18.0%)	5 (10.9%)	
G2	19 (48.7%)	27 (58.7%)	
G3	13 (33.3%)	14 (30.4%)	
Histologic subtype			0.495 (Fisher's test)
Ductal	31 (79.5%)	40 (86.9%)	
Lobular	7 (18.0%)	4 (8.7)	
Other	1 (2.5%)	2 (4.4%)	
Ki67 (HR+ Only)	N=29	N=38	0.827
14% or less	10 (34.5%)	11 (29.0%)	(Chi-Square)

>14%	19 (65.5%)	27 (71.0%)	
Pathologic complete response to treatment	4 (10.3%)	5 (10.8%)	1.000 (Fisher's test)

-Figure 1 – legend: we have deleted the interpretation that was written at the end of the legend, since it belongs to the main text (“the lack of any significant aggregation of the dendrogram branches suggests that the sample processing and preserving protocols were quite homogeneous across all study sites (N=14 hospitals)”).

Now the figure legend does not include any text stating an interpretation about the clustering.

-Figure S1-legend: we have eliminated categorical interpretations from the figure legend, and the logic conflict about how the lack of cluster observation in one method invalidates the cluster observation by other method. These matters are discussed, as shown above, in the discussion section. The figure legend is now as follows:

“Supplementary Figure S1: Study of samples clustering according to trial arm by consensus clustering and pvclust. We aimed to study further the sample clusters by treatment arm observed in Figure 1C, using two additional methodologies: consensus clustering (A, B and C) and pvclust (D). **(A)** The Cumulative Distribution Function (CDF) curve under different values of k is shown. At optimal k, the area under the CDF curve will not significantly increase with the increase of k value. **(B)** This plot shows the relative change in the area under the CDF curve under different values of k. **(C)** Finally, this panel shows the consensus matrix. Consistency values range from 0 to 1; 0 means never

clustering together (white) and 1 means always clustering together (dark blue). Each panel results from increasing $k=2$ to $k=10$. According to the consensus CDF **(A)** and delta area **(B)** it could be supported that the optimal number of clusters is $k=3$; two groups are composed of one sample each, and the third group contains all the remaining samples. We did not observe the 22-sample cluster apparent using Hierarchical Clustering. **(D)** In this panel, the values on the dendrogram correspond to *approximately unbiased* (AU) probability p-values (red, left), *Bootstrap Probability* (BP) values (green, right) and *clusterlabels* (grey, bottom). Clusters with $AU > 95$ are considered to be significant and highlighted in a box. In this case, the pvclust algorithm only found one significant cluster.”

Final comments

We would like to state that, while agreeing with you in the fact that all methodologic and statistical aspects should be as purist as they can, and that in a major journal such as Nature Communications everything has to be “flawless”, we believe that whether there are or not clusters, or are more or less clusters, do not mean any major changes to the results or conclusions of the study.

Please allow us to elaborate the statement: we simply aimed to provide biomarkers of activity of paclitaxel and understand the biological rationale behind them. To those ends, we started with a sample collection in which we did a phosphoproteomic screening. A “perfect” sample collection for a screening would be, for example, one in which the “hits” of the screening are those and only those that are then observed in the general population – in *all* patients. From that gold standard, we can go down and include collections where the samples are in bad preservation state, or sample collections where patients have received heterogeneous treatments (not just the drug under study but others in combination, which would certainly add noise), or sample collections that mix different tumor type, or a sample collection that is too small to be representative, and so

on and so forth. In case we had a “terrible” sample collection, we would have had to do a very long and time-consuming work testing many potential biomarkers in external series to confirm just a few (if the collection is bad enough, it may happen that no hit is valid). The source of the screening can thus be a “good” sample collection, a badly designed sample collection, previous *in vitro* experimental work, or even literature search – that is up to the researchers (how do they start their experimental work, what is the source of their hypothesis). Depending on how similar is the screening source to the reality of *all* the patients, the confirmation and experimental work that would follow would be longer or shorter, successful, or unsuccessful. In our case, no matter how homogeneous or heterogeneous the samples were, how many treatment combinations were included, how large or small it was, we started from a sample collection that led to 11 potential hits related to response to paclitaxel, and of those, 2 displayed a positive signal in two other patient series and the experimental work backed up their involvement in sensitivity to paclitaxel. Those are the results, and remain as main results even if we now conclude that our sample collection was terrible – the only implication of such statement would be that we would have come to those 2 hits after a more thorough work, but that does not invalidate the hits. We could have as well taken these two hits (or 200 more) from literature, and yet perform the same immunohistochemistry staining in our patient series and preclinical experiments shown in Figures 4, 5, 6 and 7, and Supp. Figures 2 to 11, and the results would have been the same. What we try to say is that it does not matter how good or bad our initial sample set was, as long as it yielded some hits that served the purpose of performing the experiments shown in Figures 4, 5, 6, and 7, and Supplementary Figures 2 to 11. We erred in how to interpret to what extent our initial sample collection was more or less heterogeneous (and, as discussed in Query #5, also in how we presented the data in Volcano plots) but, as we say, that does not change the results and conclusions of our manuscript. Indeed, hierarchical clustering shows two clusters, and in an attempt to dig more in those clusters, we studied the sample collection with other methodologies, and we were unable to find again that cluster

constituted by 22 samples. Does that mean that the cluster exists, does not exist, is it robust, or is it not robust? In our opinion, it means that that 22-sample cluster is not robust, since when we change the methodology or k we do not consistently see it. In case the Reviewer thinks differently, we would be grateful if he/she suggested a different methodology to prove it. But, above all, we think that this is not relevant at all for the main message of the study, since, as mentioned:

A) we are not trying to establish a classification of the samples of the discovery set;

B) we are not issuing conclusions out of the screening;

C) we are not trying to establish biomarkers for the 130-patient trial out of an imbalance subset of those 130 patients, but for the general TNBC population (and that is why the hits are then studied in two external series);

D) extensive biomarker confirmation work and mechanistic experimentation have been performed with the screening hits; this work is unrelated to the number of sample groups that one can establish in the discovery set.

Query #5. Figure 3 still makes me uncomfortable. I can accept the description of the large gaps in fold change, but there are still mysteries: the lower limit (flat lines) of $-\log_{10}(p)$ values; why these lower bounds differ between up- versus down-regulated peptides; and to what extent the differences in 3A and 3C might be due to potential biases in the two study arms. The authors do not have a biological explanation, so more digging into the data needs to be done to explain these anomalies---including model assumptions, distributions of the data, and appropriateness of each statistical test. The authors surely feel it is important to understand all aspects of the results being reported, so consulting a statistician at this point would be helpful.

We thank the Reviewer for this constructive point. We have consulted a statistician about it, as recommended, and have realized that we have made a terrible mistake in the previous version of this figure – i.e., we did not perform a test for normality of the data, and thus, the statistical test performed for finding out the P-value of the regulated peptides was wrong.

We have now corrected those mistakes and we can confidently say that the statistics of the manuscript are correct – which is important for the whole robustness of the manuscript. Concerning the results, and the conclusions, nothing has changed, since the peptides that ranked up in the top of the monotherapy arm are pretty much the same.

The changes are the following:

-First, we tested the normality of the distribution of the phosphopeptide intensity matrix in the monotherapy arm, whole trial, and combination arm, with the Shapiro Wilk normality test. The null hypothesis was that the distributions were normal, and H1 was that the distributions were abnormal. The results were:

-Monotherapy arm: $W=0.11549$; $P < 2.2 \cdot 10^{-16}$

-Whole trial: $W=0.11704$; $P < 2.2 \cdot 10^{-16}$

-Combination arm: $W=0.12256$; $P < 2.2 \cdot 10^{-16}$

Hence, H0 is rejected in the three cases, concluding that the distributions are not normal. These results invalidate using the T-test model to obtain the P-values to compare the average peptide intensities in the previous version.

Changes in the manuscript

We have re-written the statistics section explaining how the Volcano plots were generated. The new text is as follows:

“In order to compare phosphopeptides up- or down-regulated in responders or non-responders in the paclitaxel arm, the paclitaxel plus nintedanib arm, or in the whole trial, normality of the phosphopeptide intensity distribution was tested with the Shapiro Wilk normality test. H0 was that the distributions were normal, and H1 was that they were not. H0 was rejected in the three cases ($P < 2.2 \times 10^{-16}$ in the three cases), and thus, the assumption of non-normality of the data distribution was adopted. According to this the median value for each phosphopeptide was calculated for each condition (i.e., responders or non-responders, in Arm A, Arm B, or whole trial). The median values were compared with the non-parametric Mann-Whitney Wilcoxon test using 100000 permutations; thus, P-values were calculated with 5 decimal places. The obtained P-values were adjusted by FDR using the Benjamini-Hochberg method to account for multiple testing.

Volcano-plots were depicted using the median log fold-change intensity (X -axis) values and raw P-values (Y-axis). The plots were generated with the GraphPad Prism software version 5.04.”

-Second, since the distributions were not normal, we compared the phosphopeptide intensities with the Mann-Whitney Wilcoxon test. In addition, average intensities of each phosphopeptide were deemed non-representative of the values of each phosphopeptide across the distribution because of the non-normality. Thus, in order to build the data matrix for the Volcano plots, for each phosphopeptide and comparison [standard arm (3A), whole trial (3B) and combination arm (3C)], the median value (instead of average) of each phosphopeptide was calculated for each condition, and compared with the Mann-Whitney Wilcoxon test (instead of T-Test)

Changes in the manuscript

We have re-written the statistics section explaining how the Volcano plots were generated, as explained above.

-Third, statistical significance was defined in the basis of FDR instead of the raw P-value. A cut-off point to pick the “hits” for the subsequent external confirmation and experimentation of FDR=0.25 was chosen, the same as for the KSEAS (Figure 2).

Changes in the manuscript

-Table S3 (previously S2, since we have added Table S1 with the arm-imbalances report) has changed considerably. We have added a new column (Column D) with the FDR value for each phosphopeptide. This table lists all phosphopeptides that have a raw P-value below $10^{-2.5}$ in the three performed comparisons. In the previous version, there were 45, 30 and 11 phosphopeptides listed, for the comparison between responders and non-responders in the standard arm (3A), whole trial (3B) and experimental arm (3C). The new version, given the new statistical test, lists only 36, 8 and 1, respectively.

If we only consider those with FDR<0.25 (Column D), the list is even shorter: 9, 1 and 0. The top-regulated hits (Filamin, Vimentin) are still there.

-At the end of the second subsection of the Results section (“Kinases and phosphopeptides enrichment among responders or non-responders to paclitaxel”) we have amended the text. First, we have mentioned that Table S3 now lists *P*-values and FDRs. Also, where we listed some of the peptide IDs regulated in responders and non-responders (we have eliminated Synaptopodin and Rho GTP ases, which are no longer in the list, and listed other cytoskeletal-related proteins found in the list, such as plectin and Rab7). Also, a few lines below, we have added that we only studied those hits with FDR<0.25.

The new text is as follows: "...Phospho-peptides' IDs with $> 2^4$ (16-fold) regulation, together with their *P*-value and FDR, are listed in Supplementary Table 3. Many peptide IDs map to proteins with little or uncharacterized functional significance; however, a considerable amount of peptides up-regulated in responders to paclitaxel mapped to proteins implicated in cytoskeletal polymerization and re-arrangement such as vimentin, laminin, plectin, tensin, filamin, and Rab7 (group 1 and group 2 phospho..."

....

"... above the significance level in the far-left clouds are observed in Figures 3A, B and C). For subsequent experiments, only those phosphopeptides with $FDR < 0.25$ were studied"

-The candidate hits originated from the KSEAS has not changed (Figure S4, left-hand side). However, since the number of candidate hits originated from the phosphosites comparisons has (Figure S4, right-hand side), we have modified as well this list. Now, only Vimentin, Filamin/Filamin Ser²¹⁵², p-YAP Ser¹²⁷ and Plectin have been included in the subsequent experiments because of having an $FDR < 0.25$.

Also, the legend of Figure S4 (point 5) has been modified, including now that only those peptides with $FDR < 0.25$ were considered as candidate hits

The new text is " Concerning the phospho-sites listed in Table S2, we limited validation to those where, on top of an $FDR < 0.25$, a knowledge-based judgement allowed linking them to some potential pro-survival event in cancer cells or proteins potentially related with the activity of paclitaxel, and just from those proteins where their function is at least somehow characterized **(5)**"

-In addition, Figure S5 (the one with the antibody setup) has been shortened, as now, only 11 candidate biomarkers were tested. The rest of the antibodies have been removed. The Figure legend has been adapted accordingly

-In the 3rd sub-section of the Results section, we have corrected the list of potential biomarkers in the basis of what it is depicted in Figure S4.

The text is now as follows: “In order to translate the phospho-screening results (KSEAs and volcano plots proteins listed in Table S3) to measurable data by immunohistochemistry, we followed our previously described “mass spectrometry-to-immunohistochemistry” approach¹¹. Following the algorithm depicted in Figure S4, the approach yielded 11 potential antibodies for biomarkers of sensitivity to paclitaxel: p-P70S6K (Thr³⁸⁹), CDK4, Filamin-A, HMG-CoA Reductase, p-Vimentin (Ser⁵⁶), p-AMPK1/2 (Thr¹⁷²), p-Pan-PKC (Thr⁴⁹⁷), p-CaMKIV (Thr^{196/200}), p-Filamin A (Ser²⁵¹²), p-YAP (Ser¹²⁷) and Plectin. Antibody set-up and control stainings are shown in Figure S5 A-K.”

-The text containing the results of the association with response of the candidate biomarkers in Set 1 in the main text, has been shortened as well, eliminating the biomarker candidates that are eliminated because of the FDR (which were, regardless, non-significant in Set 1 either).

The new text is now: “We found that the following biomarker candidates were associated with pCR to paclitaxel-based neoadjuvant therapy in Set 1: p-P70S6K (Thr³⁸⁹) (2.89-fold higher chance or achieving pCR for patients in the upper H-score quartile versus patients with H-score in quartiles 2 to 4; $P = 0.037$); CDK4 (2.85-fold; $P = 0.048$); Filamin A (3.28-fold; $P = 0.062$); HMG-CoA Reductase (4.00-fold; $P = 0.064$) and p-Vim (Ser⁵⁶) (2.93-fold; $P = 0.047$). The pCR rate in the whole Set 1 cohort, or divided by breast cancer subtype (luminal, HER2 or TNBC) according to the value of p-P70S6K (Thr³⁸⁹), CDK4, Filamin A, HMG-CoA Reductase and p-Vim (Ser⁵⁶) is shown in Figure 4B. Conversely, despite showing potential association in the KSEA or volcano plot analysis, when translated to immunohistochemical staining the following biomarker candidates did not show association with pCR in Set 1: p-AMPK (Thr¹⁷²) (2.07-fold higher chance or

achieving pCR for patients in the upper H-score quartile versus patients with H-score in quartiles 2 to 4; $P = 0.25$); p-Pan-PKC (Thr⁴⁹⁷) (0.4-fold, $P = 0.12$), p-CaMK-IV (Thr^{196/200}) (1.31-fold; $P = 0.72$); p-Filamin A (Ser²¹⁵²) (1.43-fold, $P = 0.55$); p-YAP1 (Ser¹²⁷) (1.26-fold; $P = 0.62$), and Plectin (0.88-fold; $P = 0.193$)”.

-Everywhere else in the text where we mentioned “15 biomarkers” has been corrected to “11” now.

-The antibodies against the biomarkers that were removed from this version have been removed as well from the Materials and Methods section.

-Supplementary table 6 (previously S5) has been corrected as well, deleting those H-scores for antibody staining that are no longer in the manuscript.

-Fourth, regarding the visual representation of the data (Volcano plots). The P values, FDR values, and fold-regulation are the data that drive the “hit” selection for further experiments, and are shown in Table S3 (previously S2). As expected, with the correct statistical testing, the number of candidate hits is lower, and thus, the table is different (shorter) to that shown in the previous version of the manuscript. The fold-regulation values and P/FDR values are what they are and are not going to change regardless of how they are represented (Figure 3), but we offer here several versions so that the Reviewer understands why the volcanos look how they look.

The number of decimal places of the P value depends on the number of permutations performed in the Wilcoxon test to find out the probability of one event. If 1000 permutations are done, we obtain 3 decimal places; 4 places are obtained with 10000 permutations, and so on. We have included in the manuscript the volcano plots resulting of 100000 permutations (5 decimal places). That gains resolution in the p-values with

lower values (the most significant ones); thus, the “flatlines” disappear. However, in the p values with higher values, there are few changes since the range is in 1-2 decimal places. Again, we are not worried about the non-significant portion of our screening.

Below, the Reviewer can see the volcanos plotted with 5 decimal places

In the following figure, however, the Reviewer can see the charts plotted with 3 decimal places (1000 permutations only).

As the reviewer can see, there are differences in the aspect of the significant portion; several flatlines disappear as we increase the number of decimal places.

There is one additional option, which is plotting the volcanos using FDR instead of P-values, but we think that this skews the view of the data: although a number of data pass the 0.25 FDR (Table S3), most data points are in X-axis (FDR=1, thus, $-\log_{10} \text{FDR}=0$).

This last option does not show flatlines (besides the data that fall in the X axis, i.e., non-significant data with $\text{FDR}=1$). We have chosen the first representation so that the reader can see the raw distribution of the data, which, together with Table S3, makes the reader understand why and which hits were chosen. However, if the Reviewer or the Editorial Board believes that this latter figure looks better, we can change it for the final version. We eagerly await the feedback on this matter.

The chosen version (raw P -values with 5 decimal places), as expected, preserves the gaps in the fold-change, due to the nature of our data, as explained in the previous version. As for the flatlines in the $-\log(P)$ values, there are due to the decimals obtained when estimating the probability of the event during the Mann-Whitney Wilcoxon test. On this scale, it is certainly not detectable, but the flat lines disappear when we look at the 4th and 5th decimal places, being in fact different P -values (not “real flatlines”). The only P -values that are exactly the same are those whose $-\log_{10}(P)$ value is equal to 0, since

they have a raw P -value of 1. Since these values ($P=1$) are of no statistical or technical or biological interest, we are open to remove them as well in case the Reviewer or the Editorial Board think the figure looks better.

Nevertheless, we have examined the literature regarding the “flatlines”. We, and probably the Reviewer, are more used to work with and plot data from gene expression experiments. These experiments usually have less zeros, more data points, more granularity, and less “jumps” (this is just the nature of proteomics, but, as we say, we are not making a dissertation about proteomic methods and data and how does this impact in the visual representation compared to other techniques; this probably would be the subject of a review by experts in the technical field – for us is just one more technology). We have examined other volcano plots that have been published, and here are the examples, with most of them showing flatlines in the non-statistically significant portion (but also in the significant portion) of the data:

A) Mol Neurobiol 2022, 2456-71¹². There are two phosphoproteomic figures here with similar aspect to ours regarding flatlines.

B) Oncotarget 2022:659-76¹³: another phosphoproteomics plot with similar appearance

C) *Frontiers in Genetics*, 2021; 12: 789485¹⁴. Two more volcanos in Figure 2, with flatlines in the significant and non-significant portion of proteomic data.

D) Elife 2017; 6e23242¹⁵: We have even found the same flatlines in gene expression data plotting (Figure 9 of the manuscript, panel D).

Thus, it seems that this effect is quite common. We agree that we made a mistake in the statistical testing and the representation of the data in the previous version, and we deeply thank the Reviewer for detecting this; however, even with the correct tests, the appearance is not much better. What we try to say is that even with the correct testing, it seems that due to the nature of certain data matrixes, *P* values show little shifts, and this is observed in the literature in gene expression or proteomic plots, both plotted with FDR or raw *P*-value. Everybody seems to accept this; furthermore, what it is important is that the screening hits do not change, and the confirmation and experimental work conducted with those hits was state-of-the-art and deemed correct and relevant by the other Reviewers.

Changes in the manuscript

-Figure 3 now contains the Volcano plots performed with median intensity values, Mann-Whitney Wilcoxon test, and raw *P*-values. The Figure legend has been changed

accordingly. The new legend is: “**Figure 3. Volcano plots with regulated phosphopeptides among responders or non-responders to paclitaxel.** Regulated phosphopeptides in patients that achieved pCR versus those that did not in the standard arm **(A)**, whole trial **(B)** or the experimental arm **(C)**. Phospho-peptides with greater than 2^4 (16-fold) up- or down-regulation in one or another condition with Mann-Whitney Wilcoxon $P < 10^{-2.5}$ are highlighted and color-coded for each comparison; their IDs and FDRs are listed by color group and comparison in Table S3”

-Fifth, regarding whether there are differences between 3A and 3C, and in the lower bounds of the distribution, we can think of several explanations:

A) the most obvious: the statistical tests were wrong; now, that they have been corrected, 3C and 3C show similar lower bounds; but 3A still does show differences. However, if we plot FDR, there are no longer differences: in all cases, the lower bound is zero. Thus, the reason is whether the data are plotted with FDR or raw P value.

B) Arm imbalances: as shown in the previous Query, there are certain imbalances, although non-statistically significant, among the clinical and demographic characteristics of both treatment arms. That is one potential biological reason behind the differences in both plots.

C) Finally, and more important from the biologic point of view, 3A plots phosphoproteins related to the response to a single drug. It is logical to think that there may be some unique proteins related with response to one drug. Figure 3C, however, plots proteins related with the response to two drugs. Some proteins may be related with one, some with other, but since the two drugs have radically different mechanisms of action (tubulin polymerization inhibitor – paclitaxel – and tyrosin

kinase inhibitor – nintedanib), it is rather unlikely that unique proteins can explain sensitivity to both drugs or the combo. That is probably why there is no protein with $FDR < 0.25$ in this comparison.

Changes in the manuscript

We have included this point in the manuscript in case any reader is curious about whether the lower bounds of the non-significant portions change in responders and non-responders, may cause flat-lines, and why Volcanos 3A and 3C may be different.

“The imbalances between both arms, and above all, the fact that one aims to find predictors of response to a single drug and the other to a drug combo with two different mechanisms of action (microtubule binder – paclitaxel – or kinase inhibitor – nintedanib) however, may explain why there are differences between global appearance of the volcano plots shown in Figure 3A and 3C (different lower bounds for non-adjusted P-value in responders and non-responders, or apparent “flatlines” in the P-values, which are the results of changes only in the 3rd to 5th decimal place in the P-values of the non-significantly regulated peptides, thus not being really “flat”). Examples of apparent flatlines in the non-significant portion of the data are abundant in the literature reporting phosphoproteomic screenings^{12, 13, 14}, which may be an inherent issue of this discovery technique, but can also be seen in gene-expression volcano plots¹⁵.”

References

1. Mertins P, *et al.* Proteogenomics connects somatic mutations to signalling in breast cancer. *Nature* **534**, 55-62 (2016).
2. Krug K, *et al.* Proteogenomic Landscape of Breast Cancer Tumorigenesis and Targeted Therapy. *Cell* **183**, 1436-1456 e1431 (2020).

3. Huang C, *et al.* Proteogenomic insights into the biology and treatment of HPV-negative head and neck squamous cell carcinoma. *Cancer Cell* **39**, 361-379 e316 (2021).
4. Gillette MA, *et al.* Proteogenomic Characterization Reveals Therapeutic Vulnerabilities in Lung Adenocarcinoma. *Cell* **182**, 200-225 e235 (2020).
5. Huang KL, *et al.* Proteogenomic integration reveals therapeutic targets in breast cancer xenografts. *Nat Commun* **8**, 14864 (2017).
6. Quintela-Fandino M, *et al.* 18F-fluoromisonidazole PET and Activity of Neoadjuvant Nintedanib in Early HER2-Negative Breast Cancer: A Window-of-Opportunity Randomized Trial. *Clin Cancer Res* **23**, 1432-1441 (2017).
7. Mertins P, *et al.* Ischemia in tumors induces early and sustained phosphorylation changes in stress kinase pathways but does not affect global protein levels. *Mol Cell Proteomics* **13**, 1690-1704 (2014).
8. Monti S, Tamayo P, Mesirov J, Golub T. Consensus Clustering: A Resampling-Based Method for Class Discovery and Visualization of Gene Expression Microarray Data. *Machine Learning* **52**, 91-118 (2003).
9. Suzuki R, Shimodaira H. Pvcust: an R package for assessing the uncertainty in hierarchical clustering. *Bioinformatics* **22**, 1540-1542 (2006).
10. Senbabaoglu Y, Michailidis G, Li JZ. Critical limitations of consensus clustering in class discovery. *Sci Rep* **4**, 6207 (2014).
11. Zagorac I, *et al.* In vivo phosphoproteomics reveals kinase activity profiles that predict treatment outcome in triple-negative breast cancer. *Nat Commun* **9**, 3501 (2018).
12. Mees I, *et al.* Quantitative Phosphoproteomics Reveals Extensive Protein Phosphorylation Dysregulation in the Cerebral Cortex of Huntington's Disease Mice Prior to Onset of Symptoms. *Mol Neurobiol* **59**, 2456-2471 (2022).
13. Peiris MN, Meyer AN, Warda D, Campos AR, Donoghue DJ. Proteomic analysis reveals dual requirement for Grb2 and PLCgamma1 interactions for BCR-FGFR1-Driven 8p11 cell proliferation. *Oncotarget* **13**, 659-676 (2022).
14. Hu B, *et al.* Metabolomic and Proteomic Analyses of Persistent Valvular Atrial Fibrillation and Non-Valvular Atrial Fibrillation. *Front Genet* **12**, 789485 (2021).
15. Ku AT, *et al.* TCF7L1 promotes skin tumorigenesis independently of beta-catenin through induction of LCN2. *Elife* **6**, (2017).

REVIEWERS' COMMENTS

Reviewer #1 (Remarks to the Author):

The authors' have addressed primary concerns about certain data analysis claims. They have engaged a statistician and revised several of the analyses made in the process of selecting candidate markers. These changes were largely in the exploratory phase of the research and the top candidate markers continued to perform in downstream analysis, so the original conclusions remain essentially the same. I have no further scientific concerns.